



# Simulating ectomycorrhiza in boreal forests: implementing ectomycorrhizal fungi model MYCOFON into CoupModel (V5)

Hongxing He[1], Astrid Meyer[1,a], Per-Erik Jansson[2], Magnus Svensson[2], Tobias Rütting[1], Leif Klemedtsson[1]

**1** Department of Earth Sciences, University of Gothenburg, Po Box 460, Gothenburg 40530, Sweden

**2** Department of Land and Water Resources Engineering, Royal Institute of Technology (KTH), Brinellvägen 28, Stockholm 100 44, Sweden

a now at: Institute of Groundwater Ecology, Helmholtz Zentrum München, Ingolstädter Landstraße 1, Neuherberg 85764, Germany

*Correspondence to:* Hongxing He (hongxing.he@gu.se)

## Abstract

Ectomycorrhizal fungi (ECM), the symbiosis between a host plant and mycorrhizal fungi, has been shown to considerably influence the C and N flux between soil, the rhizosphere and plant in boreal forest ecosystems. However, ECM is either neglected or presented as an implicit, non-dynamic term in most ecosystem models which can potentially reduce their predictive power.

In order to investigate the necessity of an explicit consideration of ECM in ecosystem models, we implemented the previous developed MYCOFON model into a detail process-based soil-plant-atmosphere model, CoupModel. MYCOFON explicitly describes the C and N fluxes between ECM and roots. This new Coup-Mycofon model approach (ECM explicit) is compared to two simpler model approaches, of which one contains ECM implicitly as an non-dynamic N uptake function (ECM implicit) and the other represents a version where plant growth has a constant N availability (nonlim). Parameter uncertainties are quantified by using Bayesian calibration where the model outputs are constrained to current forest growth and soil conditions for four forest sites along a climate and N deposition gradient in Sweden over 100 year period.

Our results show that the nonlim approach could not describe both the forest growth and soil C and N conditions properly. The ECM implicit/explicit approach is able to describe current conditions with acceptable uncertainty. The ECM explicit Coup-Mycofon model provide a more detailed description of internal ecosystems fluxes and feedbacks of C and N fluxes between plant, soil and ECM. Our modelling highlights the need of incorporating ECM in current ecosystem models. We also provide a key set of posterior fungal parameters which can be further investigated and evaluated in future ECM studies.

## 1. Introduction

Boreal forests cover large areas on the Earth's surface and are generally considered as substantial carbon (C) sinks (Dixon et al., 1994). The sink strength is determined through the balance between the major C uptake and release processes, i.e. plant photosynthesis and both, autotrophic and heterotrophic respiration, and is largely controlled by nitrogen (N) availability (Magnani et al., 2007). For instance, numerous studies have shown that soil nitrogen availability is the main driver for plant and microbial growth (Klemedtsson et al., 2005, Lindroth et al., 2008, Luo et al., 2012, Mäkiranta et al., 2007, Martikainen et al., 1995). Thus a proper description of N dynamics in



ecosystem models is the prerequisite for precisely simulating plant-soil C dynamics and greenhouse gas (GHG) balance (Maljanen et al., 2010, Schulze et al., 2009, Huang et al., 2011). Ecosystem models however vary

considerably in their representation of N fluxes: from a very simplified presentation (e.g. the LPJguess model: Sitch et al., 2003, Smith et al., 2011) to very complex approaches which aim to capture the whole N cycle (e.g. LandscapeDNDC: Haas et al., 2012, CoupModel: Jansson and Karlberg, 2011).

Ectomycorrhizal fungi (ECM) are common symbionts of the trees in boreal forests, known as more efficient than roots in taking up different N sources from soil (Plassard et al., 1991), store vast amounts of N in their tissues

(Bååth and Söderström, 1979), and can cover a large fraction of their host plants' N demand (Leake, 20007, van der Heijden et al., 2008). They have also been shown to respond sensitively to the ecosystem N availability and are generally considered as an adaptation measure to N limited conditions (Wallenda and Kottke, 1998, Read and Perez Moreno, 2003, Kjoller et al., 2012, Bahr et al., 2013). Previous research show ECM can receive between 1 and 25% of the plants' photosynthates and constitute as much as 70% of the total soil microbial biomass, thus can

have a major impact on the soil C sequestration in boreal forests (Staddon et al., 2003, Clemmensen et al., 2013). Overall, the functions and abundance of ECM fungi constitute numerous pathways for N turnover in the ecosystem and considerably influence magnitude and dynamics of C and N fluxes.

However, ECM are so far rarely considered in ecosystem models (for an overview about modelling ectomycorrhizal traits see Deckmyn et al., 2014). To our knowledge, only five ecosystem models have

implemented ECM to a various degree: The ANAFORE model (Deckmyn et al., 2008), the MoBiLE environment (Meyer et al., 2012), the MyScan model (Orwin et al., 2011) and more recently Moore et al. (2015) and Baskaran et al. (2016) ECM models. In the ANAFORE model, ECM are described as a separate C and N pool but not distinguished between mycorrhizal hyphae and mantle. The C allocated from the host tree to ECM is simulated as a zero order function, further regulated by nutrient and water availability. ECM can also facilitate organic matter

decomposition in ANAFORE model. MyScan model uses a similar approach for ECM C uptake and dynamics but does, to our knowledge, not include the influence of water availability on ECM. In both models, ECM transfer N to the host is regulated by the C/N ratios of plant and fungi. In the MoBiLE model, C allocation to ECM is more complex than that in ANAFORE and MyScan, and the N allocation to the host by the ECM can feed back on their C gains. The N allocation to the host plant is described similarly as the other two models. In MoBiLE, mycorrhiza

are further distinguished between hyphae and mantle, but cannot degrade organic matter. Hyphae and mantle differ in their capacity to take up N and the mantle has a slower litter production rate than that of hyphae. Both Moore et al. (2015) and Baskaran et al. (2016) ECM models represent the ECM as a separate model pool and also explicitly simulates ECM decomposition but with much more simple process descriptions and the interaction with environmental functions are neglected.

The overall aim of this study is to present a new version of CoupModel that coupled with explicit description of ECM, and also to investigate how the explicit consideration of ECM affects the overall model performance and model uncertainty. We thus implemented the previously developed MYCOFON model (Meyer et al., 2010, Meyer et al., 2012) into the well-established soil-plant-atmosphere model, CoupModel (Jansson, 2012). The implemented MYCOFON model contains a very detailed description of fungal C and N pools and all major C and N exchange

processes (i.e. litter production, respiration, C uptake, N uptake). Fungal growth and N uptake respond dynamically to environmental functions and plant C supply in the new Coup-MYCOFON model (Fig. 1). This detailed ECM explicit modelling approach (thereafter called "ECM explicit") is further compared with two





simpler modelling approaches: the "ECM implicit" and "nonlim" approach which already exist in CoupModel. The "ECM implicit" does not represent the ECM as a separate pool but incorporating ECM into the roots

implicitly. Plants are thus allowed to take up additional organic N source statically and does not respond to environmental functions. Similar "ECM implicit" approach was used previously in Kirschbaum and Paul, (2002) and Svensson et al. (2008a). The "nonlim" approach assumes an open N cycle and plant growth is limited by a constant N availability (e.g. in Franklin et al. (2014)). These three ECM modelling approaches represent most of current ECM representations in ecosystem models and are tested by four forest sites situated along a climate and

N fertility gradient across Sweden (Fig. 2). Bayesian calibration is used to quantify the uncertainty of model parameters and identify key parameter sets.

## 2. Methods

### 2.1 Model description

The CoupModel ("Coupled heat and mass transfer model for soil-plant-atmosphere systems", Jansson and

Karlberg, 2011) is a one-dimensional process-orientated model, simulating all the major abiotic and biotic processes (mainly C and N) in the soil-plant-atmosphere system. The basic structure is a depth profile of the soil for which water and heat flows are calculated based on defined soil properties. Plants can be distinguished between understorey and overstorey vegetation, which allows simulating competition for light, water and N between plants. The model is driven by measured climate data: precipitation, air temperature, relative humidity, wind speed and

global radiation, and can simulate ecosystem dynamics in hourly/daily/yearly resolution. A general structural and technical overview of the CoupModel can be found in Jansson and Moon (2001) and Jansson and Karlberg (2011). A recent overview of the model was also given by Jansson (2012). The model is freely available at www.coupmodel.com. CoupModel was complemented with an ectomycorrhizal module (MYCOFON, Meyer et al., 2010) which allows to directly simulate the C and N uptake processes of ECM. The MYCOFON model is

described in detail in Meyer et al. (2010), here only key processes of plant and fungal growth, littering and respiration are described.

### 2.1.1 Plant growth in CoupModel

An overview of model functions is given in Table A.1 in the Appendix. Plant growth is simulated according to "radiation use efficiency approach" where rate of photosynthesis is assumed to be proportional to the global

radiation absorbed by the canopy but limited by temperature, water conditions and N availability (eq. 1, Table A.1(a)). Assimilated C is allocated into five main different plant C compartments: $C_{root}$, $C_{leaf}$, $C_{stem}$, $C_{grain}$ and $C_{mobile}$. Same compartments also represent the corresponding N amounts. The "mobile" pool ($C_{mobile}$, $N_{mobile}$) contains embedded reserves which are reallocated during certain time periods of the year, e.g. during leafing. Respiration is distinguished between maintenance and growth respiration where a $Q_{10}$ function response was used

for maintenance respiration (eqs. 2.1, 2.2, Table A.1(a)). Plant litter is calculated as fractions of standing biomass (eq. 3, Table A.1(a)).





### 2.1.2 Fungal C and N pools

The ECM are closely linked to the tree fine roots and consist of a C and N pool. The C pool is distinguished between the mycelia, which are responsible for N uptake, and the fungal mantle, which covers the fine roots tips.

The C pool is the difference between C gains by supply from the plant and C losses due to respiration and litter production (eq. 8.1, Table A.1(b)). Accordingly, the fungal N pool is the result of the difference between N gains by uptake and N losses by litter production and N transfer to the plant (eq. 8.2, Table A.1(b)). Fungal C and N pools are distinguished between mycelia and mantle which is of importance for simulating N uptake (only the mycelia is able to take up N) and also litter production if a more complex approach for simulating fungal litter

production is chosen (see section 2.1.4). The ratio between mycelia and mantle is determined by the parameter $FRAC_{MYC}$ which defines the fraction of mycelia C in total fungal C. For all other N and C exchange processes (growth, respiration, and N transfer to plant) the separation between mycelia and mantle is disregarded.

### 2.1.3 Growth of ectomycorrhizal fungi

ECM growth is limited by a defined maximum, i.e. only a certain amount of tree host assimilates will be directed

to the ECM. This maximum ECM growth is determined by a potential C supply from the plant and limited by N availability (eq. 5.1, Table A.1 (b)). This is defined according to the results from field and laboratory studies, that ECM biomass of mycelia and mantle can be as much as 30-50% of fine root biomass. Besides, ECM growth is driven by sink strength (see overview by Smith and Read, 2008), therefore we use these observations as the base of the calculation of the actual ECM growth: i.e. the model aims to grow ECM biomass to a certain fraction of

fine root biomass (eq. 5.2, Table A.1 (b): $FRAC_{OPT} * c_{frt}$). This is further dependent on the N supply from the ECM to the roots, $f(n_{supply})$. The model thus follow the assumption that plants feed the ECM with C as long as their investment is outweighed by their benefits obtained (Nehls et al., 2008). A minimum C supply to prevent fungi to die during C shortage is guaranteed by the term during time periods when plant photosynthesis is limited and belowground C supply to root and ECM becomes zero (eq. 5.3, Table A.1 (b)). The C supply is defined by a

constant fraction of the root C gain and reduced by the function $f(c_{fungiavail})$ as soon as a defined value of soil available N is exceeded, i.e. in the model the potential ECM growth declines with rising soil N. This scaling function is based on observations from field and laboratory experiments, which showed that the majority of ECM decrease in abundance and functioning when the soil N levels are high (e.g. Wallander, 2005, Wallenda and Kottke, 1989, Högberg et al., 2010).

### 140 2.1.4 Respiration and litter production of ectomycorrhizal fungi

Same approach is used here for ECM and root respiration simulation and is distinguished between maintenance and growth respiration, respectively (see eq. 2 and eq. 6, Table A.1). Two approaches are available to simulate fungal litter production, which differ in complexity. The simple approach (eqs. 7.1, 7.2, Table A.1) uses one common litter rate $L$ for both, the fungal mantle and the fungal mycelia. Consequently, possible specific effects

of the mantle and mycelia tissue on litter production are neglected. The alternative "detailed" approach (eqs. 7.3, 7.4, Table A.1) has specific litter rates for ECM mantle and mycelia ($L_M$, $L_{MYC}$). This set-up is recommended when investigating different biomass ratios between mycelia and mantle and their effects on overall litter production. The fraction between ECM mantle and mycelia is determined by the parameter $FRAC_{MYC.}$ Irrespective of the





approach used for litter production, ECM have the capability to retain a defined amount of N during senescence
(eqs. 7.2, 7.5, Table A.1 (b): $nret_{fungi}$). In this study, the simple approach was applied.

### 2.1.5 Ectomycorrhizal fungal N uptake

ECM can take up both, mineral and organic N. For both N forms, a potential fungal uptake is defined at first,
which is determined by the size of fungal C pool, the fraction of fungal C which is capable to take up N (the
mycelia, $FRAC_{MYC}$), and an uptake rate ($NO3_{RATE}$, $NH4_{RATE}$, $NORG_{RATE}$ (eqs. 11.1, 11.3, 11.4, 11.6, Table A.1 (b)).
This function is based on the assumption that only the fungal mycelia can take up N. Values for $NO3_{RATE}$, $NH4_{RATE}$,
and $NORG_{RATE}$ were derived from published values (Tab. 1). The actual N uptake is dependent on the available
soil N as well as the fungal N demand (eq. 11.2, Table A.1). The N availability function $f(n_{avfungi})$ determines the
fraction of soil N which is available for fungal uptake and is controlled by the parameters $NUPT_{ORGFRACMAX}$ and
$NUPT_{FRACMAX}$. N availability for fungi corresponds to the plant available N (eq. 16, Table A.1), but as fungi have
are more efficient in the uptake of nutrients, the availability is enhanced for both mineral and organic N (eqs. 17.1,
17.2, 17.3, Table A.1). To prevent the fungal N demand to be covered by one N form only, the parameter $r_{NO3}$,
$r_{NH4}$, $r_{LIT}$, and $r_{HUM}$, are included and correspond to the ratio of nitrate and ammonium in total available soil N. If
the potential N uptake exceeds the available soil N, the actual uptake corresponds to the available N (eq. 11.2 and
eq. 11.5, Table A.1 (b)).

### 2.1.6 Plant mycorrhization degree, plant N uptake, and fungal N transfer to plant

According to field investigations, the mycorrization degree can vary considerably between species. For spruce,
typical mycorrization degrees of over 90% have been reported (Fransson et al., 2010, Leuschner, 2004). The
impact of the ECM mantle on fine root nutrient uptake has been controversially discussed but the majority of
studies indicate that the root is isolated from the soil solution, i.e. the nutrient uptake is hampered so that the plant
is highly dependent on ECM supplies (Taylor and Alexander, 2005). Therefore mycorrhization degree is of major
importance when plant-ECM-soil N exchange and plant nutrition are of interest. In the explicit Coup-MYCOFON
model, the mycorrhization degree is calculated as the ratio between ECM C pool and the defined optimum ECM
C pool, multiplied by the defined optimum mycorrhization degree (eq. 9, Table A1 (b)). The optimum
mycorrhization degree needs to be defined with care, as there is often a discrepancy between applied root diameter
in experimental studies and models: in experiments, mycorrhization degrees usually refer to fine roots ≤ 1 mm
whereas models often consider fine roots as roots with a diameter of up to 2 mm.
The mycorrhizal mantle has an impact on the mineral plant N uptake. Generally, plant ammonium and nitrate
uptake is regulated by the plant N demand (eqs. 4.1, 4.2, Table A.1). Actual uptake is estimated by the N
availability function (eqs. 15, 16, 17, Table A.1: $f(n_{avail})$, $f(n_{mhumavail})$) based on the assumption that only a certain
fraction of soil ammonium and nitrate is available for plant uptake. The fungal mantle reduces this availability in
such a way that reduction is highest at maximum biomass. In a balanced symbiosis, the fungus provides nutrients
to the plant in exchange for the plant C supply. In the Coup-MYCOFON model, the amount of fungal N transferred
to the plant is determined by either the plant N demand or, if the plant N demands exceeds the fungal capacity,
the available fungal N (eqs. 10.1, 10.2, Table A.1). This is the amount of "excess" N which is available after the
ECM have fulfilled their defined minimum demand as defined by the fungal C/N ratio (eq. 10.2, Table A.1). This




relation is based on the theory that the fungi will only supply the plant with N as long as its own demand is fulfilled (Nehls et al., 2008).

### 2.2 Transect modelling approach

#### 2.2.1 Three ECM modelling approaches

Three modelling approaches applied in this study differing in their complexity. The basic "nonlim" approach is conducted to test if a plant N uptake can be described as proportional to the C demand of the plants of the respective sites. In this case, the plant N uptake is not regulated by the N availability and N is used from a virtual source exceeding the soil N availability, thus an open N cycle. The "ECM implicit" approach simulates the plant uptaking of organic N which is assumed to be of ECM origin, i.e. ECM are considered implicitly as a N source but they do 195 not physically exist in the model. The rate of the organic N uptake is determined by the plant N demand and restricted by the availability of organic N in the soil humus pools (eqs. 4.4, 4.5, Table A.1). Plants can also additionally take up ammonium and nitrate (eqs. 4.1, 4.2, Table A.1). In the "ECM explicit" approach, fungi are fully physically considered as described above. Fungal growth interacts dynamically with plant growth and responds to changes in soil N availability and soil temperature. ECM fungi can take up both, mineral and organic 200 N forms.

#### 2.2.2 Simulated regions and database

Simulations were performed for four forests sites: Lycksele, Mora, Nässjö and Lungbyhed situated along a climate and N deposition gradient in Sweden (Fig. 2). Climate and site information is given in Tab. 2. Climate data were taken from the Swedish Meteorological and Hydrological Institute (SMHI). Data on forest standing stock volumes 205 as well as forest managements were derived from the database and practical guidelines of the Swedish Forest Agency (2005) and were applied as previously described in Svensson et al. (2008a). Soil C content as well as soil C/N ratio were previously determined by Berggren Kleja et al., (2008) and Olsson et al., (2007) and used to describe soil properties in the initial model setting up. For all simulated sites and for all modelling approaches, the development of managed Norway spruce forests was simulated in daily step over a 100-year period from a newly 210 established to a closed mature forest. The measured 100-year old forest standing biomass as well as soil C/N ratio were used for model calibration. Climate input data were quadrupled in order to cover the entire period and thus climatic warming effects is not considered here. A minimum of specific regional data were used at input values (Tab. 2). Otherwise, model parameters were kept identical between modelling approaches in order to evaluate the general model applicability. An overview of the parameter values is shown in Table A.1 (d) in Appendix. For a 215 more detailed site description and CoupModel setup, see Svensson et al. (2008a).

### 2.3 Bayesian calibration

#### 2.3.1 Overview

We performed a Bayesian calibration for all modelling approaches and sites. In this study, we emphasise the models' predictability in precisely describing the long term C and N in the soils and standing stock in the forest, 220 also aiming at a maximised model flexibility. Measured data including tree biomass and the C/N ratio of soil organic matter are thus used as accept criteria. This allowed us to investigate the distributions and uncertainty of





key parameters of the respective ECM modelling approaches ("nonlim", "implicit", "explicit"), analyse model uncertainties and dependencies between parameters. Uncertainties in parameter values are expressed as probability distributions. The posterior probability distributions of parameters are estimated by considering the prior

distribution and the likelihood function in the calibration procedure. The likelihood function is determined by the measured data on output variables and the respective error estimates of the simulated model output. The Bayesian calibration as applied in this study is briefly described below, for a detailed description of the general methodology see e.g. van Oijen et al. (2005) or Klemedtsson et al. (2008).

### 2.3.2 Bayesian calibration procedure

The data likelihood function which determines the parameter sets being candidate of the posterior distributions is based on the assumption that the model errors, i.e. the differences between simulated and observed values, are normally distributed and uncorrelated (van Oijen et al., 2005). Furthermore model errors are assumed to be additive so that the log-likelihood function reads:

$$\log L = \sum_{i=1}^{n} \left( -0.5 \left( \frac{y_i - f(\omega_i \cdot \theta_i)}{\sigma_i} \right) - 0.5 \cdot \log(2\pi) \right) - \log(\sigma_i)$$

where $y_i$ = observed values, $f(\omega_i \cdot \theta_i)$ = simulated values for a given model input $\omega_i$ and parameter set $\theta_i$, $\sigma_i$ = standard deviation across the measured replicates, $n$ = number of variables measured.

In this study, measured uncertainty of 10% for both, the soil C/N ratio and the standing stock biomass data is used. The uncertainty estimate is low (van Oijen et al., 2005), as our intention was to force the model to simulate tree biomass and soil C/N ratio precisely for a better constrain of posterior parameter distributions for the respective

model approaches and sites. Candidate parameter sets are generated by investigating the parameter space using the Markov chain Monte Carlo method, also the Metropolis-Hastings walk. For each simulation, the model likelihood is evaluated for a certain parameter set. After each run, a new parameter set is generated by adding a vector of random numbers ε to the previous parameter vector:

$$q_{i+1} = q_i + e$$

where $\theta_i$ = previous parameter vector, $\theta_{i+1}$ = new parameter vector, $\varepsilon$ = random numbers.

The random numbers are generated normally distributed having a mean of zero and a step length of 0.05, i.e. 5% of the prior parameter range as proposed by van Oijen et al. (2005). We performed $10^4$ runs for each ECM modelling approach and site to ensure posterior convergence.

### 2.3.3 Model parameters chosen for calibration

The different ECM modelling approaches are calibrated for a comprehensive set of key parameters which are chosen according to their function as regulating factors of the C and N fluxes in the plant-soil-mycorrhiza continuum (Tab. 3). In the "nonlim" approach, the constant N supply parameter: *ConstantNsupply* for the spruce tree, is selected as calibrated parameters. In the "implicit" approach, the fraction of organic N available for plant uptake (*NUPT_ORGFRACMAX*) is included in the calibration based on Svensson et al. (2008a). For the ECM "explicit"

approach, all fungal parameters in MYCOFON including: fungal growth (C and N assimilation and uptake, C and





N losses), overall N uptake and plant N supply, respiration and littering are calibrated. For all three approaches, the humus decomposition rate ($K_H$), the C/N ratio of microbes ($CN_{mic}$), regulating soil mineralization and the fraction of plant C assimilates allocated to the rooting zone ($F_{ROOT}$), regulating fungal growth are also calibrated. Overall, we include a rather generous number of parameters for Bayesian calibration following Klemedtsson et al., (2008) which emphasized the importance of a holistic perspective when considering model parameters. Prior distributions of parameters are assumed to be uniform, i.e. each value is equally probable, with a given minimum and maximum value (Tab. 3). Values were chosen based on either previous modelling applications (e.g. plant parameters determined by Svensson et al. 2008a, b), or literature data (Tab. 3).

## 3. Results

### 3.1 Comparison of the three modelling approaches

#### 3.1.1 General ability to reproduce tree growth and soil C/N

Three modelling approaches show different ability in reproducing current plant growth and soil C/N ratio after calibration (Tab. 3B). The posterior model in the "implicit" and "explicit" approaches show a clear better performance of simulating soil C and N, as indicated by soil C/N ratio, than the "nonlim" approach. The latter tends to simulate a lower soil C/N indicated by the negative mean errors (ME) in the posterior model (ME, difference between the simulated and measured values) (Tab. 3B). The ME by the "nonlim" approach is also two to five times higher than that of using the "implicit" and "explicit" approach (Tab. 3B). The "nonlim" approach tends to overestimate the plant growth as the posterior mean of ME for plant C is always positive, while the "implicit" and "explicit" approaches tend to show an underestimation (Tab. 3B).

All posterior models underestimate soil C/N for the northern sites which are generally N more limited, but gradually switch to overestimation at the southern sites. The model with "nonlim" approach simulates a better plant growth for the most southern site, Ljungbyhed than the other sites. Modelled plant growth at Ljungbyhed show overestimation by "implicit" approach but change to underestimation when "explicit" approach is used (Tab. 3B). The acceptance of model runs in posterior is higher for the "nonlim" approach (25 to 48%), and the "implicit" (42 to 50%), followed by "explicit" approaches (30 to 33%) which can be explained by the model complexity, i.e. the more processes and parameters included for calibration, less likely of finding an accept combination of parameter sets. No major differences could be found for the summed log-likelihood for both calibration variables (Tab. 3B).

#### 3.1.2 C and N budget

Modelled major ecosystem N fluxes and soil C, N balance in the posterior are shown in Fig. 3. In general, the "nonlim" approach show a much larger uncertainties in the modelled N fluxes than that of ECM "implicit" and "explicit" approaches. The "nonlim" approach simulate soil sequestration of N, up to 2 g N m$^{-2}$ yr$^{-1}$ for all the sites, but much lower or close to zero are found when using other two modelling approaches (Fig. 3). Soil N is expected to reach a steady state over a period of 100 years (Svensson et al., 2008a). Therefore the "nonlim" approach largely overestimates the soil N sequestration. This can be attribute to the assumed "virtual" constant N uptake from the unlimited source. According to our model prediction, this "virtual" N fraction accounts for 20 to 30% of the total plant N uptake. Besides the simulated soil C balance by "nonlim" approach also contrasts with





that of soil N, where the soil sequestrate C at most north site, Lycksele but loss C at a rate of 6 to 17 g C m$^{-2}$ yr$^{-1}$ for the other three sites (Fig. 3). Therefore soil C and N are not in a steady state and decoupled in the "nonlim"

approach over the simulated 100 year period.

However, the "implicit" and "explicit" approaches show a strong coupling between soil C and N (Fig. 3). i.e. in "implicit" approach, Lycksele and Mora sites overall loss soil C by 6 and 5 g C m$^{-2}$ yr$^{-1}$, while Nässjö and Ljungbyhed sites gain soil C by 3 and 13 g C m$^{-2}$ yr$^{-1}$. Similarly, Lycksele and Mora loss N by 0.2 and 0.1 g N m$^{-2}$ yr$^{-1}$ while Nässjö and Ljungbyhed gain N by 0.3 and 0.6 g N m$^{-2}$ yr$^{-1}$. For "explicit" approach, soil C and N losses

at the two northern sites are slightly higher than that in "implicit" approach (Fig. 3). In contrast with the "implicit" approach, the two southern sites also show an overall minor C and N losses with large standard deviation (Fig. 3). Modelled N litter production increase by 1 to 30% compared to the "implicit" approach but N losses due to uptake and leaching also increase by 10 to 50 % (for Lycksele and Ljungbyhed, respectively, Fig. 3). The increased litter addition of easily degradable C and N stimulates microbial activity thus led to a higher microbial respiration,

which explains the minor losses of C and N in the southern sites in "explicit" model. The higher N leaching in "explicit" model can be attributed to a higher uptake from organic N (eqs. 11.5, 11.6, Table 1.B) and also a stimulated microbial growth thus increase net mineralization, both of which leaves more mineral N in the soil (Fig. 4).

Simulated plant gross primary production (GPP) using "explicit" and "implicit" approaches both show an

increasing trend from North to South due to a more favour climate and N availability for Spruce forest to grow, but "explicit" approach show a predicted higher GPP than the "implicit", i.e. 7% in Ljungbyhed to 12% in Lycksele (Fig. 5). C losses from autotrophic respiration are lower in "explicit" approach (Fig. 5). Trees in the northern regions seems slightly more efficient in taking up C shown by the higher biomass efficiency (NPP/GPP, Fig. 5). Overall, our results show explicitly account for ECM in boreal forest ecosystem can have a considerable

impact on the predicted C and N dynamics both for the plants and soil.

### 3.2. Posterior parameter distributions

### 3.2.1. Posterior distributions of common parameters

The posterior distributions differ from the prior uniform distributions for all modelling approaches and parameters, reflecting the efficiency of Bayesian calibration (Fig. 6 and Fig. 7). The posterior *constantNsupply* parameter in

the "nonlim" approach show the lowest values at Lycksele and highest at Ljungbyhed, which means a higher N supply is necessary at the southern sites to explain the observed tree biomass and soil C/N ratio. No significant differences in parameter values: microbial C/N ratio ($CN_{MIC}$), humus decomposition coefficient ($K_H$), and the fraction of C allocated to roots, $F_{ROOT}$, in the "nonlim" approach are found for the different sites (data not shown). The organic N uptake parameter in the "implicit" and "explicit" approaches ($NUPT_{ORGFRACMAX}$) show an opposite

pattern with highest values for Lycksele and lowest for Ljungbyhed (Fig. 6). Similar posterior distribution of $NUPT_{ORGFRACMAX}$ parameters are found in both approaches except a larger uncertainties in the "explicit". Besides, a much wider range of posterior parameter values are found for the northern sites than that of the southern sites (Fig. 6). This also explains the smaller simulated ME of soil C/N in the southern sites (Tab. 3). Both approaches demonstrate that the plant and soil conditions at the northern sites could not be simulated without an enhanced

uptake of organic N.



When "implicit" approach is used, the posterior humus decomposition coefficient $K_H$ show a higher values for the northern sites and decrease along the studied transect, demonstrating a modelled higher organic matter turnover thus soil mineralization for northern sites (Fig. 7). Less clear tendency is identified for the fraction of C allocated to roots, $F_{ROOT}$ parameter, but with a slightly tendency towards higher values at the southern sites. Microbial C/N ratio, $CN_{MIC}$ parameter for both "implicit" and "explicit" approaches show similar posterior distributions for the three northern sites, but much lower values are obtained for the south most Ljungbyhed site (Fig. 7) reflecting a more soil N rich environment. Overall, parameters are less constrained and only minor differences between sites are found when "explicit" approach is used (Fig. 7).

### 3.2.2 Fungal specific parameters:

Posterior distributions of all fungal specific parameters, are all constrained to log-normal or normal distributions (data not shown). The mean values of the N uptake parameters ($NORG_{RATE}$, $NH4_{RATE}$, $NO3_{RATE}$) show a decreasing trend from north to south sites (Fig. 8). This again means a higher ECM fungal N uptake rate is necessary to explain the observed soil and plant data at the more N-limited northern sites. Similarly lower values for the northern and higher values for the southern regions are also found for the minimum fungal C/N ratio parameter ($CN_{FMIN}$). The optimum ratio between fungal and root C content, $FRAC_{OPT}$, tends to be higher at the northern sites and lower at the southern sites, also implying that a modelled higher ECM biomass at northern sites (Fig. 8). $MIN_{SUPL}$, the minimum supply of N from fungi to the host plant parameter does not show a clear trend. Differences of the other ECM parameters for the four regions are minor (Fig. 8).

### 3.2.3 Correlation between parameters

An overview of correlations for all posterior model parameters can be found in the appendix in Table A2, A3 and A4. Parameters show correlation with each other (defined here as a Pearson correlation coefficient r ≥ 0.3 or ≤ -0.3) are identified as the key parameter sets, shown in Fig. 9. When "implicit" approach used, a significant positive correlation is obtained between the humus decomposition rate, $K_H$ and the fraction of C allocated to rooting zone, $F_{Root}$. The organic N uptake parameter, $NUPT_{ORGFRACMAX}$ and microbial C/N ratio, $CN_{MIC}$ are negative correlated except a weak correlation for Ljungbyhed (Fig. 9). A weak correlation between $NUPT_{ORGFRACMAX}$ and $F_{ROOT}$ is found for Nässjö site only (see Table A2 Appendix). For the "explicit" approach, the correlation coefficients between $K_H$ and $F_{ROOT}$ are all decreased, and also a weaker correlation between $NUPT_{ORGFRACMAX}$ and $CN_{MIC}$ for all sites comparing to that in "implicit" approach (Fig. 9). No clear correlation between common and fungal parameters is obtained. A negative correlation occurred between microbial C/N ratio, $CN_{MIC}$ and the fungal N uptake rates ($Norg_{RATE}$, $NH4_{RATE}$, $NO3_{RATE}$), but only for the Northern sites Lycksele and Mora (Table A4). A moderate correlation is found for $K_H$ and the fungal litter rate, $L$ for Ljungbyhed. Among fungal parameters, the N uptake rates correlated moderately to the litter production rate, $L$ at the northern sites, but correlations at Nässjö and Ljungbyhed are either non-existent or weak (Table A4). Our identified inter-connections and correlations between the parameters in general reflect the complex and interrelated nature of ECM, soil and plant interactions. But more importantly, they also highlight the need to calibrate a number of parameters simultaneously rather than calibrating just single parameter when applying such detailed ecosystem models (He et al., 2016, Klemedtsson et al., 2008).





## 4. Discussion

Our new version of CoupModel provides a detailed model predictive framework to explicitly account for ECM in
the plant-soil-ECM continuum. Model comparison to two earlier ECM modelling approaches show that ECM
have to be included in ecosystem models ("implicitly or explicitly") to be able to describe the long term plant, and
soil C and N development. Overall, the model perform similarly in "implicitly or explicitly" approaches while the
"nonlim" approach significantly overestimates soil N uptake. Our results thus confirm that ECM have a substantial
effect on soil C and N storage, and can also impact on forest plant growth. But more importantly, including them
into ecosystem models is both important and feasible.

### 4.1 ECM alter plant-soil C and N dynamics

"Nonlim" model in this study show overestimations of plant growth and also a clear larger biases in soil N than
"implicit and explicit" approaches even after calibration (Tab. 3). A previous CoupModel application by Wu et
al. (2012) demonstrated that the "nonlim" approach can possibly describe short term carbon and water dynamics
in a Finnish forest site. Same approach with open N cycle was also used in Franklin et al. (2014) to simulate the
Swedish forest biomass growth and its competition with ECM. It therefore seems that with "nonlim" approach,
plant growth thus C cycle can be simulated reasonably, although with slightly trend of overestimation as shown
here. However, our modelling further indicates this simplified approach has an uncoupled soil C and N in its model
structure (Fig. 3) thus not recommended for future long term soil C and N predictions. This is also reflected in the
posterior model parameter distributions where the *constantNSupply* rate parameter show primary control on the
modelled plant growth and soil conditions. Other parameters have minor or no importance for the model results,
reflecting an oversimplified soil C and N model structure. Thus in the following discussion we focus on the other
two modelling approaches.

Moore et al. (2015) demonstrated that ECM have a substantial effect on soil C storage, and its impact is largely
depended on plant growth. Present study additionally show that ECM representation in ecosystem models could
also feedback on the predicted plant growth through the feedback of N. As when ECM implicitly included, the
model simulates a 78 (average of four sites, std: 102) g C m$^{-2}$ lower plant biomass comparing to "nonlim" approach
and when explicitly included the different become even larger, 214 (50) g C m$^{-2}$ (Tab. 3). Including ECM in the
model thus show a decreased plant growth. This somehow differs with what is generally assumed that growth
should be higher in mycorrhized plants, i.e. boreal forests, due to optimized nutrient supply (Pritsch et al., 2004;
Finlay et al., 2008, see also review by Smith and Read, 2008). This discrepancy could be possibly due to; 1) an
enhanced root litterfall, due to a higher turnover of fungal mycelia, shown by higher litter turnover rate (calibrated
litter rate of ECM is 0.0075 d$^{-1}$, Fig. 8, whereas litter rate of roots is 0.0027 d$^{-1}$, Table A1(d)). When ECM is
explicitly considered, litter production is modelled to be higher than the "implicit" approach (difference from 50
to 110 g C m$^{-2}$ yr$^{-1}$, data not shown). These two modelling approach thus show large difference in simulating litter
production. Field data are further needed to clarify this out; 2) an enhanced N immobilization in ECM under N-
limited conditions, due to the ECM retains more N in its own biomass in response to plant allocation of newly
assimilated C (Nehls et al., 2008). The constrained optimum fungi C allocation fraction parameter show an
increasing trend towards more northern sites (Fig. 8), indicating a higher proportional C "investment" by the forest
plants on ECM in more north N limited conditions. The resulting ECM-plant competition for N could then
potentially result in a decreased plant N uptake thus plant growth (Näsholm et al., 2013); and 3) biases in





simulating ECM N uptake due to model/parameter uncertainties caused by high variability among fungal species and the scarcity of direct measurements in the field (Smith and Read, 2008, Clemmensen et al., 2013). Current "explicit" approach implemented many biotic interactions and internal feedbacks within the plant-soil-ECM

continuum. However, increasing the number of processes and interactions in an already complex ecosystem model will not necessarily generate more reliable model predictions, as shown here the parameters in "explicit" approach have a large uncertainty range even after calibration. Thus future model evaluation together with more detailed ECM data is of need to better understand the tightly coupled soil-ECM-plant continuum.

Both approaches simulate the soil C and N stock well (Tab. 3). The respective net change in the soil C pools of

the "implicit" approach corresponds well to the results by Svensson et al. (2008a), also suggesting a small loss of soil C in the north while a gain in the south. However, when "explicit" approach used, the soils in south are also predicted to loss C and N, mostly due to an enhanced soil respiration (see section 3.1.2). It is difficult to evaluate which approach gives a more realistic prediction, since field data not available. However, Lindroth et al. (2008) measured C fluxes at three sites in Sweden, which are situated at comparable latitudes and on comparable soils.

They also found a similar trend in the soil net C change as simulated by "explicit" approach here but with a higher loss rate between 24 to 133 g C m$^{-2}$ year$^{-1}$ (Tab. 4).

### 4.2 Parameter and model responses to different environmental conditions

Our modelling results show a consistent pattern with observations (e.g. Hyvönen et al., 2008, Näsholm et al., 2013) that at the northern N limited sites, organic N uptake by ECM is highly important for plant growth, and it

becomes less important as N availability increases southwards. As indicated by the "explicit" approach the mycorrhization degree of tree roots at Lycksele and Mora (> 90%) is much higher than that of Ljungbyhed (15%) thus majority of modelled N uptake is through fungal mycelia in northern sites. The constrained fungal organic and mineral N uptake parameters also show a decreasing trend (Fig. 8). Similarly, the organic N uptake parameter, $NUPT_{ORGFRACMAX}$ in the "implicit" approach decrease from north to south, but with a more clear site to site

difference, thus a stronger response to environmental conditions (Fig. 6). This is expected as more detailed ECM processes in "explicit" approach should result in more internal interactions and feedbacks thus damping the direct environmental regulations. Current modelling also indicates a higher mineralisation, shown by humus decomposition coefficient, $K_H$ in the northern sites. However, the decomposition also mineralization is enhanced when ECM is "explicit" considered (Fig. 7). This collaborates well with findings from the field measurements and

recently modelling studies, that ECM is able to degrade complex N polymers in humus layers thus enhance soil N transformation under low N conditions (Moore et al., 2015, Lindahl and Tunlid, 2015, Baskaran et al., 2016).

In "implicit" approach, the humus decomposition coefficient, $K_H$ was found to correlate with the fraction of C that allocates to rooting zone, $F_{ROOT}$. Since ECM is implicitly included in the roots, this correlation thus indirectly indicates a strong connection of the root-ECM symbiosis and soil N availability. But when ECM are explicitly

considered, this becomes less important, again due to a more detailed internal cycling of N supply and uptake from the fungi, i.e. plant N supply is further regulated by simulated higher litter input and N uptake from the soil in the "explicit" model (Fig. 3, Fig. 9). Our modelling show the fungal litter rates correlate to fungal N uptake rates in "explicit" model and the fungal N uptake rates have significant correlations to the microbial C/N ratio, $CN_{MIC}$ for the northern sites (Fig. 9). This indicates the close coupling between fungal N uptake (N loss from the





soil) and fungal litter production (N input to the soil). Such incorporated tight cycle is of major importance for the overall plant N supply thus C and N dynamics of plant and soil at the N limited sites in the boreal forests. Most fungal parameters in "explicit" approach are not or only weakly dependent on the differing environmental conditions along the modelled transect, except the N uptake parameters and fungal minimum C/N ratio, $CN_{FMIN}$ show different mean values (Fig. 8). Thus these parameters need to be calibrated carefully when further applied

the model to other sites with different soil nutrient level or climate conditions. One major difficulties of explicitly inclusion of ECM in ecosystem models is the unknown turnover of fungal mycelia (Ekblad et al., 2013). Previously reported turnover rates of newly formed mycelia vary from days to weeks, even up to 10 years (Staddon et al., 2003, Wallander et al., 2004), mostly due to the high variability in ECM species and structures (see review by Ekblad et al., 2013). Besides, root turnover rates can also vary considerably between species, soils, and climate

zones (Brunner et al., 2012). So far very few studies have reported parameterization of C and N cycling for ECM in boreal forests. Our calibration study thus provide a key set of ECM parameters that can be further tested through field observation, and more importantly together with the identified correlations with the variables can act as a guidelines for future ECM modelling studies.

## 5. Conclusions

The key components and features of Coup-Mycofon model have been described. The new version of CoupModel simulates C and N fluxes and pools, with the capacity of explicitly accounting for links and feedbacks between the ECM, soil and the plant. The comparison of three commonly ECM model approaches differing in complexity demonstrates that the simple "nonlim" approach cannot describe the measured soil C/N ratio, also overestimates measured forest growth. When including ECM either implicitly or explicitly, both deliver accurate long-term

quantitative predictions on forest C and N cycling with simultaneous consideration of the impact of ECM fungi on ecosystem dynamics, but slightly underestimate forest growth. The ECM explicit Coup-Mycofon model provides a more detailed description of internal ecosystems fluxes and feedbacks of C and N fluxes. The constrained ECM parameter distributions presented in this study can be used as a guideline for future model applications. Our model implementation and comparison overall suggest ecosystem models need to incorporate

ECM fungi into their model structure for a better prediction of ecosystem C and N dynamics, and the new version of CoupModel now provides such an alternative.

## 6. Code and data availability

The model and extensive documentation with tutorial excises are freely available from the CoupModel home page http://www.coupmodel.com/ (CoupModel, 2015). The source code can be requested for non-commercial purposes

from Per-Erik Jansson (pej@kth.se). CoupModel is written in the C programming language (code also available in Fortran) and run mainly under Windows/Linux systems. Inputs and outputs are in binary format. The version used as the basis for the present development was version 5 from 12 April 2017. The simulation files including the model and calibration set-up, the used parameterization and corresponding input and validation files can be requested from Hongxing He (hongxing.he@gu.se). However, majority of the input and output data used for

current modelling is public available, i.e. through SMHI or previous publication, i.e. Svensson et al. (2008). Please



contact the first author of this publication or Per-Erik if you plan an application of the model and further collaboration.

*Acknowledgements*: Financial support came from the Swedish Research Council for Environment, Agricultural Sciences and Spatial Planning (FORMAS), the strategic research area BECC (Biodiversity and Ecosystem services in a Changing Climate, www.cec.lu.se/research/becc), the Linnaeus Centre LUCCI (Lund University Centre for studies of Carbon Cycle and Climate Interactions).


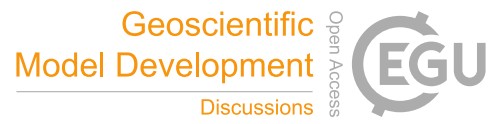

APPENDIX:

**Table A.1 Model functions describing plant growth, fungal growth, model parameters, and response functions of plant and ECM. Parameters are always entitled with capital letters**

**Table A.1 (a) Description of plant model functions. (i = fine roots, coarse roots, stem, leaves, grain, mobile)**

| No. | Equation |
| --- | --- |

Plant photosynthesis (g C m$^{-2}$ d$^{-1}$):

1 $$c_{atm \to plant} = \varepsilon_L \times f(T_1) \times f(CN_1) \times f\left(\frac{E_{ta}}{E_{tp}}\right) \times r_S$$

$\varepsilon_L$= coefficient for radiation use efficiency, $f(T_l)$, $f(CN_l)$, $f(E_{ta}/E_{tp})$ = response functions to leaf temperature, leaf CN and air moisture (see Table A.1 (c)), $r_s$ = global radiation absorbed by canopy.

Plant maintenance respiration (g C m$^{-2}$ d$^{-1}$):

2.1 $$c_{plantM \to atm} = c_i \times K_{RMi} \times f(T_l)$$

$c_i$ = C content of each respective plant compartment i (g C m$^{-2}$) and $K_{RMi}$ is a coefficient.

Plant growth respiration (g C m$^{-2}$ d$^{-1}$):

2.2 $$c_{plantG \to atm} = c_{m \to i} \times K_{RGi}$$

$c_{m \to i}$ = C gain (growth) of each plant compartment i (g C m$^{-2}$ d$^{-1}$) and $K_{RGi}$ is a coefficient.

Plant litter production (g C m$^{-2}$ d$^{-1}$):

3 $$c_{i \to lit} = c_i \times L_i$$

where $C_i$ is the C content of each plant compartment i (g C m$^{-2}$) and $L_i$ (= 0.0027 d$^{-1}$) is a coefficient.

Plant nitrate and ammonium uptake (g N m$^{-2}$ d$^{-1}$) (only shown for nitrate, equivalent for ammonium):

4.1 $$n_{NO3 \to plant} = dem_{Nplant} \times r_{NO3}$$  if $f(n_{minavail})$ x $n_{NO3soil} \geq dem_{Nplant}$ x $r_{NO3}$

4.2 $$n_{NO3 \to plant} = f(n_{minavail}) \times n_{NO3soil} \times dem_{Nplant}$$  if $f(n_{minavail})$ x $n_{NO3soil} \leq dem_{Nplant}$ x $r_{NO3}$

and where

4.3 $$dem_{Nplant} = \sum \frac{c_{a \to i} - c_{i \to atm}}{CN_{iMIN}}$$

$f(n_{NO3avail})$ = fraction of soil NO$_3$ available for plant uptake (see response functions Table A.1 (d) ), $n_{NO3soil}$ = soil NO$_3$-N content (g N m$^{-2}$), $dem_{Nplant}$ = plant N demand (g N m$^{-2}$ d$^{-1}$), $r_{NO3}$ = fraction of soil NO$_3$-N in total mineral soil N, $c_{a \to i}$ = plant C gain ( g C m$^{-2}$ d$^{-1}$), $c_{i \to atm}$ = respiration of respective plant compartment i (g C m$^{-2}$ d$^{-1}$), $CN_{iMIN}$ = defined minimum C:N ratio of each plant compartment i.

Plant organic N uptake (g N m$^{-2}$ d$^{-1}$) from the humus layer:

4.4 $$n_{hum \to plant} = dem_{Nplant} \times r_{hum}$$  if $f(n_{humavail})$ x n$_{humsoil} \geq dem_{Nplant}$ x $r_{hum}$

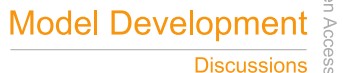

4.5     $n_{hum \rightarrow plant} = f(n_{humavail}) \times n_{humsoil}$      if $f(n_{humavail})$ x $n_{humsoil} < dem_{Nplant}$ x $r_{hum}$

$f(n_{humavail})$ = response function for plant available N from the humus layer, $n_{humsoil}$ = soil N content in humus layer (g N m$^{-2}$)


**Table A.1 (b) Functions describing processes related to fungal growth and N exchange to plant**

Fungal maximum C supply (g C m$^{-2}$ d$^{-1}$):

5.1     $c_{a \rightarrow fungi} = c_{a \rightarrow root} \times FRAC_{FMAX} \times f(c_{fungiavail})$

Fungal actual growth (g C m$^{-2}$ d$^{-1}$):

5.2     $c_{a \rightarrow fungi} = ((c_{frt} \times FRAC_{OPT}) - c_{fungi}) \times f(n_{supply})$

$c_{a \rightarrow root}$ = C available for root and mycorrhiza growth (g C m$^{-2}$ d$^{-1}$), $FRAC_{FMAX}$ = maximum fraction of total root and mycorrhiza available C which is available for ECM, $f(c_{fungiavail})$ = response function which relates fungal growth to N availability, $c_{frt}$ = total root C content (g C m$^{-2}$), $FRAC_{OPT}$ = optimum ratio between root and fungal C content, $c_{fungi}$ = total ECM C content (g C m$^{-2}$), $f(n_{supply})$ = response function of fungal

growth to the amount of N which is transferred from ECM to plant.

Minimum fungal C supply (g C m$^{-2}$ d$^{-1}$):

5.3     $c_{a \rightarrow fungi} = c_{fungi \rightarrow atm}$      if $c_{a \rightarrow root} \leq 0$

Total fungal respiration (g C m$^{-2}$ d$^{-1}$):

6.1     $c_{fungi \rightarrow atm} = c_{mfungi \rightarrow a} + c_{gfungi \rightarrow a}$

where $c_{mfungi \rightarrow a}$ = fungal maintenance respiration and $c_{gfungi \rightarrow a}$ = fungal growth respiration (all in g C m$^{-2}$ d$^{-1}$).

Fungal maintenance respiration (g C m$^{-2}$ d$^{-1}$):

6.2     $c_{mfungi \rightarrow a} = c_{fungi} \times K_{RM} \times f(T_l)$

$c_{fungi}$ = total ECM C content (g C m$^{-2}$), $K_{RM}$ = maintenance respiration coefficient, $f(T_l)$ = temperature response function

Fungal growth respiration (g C m$^{-2}$ d$^{-1}$):

6.3     $c_{gfungi \rightarrow a} = c_{a \rightarrow fungi} \times K_{RG}$

$c_{a \rightarrow fungi}$ = fungal growth (g C m$^{-2}$ d$^{-1}$), $K_{RG}$ = growth respiration coefficient.


Fungal C and N litter production ($c_{fungi \rightarrow lit}$: g C m$^{-2}$ d$^{-1}$, $n_{fungi \rightarrow lit}$: g N m$^{-2}$ d$^{-1}$):

If fungal growth = simple

7.1     $c_{fungi \rightarrow lit} = c_{fungi} \times L$

7.2     $n_{fungi \rightarrow lit} = n_{fungi} \times L - nret_{fungi}$

7.3     $nret_{fungi} = n_{fungi} \times L \times (1 - N_{RET})$





$c_{fungi}$ = ECM C content (g C m$^{-2}$), $n_{fungi}$ = fungal N content (g N m$^{-2}$), $L$ = litter rate, $nret_{fungi}$: fungal N which is retained in fungal tissue, $N_{RET}$ = fraction of N retained in fungal tissue from senescence

If fungal growth = detailed

7.4 $\quad c_{fungi \rightarrow lit} = c_{fungi} \times (FRAC_{MYC} \times L_{MYC} + ((1 - FRAC_{MYC}) \times L_M))$

7.5 $\quad n_{fungi \rightarrow lit} = n_{fungi} \times (FRAC_{MYC} \times L_{MYC} + ((1 - FRAC_{MYC}) \times L_M)) - nret_{fungi}$

7.6 $\quad FRAC_{MYC}$ = fraction of mycorrhizal hyphae in total fungal biomass, $L_{MYC}$ = litter rate of mycorrhizal hyphae, $L_M$= litter rate of fungal mantle tissue.

Fungal biomass (g C m$^{-2}$, g N m$^{-2}$)

8.1 $\quad c_{fungi} = c_{a \rightarrow fungi} - c_{fungi \rightarrow litter} - c_{fungi \rightarrow a}$

8.2 $\quad n_{fungi} = n_{N \rightarrow fungi} - n_{fungi \rightarrow litter} - n_{fungi \rightarrow plant}$

Mycorrhization degree

9 $\quad m = \dfrac{c_{fungi}}{c_{frt} \times FRAC_{OPT} \times M_{OPT}}$

$\quad c_{frt}$ = fine root biomass (g C m$^{-2}$), $FRAC_{OPT}$ = coefficient defining optimum ratio between fungal and fine root biomass, $M_{OPT}$ = optimum mycorrhization degree.

Uptake and transfer processes of ECM and plant

N transfer from ECM to plant (g N m$^{-2}$ d$^{-1}$)

10.1 $\quad n_{fungi \rightarrow plant} = dem_{Nplant}$ $\qquad\qquad$ if dem$_{Nplant}$ ≤ n$_{fungiavail}$

$\quad n_{fungi \rightarrow plant} = n_{fungiavail}$ $\qquad\qquad$ if dem$_{Nplant}$ > n$_{fungiavail}$

$dem_{Nplant}$ = plant N demand, $n_{fungiavail}$ = fungal available N for transfer to plant (all g N m$^{-2}$ d$^{-1}$ )

10.2 $\quad n_{fungiavail} = n_{fungi} - \dfrac{c_{fungi}}{CN_{FMAX}}$

$c_{fungi}$ = ECM biomass (g C m$^{-2}$), $CN_{FMAX}$ = maximum C:N ratio of fungal tissue, which allows N transfer
to plant

Fungal nitrate and ammonium uptake (given for nitrate, equivalent for ammonium with ammonium specific parameter)

11.1 $\quad n_{NO3 \rightarrow fungi} = n_{NO3pot \rightarrow fungi} \times r_{NO3} \times f(n_{demfungi})$ $\qquad$ if $N_{NO3pot \rightarrow fungi} < n_{NO3soil} \, x f(n_{avfungi})$

11.2 $\quad n_{NO3 \rightarrow fungi} = n_{NO3soil} \times f(n_{avfungi})$ $\qquad$ if $N_{NO3pot \rightarrow fungi} > n_{NO3soil} \, x f(n_{avfungi})$

11.3 $\quad n_{NO3pot \rightarrow fungi} = NO3_{RATE} \times c_{fungi} \times FRAC_{MYC}$

$n_{NO3pot \rightarrow fungi}$=potential ECM nitrate uptake (g N m$^{-2}$ d$^{-1}$), $r_N$ = fraction of ammonium-N and total mineral-N in the soil, $f(n_{demfungi})$= N uptake response to N demand, n$_{NO3soil}$ = soil nitrate content (g N m$^{-2}$), $f(n_{avfungi})$

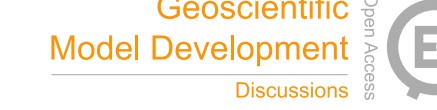

= N uptake response to soil availability, NO3$_{RATE}$ = nitrate specific uptake rate (g N m$^{-2}$ d$^{-1}$), $c_{fungi}$ = fungal biomass (g C m$^{-2}$), FRAC$_{MYC}$ = fraction of mycorrhizal mycelia in total fungal biomass

Fungal organic N uptake from litter and humus (given for litter, equivalent for humus with humus specific parameter)

11.4 $\quad n_{lit \to fungi} = n_{litpot \to fungi} \times r_{lit} \times f(n_{demfungi})$ $\qquad$ if n$_{litpot \to fungi}$ x $r_{lit}$ < $n_{litsoil}$ x$f(n_{litavfungi})$x$r_{lit}$

11.5 $\quad n_{lit \to fungi} = n_{litsoil} \times f(n_{litavfungi}) \times r_{lit}$ $\qquad$ if n$_{litpot \to fungi}$ x $r_{lit}$ > $n_{litsoil}$ x$f(n_{litavfungi})$ x$r_{lit}$

11.6 $\quad n_{litpot \to fungi} = LIT_{RATE} \times c_{fungi} \times FRAC_{MYC}$

where n$_{litpot \to fungi}$ = potential ECM organic N uptake from litter (g N m$^{-2}$ d$^{-1}$), $r_{lit}$ = fraction of litter-N in total organic-N in the soil, $f(n_{demfungi})$= N uptake response to N demand, , n$_{litsoil}$ = soil litter content (g N m$^{-2}$), NLIT$_{RATE}$ = litter specific uptake rate (g N g C$^{-1}$ d$^{-1}$), $c_{fungi}$ = fungal biomass (g C m$^{-2}$), FRAC$_{MYC}$ = fraction of mycorrhizal mycelia in total fungal biomass


**Table A1 (c) Overview of response functions of plant and fungal growth and N uptake**

| No. | Equation | | |
|---|---|---|---|
| Plant response to air temperature | | | |
| | | 0 | T$_1$ < P$_{mn}$ |
| | | (T$_1$ - p$_{mn}$) / (p$_{O1}$ - p$_{mn}$) | p$_{mn}$ ≤ T$_1$ ≤ p$_{O1}$ |
| 12 | $f(T_l) =$ | 1 | p$_{O1}$ < T$_1$ < p$_{O2}$ |
| | | 1- (T$_1$ - p$_{O2}$) / (p$_{mx}$ - P$_{O2}$) | p$_{O2}$ < T$_1$ < p$_{mx}$ |
| | | 0 | T$_1$ > p$_{mx}$ |

where T$_1$ = leaf temperature (°C) and P$_{MN}$ (-4°C), PO1 (10°C), PO$_2$ (25°C), P$_{MX}$ (40°C) are coefficients.


| | Photosynthetic response to leaf C/N ratio | | |
|---|---|---|---|
| | | 1 | CN$_1$ < p$_{CNOPT}$ |
| 13 | $f(CN_l) = 1 + (\dfrac{cn_l - p_{CNOPT}}{p_{COPT} - p_{CNTH}})$ | | p$_{CNTH}$ ≤ CN$_1$ ≥ p$_{CNOPT}$ |
| | | 0 | CNl > p$_{CNTH}$ |

where CN$_1$ = leaf C:N ratio and p$_{CNOPT}$ (25) and p$_{CNTH}$ (75) are parameters.

Plant response to soil moisture

14 $\quad f(\dfrac{E_{ta}}{E_{tp}}) = \dfrac{E_{ta}}{E_{tp}}$

where E$_{ta}$ = actual transpiration and E$_{tp}$ = potential transpiration (mm d$^{-1}$)

620

Plant mineral N uptake response to N availability and fungal mantle

15 $\quad f(n_{min\,avail}) = NUPT_{FRACMAX} \times e^{(-FM \times m)}$





Where $NUPT_{FRACMAX}$, coefficient describing fraction of soil N available, and $FM$, uptake reduction due to fungal mantle

625

Plant organic N uptake response to N availability and fungal mantle (given for litter, equivalent for humus)

16     $f(n_{humavail}) = NUPT_{ORGFRACMAX} \times e^{(-FM \times m)}$

Where $NUPT_{FRACMAX}$ is the respective uptake coefficient for N from humus (included in calibration), and $FM$ the uptake reduction due to fungal mantle

630

ECM N uptake response to N availability

17.1     $f(n_{avfungi}) = NUPT_{FRACMAX} \times UPT_{MINENHANCE}$            for nitrate

17.2     $f(n_{avfungi}) = NUPT_{FRACMAX} \times UPT_{MINERAL} \times UPT_{NH4}$       for ammonium

17.3     $f(n_{orgavfungi}) = NUPT_{ORGFRACMACX} \times UPT_{ORG}$          for litter/humus

635

ECM N uptake response to N demand

18     $f(n_{demfungi}) = 1 - \dfrac{CN_{FMIN}}{cn_{fungi}}$

where $CN_{FMIN}$ = minimum ECM C/N ratio

19     $f(c_{fungiavail}) = e^{(-N_{AVAILCOEF} \times n_{minsoil}^2)^3}$

Where $NAVAIL_{COEF}$ is a coefficient and $N_{minsoil}$ is the total soil content of ammonium and nitrate (g N m$^{-2}$)

20.1     $f(n_{supplyfungi}) = 1$               if $\min_{NPlant} < n_{fungi \to plant}$

20.2     $f(n_{supplyfungi}) = \dfrac{n_{fungi \to plant}}{n_{fungi \to plant} + n_{soil \to plant}}$     if $\min_{NPlant} > n_{fungi \to plant}$

20.3     $\min_{NPLant} = MIN_{SUPL} \cdot (n_{fungi \to plant} + n_{soil \to plant})$

Where $\min_{NPlant}$ = defined minimum fungal N supply in plant N uptake, $n_{fungi \to plant}$ = actual ECM N supply to plant (g N m$^{-2}$ d$^{-1}$), $n_{soil \to plant}$ = total plant N uptake from mineral and organic fraction (g N m$^{-2}$ d$^{-1}$)

**Table A1 (d) Overview of model parameters, previous CoupModel parameters are mostly from Svensson et al., (2008a) and ECM parameters are from literature value (references in the paper text)**


| Parameter | Description | Value | Unit |
|---|---|---|---|
| $CN_{FMIN}$ | Minimum fungal C/N ratio for fungal N demand | 18 | gC gN$^{-1}$ |
| $CN_{FMAX}$ | Maximum fungal C/N ratio for N transfer to plant | 30 | gC gN$^{-1}$ |
| $CN_{iMIN}$ | Minimum C/N ratio of fine roots, | 40 | gC gN$^{-1}$ |
| | Needles /leaves | 22 | gC gN$^{-1}$ |
| | Coarse roots and stem | 450 | gC gN$^{-1}$ |
| $E_L$ | coefficient for radiation use efficiency | 8 | |




| | | | | |
|---|---|---|---|---|
| | $E_{NH4}$ | fungal NH$_4$ uptake enhancement factor | 5 | |
| | FM | plant N uptake reduction due to ECM mantle | 0.5 | |
| 660 | FRAC$_{FMAX}$ | Maximum fraction of C allocated to rooting zone which is made available for ECM | 0.5 | |
| | FRAC$_{MYC}$ | Fraction of fungal mycelia in total biomass | 0.5 | |
| | FRAC$_{OPT}$ | Optimum fraction between root and fungal biomass | 0.3 | |
| | K$_{RGF}$ | Growth respiration coefficient of ECM | 0.21 | d$^{-1}$ |
| 665 | K$_{RMi}$ | Maintenance respiration coefficient of plant compartment i | | |
| | | (i = fine roots, coarse roots, stem, leaves) | 0.001 | d$^{-1}$ |
| | K$_{RGi}$ | Growth respiration coefficient of plant compartment i | 0.21 | d$^{-1}$ |
| | L$_{FRT}$ | Litter rate of fine roots | 0.0027 | d$^{-1}$ |
| 670 | L$_{CRT}$ | Litter rate of coarse roots | 0.000027 | d$^{-1}$ |
| | L$_{LEAF}$ | Litter rate of needles | 0.0002 | d$^{-1}$ |
| | L$_{STEM}$ | Litter rate of stem | 0.000027 | d$^{-1}$ |
| | L | Litter rate of ECM (if fungal growth = simple) | 0.004 | |
| | L$_M$ | Litter rate of fungal mantle | | |
| 675 | | (if fungal growth = detailed) | 0.0014 | d$^{-1}$ |
| | L$_{MYC}$ | Litter rate of fungal mycelia (if fungal growth = detailed) | 0.01 | d$^{-1}$ |
| | M$_{OPT}$ | Optimum mycorrhization degree of fine roots < 2 mm | 0.5 | |
| 680 | N$_{RET}$ | N retained by ECM from senescence | 0.54 | d$^{-1}$ |
| | NUPT$_{FRACMAX}$ | fraction of mineral N available for uptake | 0.08 | d$^{-1}$ |





**Table A2 Correlation between common model parameters for all simulated sites with the "implicit" and "explicit" approaches, respectively. Correlation is given as the Pearson correlation coefficient**

| | | implicit | | | | explicit | | | |
|---|---|---|---|---|---|---|---|---|---|
| | | $K_H$ | $NUPT_{OFM}$ | $F_{ROOT}$ | $CN_{MIC}$ | $K_H$ | $NUPT_{OFM}$ | $F_{ROOT}$ | $CN_{MIC}$ |
| Lycksele | $K_H$ | 1 | -0.20 | 0.67 | 0.23 | 1 | -0.08 | 0.28 | 0.21 |
| | $NUPT_{OFM}$ | | 1 | 0.24 | -0.57 | | 1 | 0.02 | -0.35 |
| | $F_{ROOT}$ | | | 1 | 0.18 | | | 1 | 0.02 |
| | $CN_{MIC}$ | | | | 1 | | | | 1 |
| Mora | $K_H$ | 1 | -0.13 | 0.73 | 0.11 | 1 | 0.08 | 0.22 | 0.04 |
| | $NUPT_{OFM}$ | | 1 | 0.18 | -0.64 | | 1 | 0.10 | -0.46 |
| | $F_{ROOT}$ | | | 1 | 0.13 | | | 1 | 0.12 |
| | $CN_{MIC}$ | | | | 1 | | | | 1 |
| Nässjö | $K_H$ | 1 | 0.03 | 0.70 | -0.08 | 1 | 0.13 | 0.29 | 0.16 |
| | $NUPT_{OFM}$ | | 1 | 0.31 | -0.60 | | 1 | 0.29 | -0.53 |
| | $F_{ROOT}$ | | | 1 | 0.02 | | | 1 | 0.12 |
| | $CN_{MIC}$ | | | | 1 | | | | 1 |
| Ljungbyhed | $K_H$ | 1 | 0.03 | 0.66 | -0.18 | 1 | 0.33 | 0.26 | -0.19 |
| | $NUPT_{OFM}$ | | 1 | 0.17 | -0.28 | | 1 | 0.23 | -0.26 |
| | $F_{ROOT}$ | | | 1 | 0.24 | | | 1 | 0.07 |
| | $CN_{MIC}$ | | | | 1 | | | | 1 |





**Table A3 Correlation between fungal and common model parameters with "explicit" approach for all sites. Correlation is given as the Pearson correlation coefficient**


| | | $Norg_{RATE}$ | $NH4_{RATE}$ | $NO3_{RATE}$ | $K_{RM}$ | $L_{MYC}$ | $L_M$ | $CN_{FMIN}$ | $MIN_{SUPL}$ | $FRAC_{OPT}$ | $NAVAIL_{COEF}$ |
|---|---|---|---|---|---|---|---|---|---|---|---|
| Lycksele | $K_H$ | 0.17 | 0.16 | 0.16 | 0.01 | -0.30 | -0.27 | 0.02 | 0.00 | -0.17 | -0.13 |
| | $NUPT_{OFM}$ | -0.32 | -0.28 | -0.28 | 0.09 | 0.13 | 0.13 | 0.18 | 0.10 | 0.01 | 0.02 |
| | $F_{ROOT}$ | 0.06 | 0.03 | 0.03 | -0.05 | 0.03 | 0.03 | 0.00 | 0.06 | -0.04 | -0.15 |
| | $CN_{MIC}$ | -0.33 | -0.34 | -0.34 | 0.00 | 0.21 | 0.21 | 0.23 | 0.03 | -0.12 | -0.01 |
| Mora | $K_H$ | 0.22 | 0.20 | 0.20 | -0.09 | -0.25 | -0.21 | -0.02 | 0.08 | -0.14 | -0.04 |
| | $NUPT_{OFM}$ | -0.15 | -0.09 | -0.09 | 0.08 | 0.02 | 0.02 | 0.05 | 0.11 | -0.08 | 0.00 |
| | $F_{ROOT}$ | -0.11 | -0.12 | -0.12 | 0.06 | 0.26 | 0.26 | 0.25 | -0.06 | -0.01 | 0.01 |
| | $CN_{MIC}$ | -0.38 | -0.40 | -0.40 | -0.03 | 0.29 | 0.29 | 0.33 | -0.10 | -0.08 | -0.08 |
| Nässjö | $K_H$ | 0.20 | 0.18 | 0.18 | -0.06 | -0.33 | -0.32 | -0.13 | 0.08 | -0.03 | -0.08 |
| | $NUPT_{OFM}$ | -0.07 | -0.03 | -0.03 | -0.05 | -0.11 | -0.11 | -0.12 | -0.03 | 0.18 | -0.06 |
| | $F_{ROOT}$ | -0.06 | -0.03 | -0.03 | 0.01 | 0.08 | 0.08 | 0.09 | -0.08 | 0.14 | -0.08 |
| | $CN_{MIC}$ | -0.23 | -0.20 | -0.20 | 0.05 | 0.11 | 0.11 | 0.15 | -0.02 | -0.17 | 0.09 |
| Ljungbyhed | $K_H$ | 0.34 | 0.36 | 0.36 | -0.08 | -0.51 | -0.53 | -0.13 | 0.18 | -0.22 | -0.20 |
| | $NUPT_{OFM}$ | 0.10 | 0.16 | 0.16 | 0.05 | -0.21 | -0.21 | -0.24 | 0.06 | -0.13 | -0.07 |
| | $F_{ROOT}$ | -0.11 | -0.07 | -0.07 | 0.19 | 0.10 | 0.10 | 0.11 | 0.04 | 0.02 | -0.02 |
| | $CN_{MIC}$ | -0.22 | -0.21 | -0.21 | 0.01 | 0.15 | 0.15 | 0.18 | -0.05 | 0.02 | 0.07 |





**Table A4 Correlation between fungal model parameters with the "explicit" approach for all sites. Correlation is given as the Pearson correlation coefficient**

| | | $Norg_{RATE}$ | $NH4_{RATE}$ | $NO3_{RATE}$ | $K_{RM}$ | $L_{MYC}$ | $L_M$ | $CN_{FMIN}$ | $MIN_{SUPL}$ | $FRAC_{OPT}$ | $NAVAIL_{COEF}$ |
|---|---|---|---|---|---|---|---|---|---|---|---|
| Lycksele | $Norg_{RATE}$ | 1 | 0.91 | 0.91 | 0.01 | -0.55 | -0.59 | -0.10 | -0.07 | 0.07 | -0.03 |
| | $NH4_{RATE}$ | | 1 | 0.99 | 0.01 | -0.50 | -0.56 | -0.07 | -0.05 | 0.07 | -0.03 |
| | $NO3_{RATE}$ | | | 1 | 0.01 | -0.50 | -0.56 | -0.07 | -0.05 | 0.07 | -0.03 |
| | $K_{RM}$ | | | | 1 | -0.1 | -0.1 | -0.06 | -0.07 | -0.03 | -0.04 |
| | $L_{MYC}$ | | | | | 1 | 0.95 | 0.04 | 0.07 | -0.17 | -0.03 |
| | $L_M$ | | | | | | 1 | 0.04 | 0.07 | -0.13 | -0.02 |
| | $CN_{FMIN}$ | | | | | | | 1 | 0.05 | 0.07 | 0.05 |
| | $MIN_{SUPL}$ | | | | | | | | 1 | 0 | 0.05 |
| | $FRAC_{OPT}$ | | | | | | | | | 1 | 0.17 |
| | $NAVAIL_{COEF}$ | | | | | | | | | | 1 |
| Mora | $Norg_{RATE}$ | 1 | 0.88 | 0.88 | -0.09 | -0.40 | -0.48 | 0.02 | -0.05 | 0.04 | 0.06 |
| | $NH4_{RATE}$ | | 1 | 0.99 | -0.08 | -0.32 | -0.43 | 0.01 | -0.03 | 0.09 | 0.08 |
| | $NO3_{RATE}$ | | | 1 | -0.08 | -0.32 | -0.43 | 0.01 | -0.03 | 0.09 | 0.08 |
| | $K_{RM}$ | | | | 1 | -0.07 | -0.06 | 0.01 | -0.15 | 0.05 | 0.05 |
| | $L_{MYC}$ | | | | | 1 | 0.95 | -0.08 | 0.05 | -0.21 | -0.02 |
| | $L_M$ | | | | | | 1 | -0.07 | 0.07 | -0.19 | -0.03 |
| | $CN_{FMIN}$ | | | | | | | 1 | -0.08 | -0.01 | 0.04 |
| | $MIN_{SUPL}$ | | | | | | | | 1 | 0.06 | 0.13 |
| | $FRAC_{OPT}$ | | | | | | | | | 1 | 0.02 |
| | $NAVAIL_{COEF}$ | | | | | | | | | | 1 |





|  |  | Norg$_{RATE}$ | NH4$_{RATE}$ | NO3$_{RATE}$ | K$_{RM}$ | L$_{MYC}$ | L$_M$ | CN$_{FMIN}$ | MIN$_{SUPL}$ | FRAC$_{OPT}$ | NAVAIL$_{COEF}$ |
|---|---|---|---|---|---|---|---|---|---|---|---|
| Nässjö | Norg$_{RATE}$ | 1 | 0.86 | 0.86 | 0.05 | -0.13 | -0.20 | -0.08 | -0.10 | 0.09 | -0.02 |
|  | NH4$_{RATE}$ |  | 1 | 0.99 | 0.11 | 0.00 | -0.07 | -0.09 | -0.07 | 0.15 | -0.02 |
|  | NO3$_{RATE}$ |  |  | 1 | 0.11 | 0.00 | -0.07 | -0.09 | -0.07 | 0.15 | -0.02 |
|  | K$_{RM}$ |  |  |  | 1 | -0.05 | -0.06 | 0.01 | 0.05 | -0.01 | 0.01 |
|  | L$_{MYC}$ |  |  |  |  | 1 | 0.96 | 0.07 | 0.06 | -0.11 | -0.02 |
|  | L$_M$ |  |  |  |  |  | 1 | 0.06 | 0.07 | -0.11 | -0.05 |
|  | CN$_{FMIN}$ |  |  |  |  |  |  | 1 | -0.07 | -0.07 | 0.08 |
|  | MIN$_{SUPL}$ |  |  |  |  |  |  |  | 1 | -0.05 | -0.04 |
|  | FRAC$_{OPT}$ |  |  |  |  |  |  |  |  | 1 | 0.02 |
|  | NAVAIL$_{COEF}$ |  |  |  |  |  |  |  |  |  | 1 |
| Ljungbyhed | Norg$_{RATE}$ | 1 | 0.86 | 0.86 | -0.13 | -0.32 | -0.40 | -0.06 | 0.07 | 0.04 | -0.03 |
|  | NH4$_{RATE}$ |  | 1 | 0.99 | -0.07 | -0.21 | -0.28 | -0.05 | 0.02 | 0.06 | 0.00 |
|  | NO3$_{RATE}$ |  |  | 1 | -0.07 | -0.21 | -0.28 | -0.05 | 0.02 | 0.06 | 0.00 |
|  | K$_{RM}$ |  |  |  | 1 | -0.09 | -0.08 | -0.01 | 0.01 | -0.03 | -0.05 |
|  | L$_{MYC}$ |  |  |  |  | 1 | 0.96 | 0.01 | -0.08 | -0.04 | 0.12 |
|  | L$_M$ |  |  |  |  |  | 1 | 0.02 | -0.10 | -0.04 | 0.10 |
|  | CN$_{FMIN}$ |  |  |  |  |  |  | 1 | -0.03 | 0.16 | 0.04 |
|  | MIN$_{SUPL}$ |  |  |  |  |  |  |  | 1 | -0.07 | -0.03 |
|  | FRAC$_{OPT}$ |  |  |  |  |  |  |  |  | 1 | 0.01 |
|  | NAVAIL$_{COEF}$ |  |  |  |  |  |  |  |  |  | 1 |


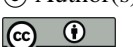



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





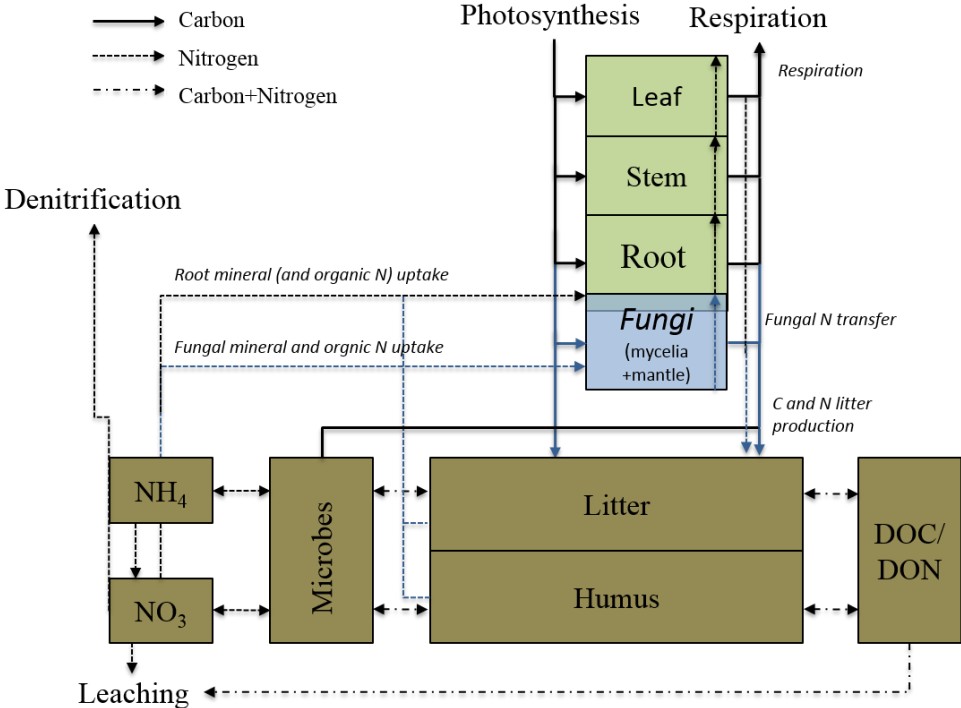

**Figure 1 A simplified overview of C and N fluxes between plants, mycorrhiza fungi and the soil in the Coup-MYCOFON model. Light blue indicates the newly implemented MYCOFON model**




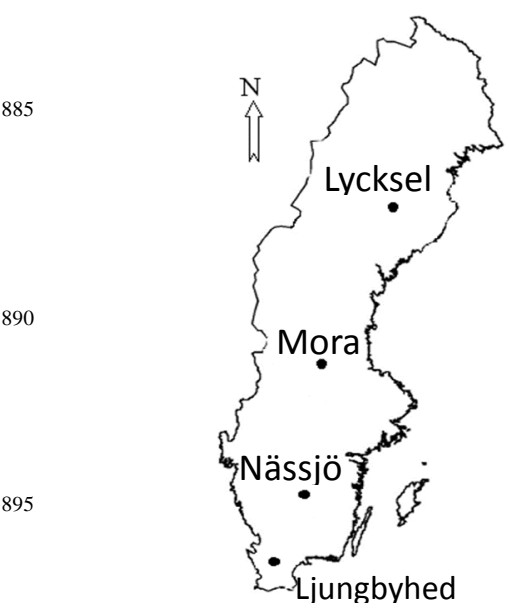



**Figure 2 Position of the four study sites in Sweden**



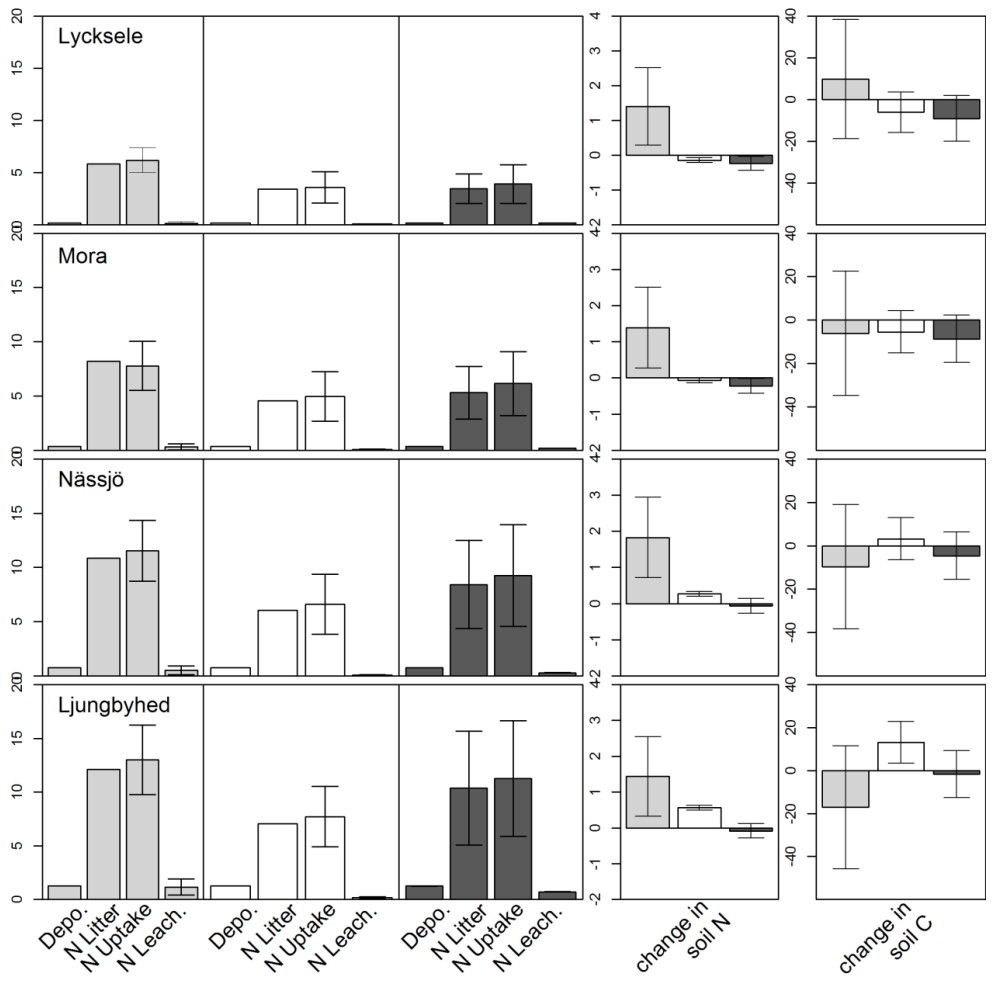

Figure 3 Soil N fluxes for the nonlim (grey columns, left), implicit (white, 2nd left column) and explicit (black, 3rd left column) model approaches. Presented are the major N inputs (N deposition, total N litter production) and outputs (N uptake from the plant/fungi, N leaching) and the net change in the total soil N pool (mineral and organic). For C the net change is presented (right column). Error bars indicate the 90th percentile of accepted model runs (posterior). Units for N are g N m$^{-2}$ yr$^{-1}$ and g C m$^{-2}$ yr$^{-1}$ for C




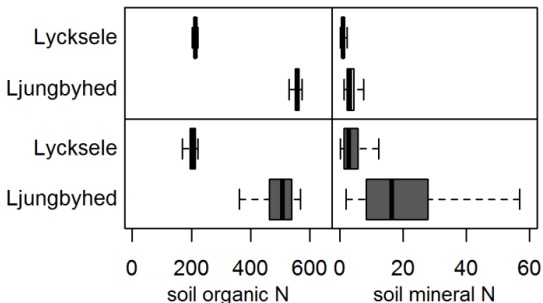


**Figure 4 Average soil organic and mineral (ammonium and nitrate) content (g N m⁻²) in implicit ECM model (upper graph) and explicit ECM model (lower graph) for the two sites Lycksele and Ljungbyhed. Box plots indicate the median (bold line), the 25th and 75th percentile (bars) and the 10th and 90th percentile (whiskers)**


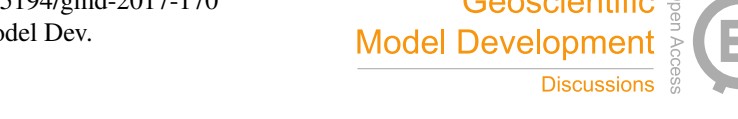



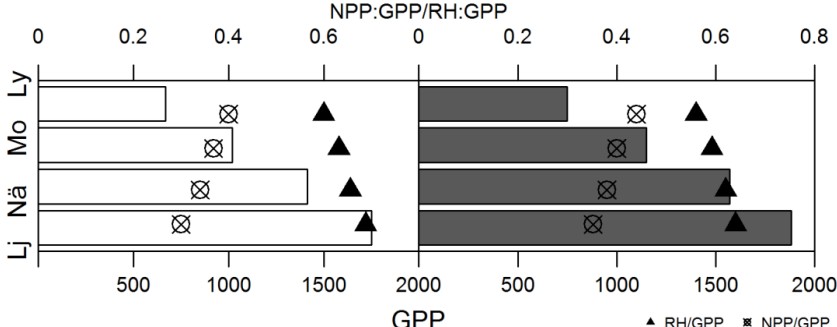

**Figure 5 GPP (bars), Rh/GPP ratio (triangle) and NPP/GPP ratio (cross circles) for all four sites simulated with the implicit (left) and explicit (right) ECM model approach**




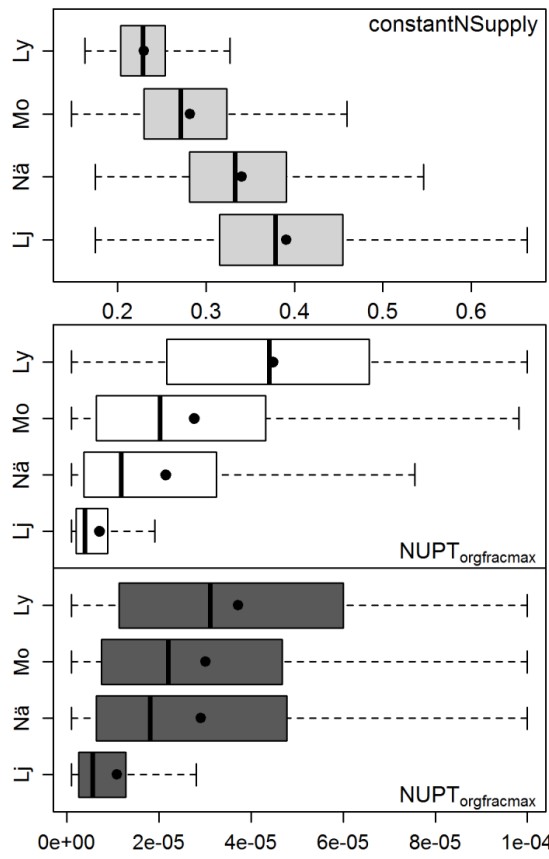

**Figure 6 Posterior parameter distributions for N uptake parameters: constant N supply rate in the "nonlim" approach (light grey), and organic N uptake capacity in the implicit (white) and explicit (dark grey) ECM model approaches. Distributions are presented as box plots over the prior range of variation (corresponding to the range in the x-axis). Box plots depict the median (bold line), the 25th and 75th percentile (bars) and the 10th and 90th percentile (whiskers)**





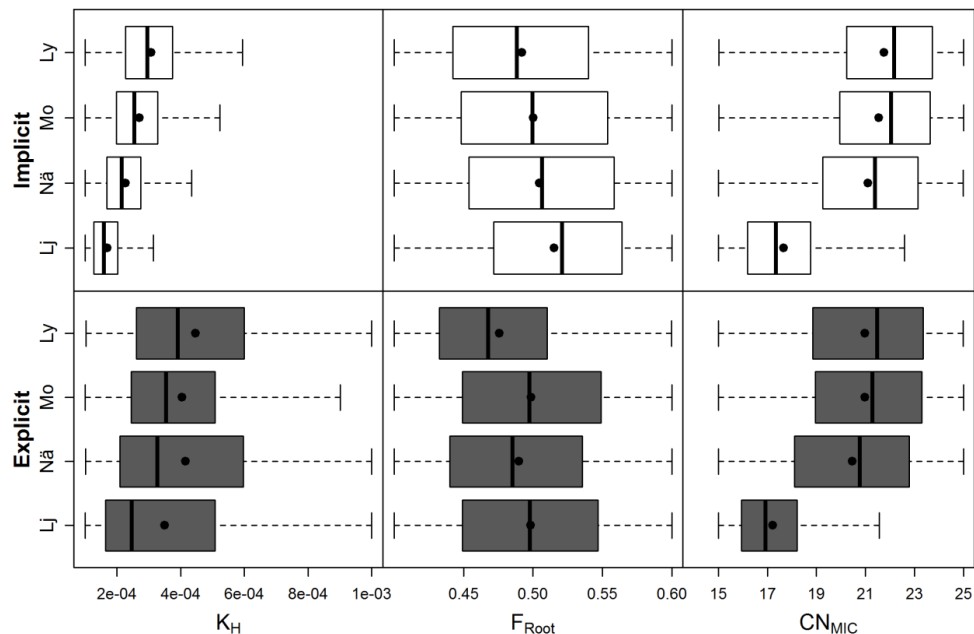

**Figure 7 Posterior parameter distributions for common parameters using the implicit (top: white boxes) and explicit (bottom: dark grey boxes) ECM approaches for four different sites from North to South. Distributions are presented as box plots over the prior range of variation (corresponding to the range in the x-axis). Box plots depict the median (bold line), the 25th and 75th percentile (bars) and the 10th and 90th percentile (whiskers). The parameters shown are: $K_H$: the humus decomposition coefficient, $F_{Root}$: the fraction of C assimilates distributed to the roots and EM, $CN_{MIC}$: the microbial C/N ratio**






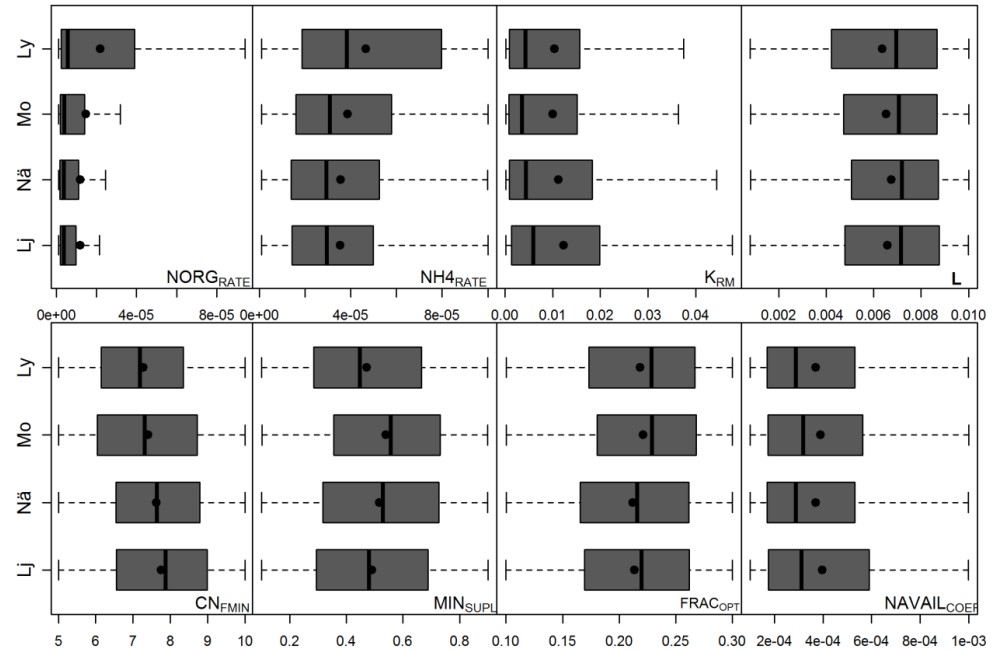

**Figure 8 Posterior parameter distributions of fungal specific parameters (from top left to bottom right): organic N uptake rate ($NORG_{RATE}$), ammonium uptake rate ($NH4_{RATE}$), respiration coefficient ($K_{RM}$), fungal litter rate coefficient ($L$), minimum fungal C/N ratio ($CN_{FMIN}$), fungal minimum N supply to plant ($MIN_{SUPL}$), optimum ratio between fungal and root C content ($FRAC_{OPT}$), N sensitivity coefficient ($NAVAIL_{COEF}$). Distributions are presented as box plots over the**
**prior range of variation (corresponding to the range in the x-axis). Box plots depict the median (bold line), the mean (black point), the 25th and 75th percentile (bars) and the 10th and 90th percentile (whiskers)**





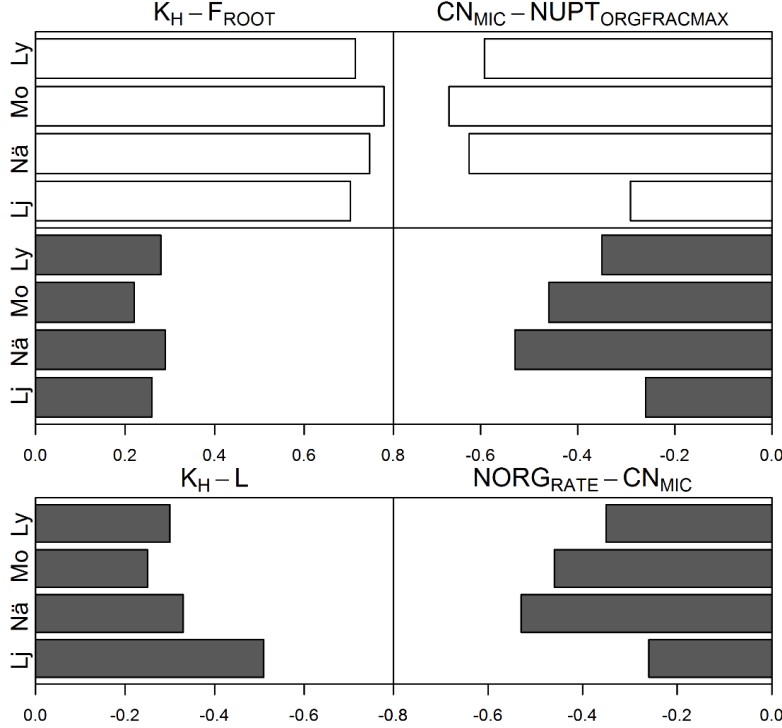

**Figure 9 Correlation between model parameters, given as the Pearson correlation coefficient, for the implicit and explicit ECM approaches. Top left: correlation between humus decomposition coefficient ($K_H$) and the fraction of C assimilates directed to EM and roots ($F_{ROOT}$). Top right: C/N of microbes ($CN_{MIC}$) and fraction of organic N available for uptake ($NUPT_{ORGFRACMAX}$). Correlation between fungal parameters: bottom left: humus decomposition coefficient ($K_H$) and fungal litter rate (L). Bottom right: fungal organic N uptake ($NORG_{RATE}$) and C/N of microbes ($CN_{MIC}$)**




**Table 1 Maximum and minimum parameters values prior to Bayesian calibration for nonlim, implicit and explicit model approaches**

**A.** Common parameters (all three approaches, including "implicit" approach)

| Parameter | Unit | Min | Max |
|---|---|---|---|
| *Humus decomposition* | | | |
| $K_H$ | $d^{-1}$ | 0.0001 | 0.001 |
| *Fraction of organic N available for uptake* | | | |
| $NUPT_{ORGFRACMAX}$ | $d^{-1}$ | 0.000001 | 0.0001 |
| *Fraction of root C allocation in mobile C* | | | |
| $F_{ROOT}$ | $d^{-1}$ | 0.4 | 0.6 |
| *C/N ratio of decomposing microbes* | | | |
| $CN_{MIC}$ | $d^{-1}$ | 15 | 25 |

**B.** Parameters of the "nonlim" approach

| Parameter | Unit | Min | Max |
|---|---|---|---|
| *Plant N Supply* | | | |
| ConstantNSupply | - | 0.1 | 0.7 |

**C.** Fungal parameters of the "explicit" approach

| Parameter | Unit | Min | Max |
|---|---|---|---|
| *Fungal N uptake* | | | |
| $NORG_{RATE}$ | g N gdw$^{-1}$ d$^{-1}$ | 0.000001[a] | 0.0001 |
| $NH4_{RATE}$ | g N gdw$^{-1}$ d$^{-1}$ | 0.000001[a] | 0.0001 |
| $NO3_{RATE}$ | g N gdw$^{-1}$ d$^{-1}$ | 0.000001[a] | 0.0001 |
| *Fungal respiration coefficient* | | | |
| $K_{RM}$ | $d^{-1}$ | 0.0002[b] | 0.05 |
| *Fungal litter rate* | | | |
| L | $d^{-1}$ | 0.0008[c] | 0.01 |
| *Minimum fungal C/N ratio* | | | |
| $CN_{FMIN}$ | $d^{-1}$ | 5[d] | 10 |
| *Fungal minimum N supply to plant* | | | |
| $MIN_{SUPL}$ | $d^{-1}$ | 0.1[e] | 0.9 |
| *Optimum fungi C allocation fraction* | | | |
| $FRAC_{OPT}$ | $d^{-1}$ | 0.1[f] | 0.3[f] |

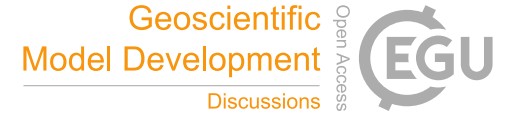

*N sensitivity coefficient*

| | | | |
|---|---|---|---|
| NAVAIL$_{COEF}$ | d$^{-1}$ | 0.0001 | 0.001 |

[a] Plassard et al., (1991), Chalot et al., (1995), Smith and Read, (2008)

[b] set equally to trees according to Thornley and Cannell, (2000)

[c] Staddon et a., (2003), Ekblad et al., (2013)

[d] Högberg and Högberg, (2002), Wallander and Nilsson, (2003)

[e] estimated

[f] Leake (2007), Staddon et al., (2003), Johnson et al., (2005)




**Table 2 Climatic and soil data, and initial settings of the four study soils applied in all model approaches**

| Sites | Location | Altitude (m asl) | Air temperature[a] (°C) | Precipitation[a] (mm) | Soil C (g C m$^{-2}$) | Soil N (g N m$^{-2}$) | Soil C/N[c] | Standing stock (g C m$^{-2}$)[c] | N deposition (kg N ha$^{-1}$ year$^{-1}$)[d] |
|---|---|---|---|---|---|---|---|---|---|
| Lycksele | 64°59'N 18°66'E | 223 | 0.7 | 613 | 7006 | 223 | 31.5 | 5371 | 1.5 |
| Mora | 61°00'N 14°59'E | 161 | 3.3 | 630 | 8567 | 295 | 29.1 | 7815 | 3.5 |
| Nässjö | 57°64'N 14°69'E | 305 | 5.2 | 712 | 9995 | 367 | 27.2 | 10443 | 7.5 |
| Ljungby-hed | 56°08'N 13°23'E | 76 | 7.1 | 838 | 10666 | 539 | 19.8 | 11501 | 12.5 |

[a] 30-year average/sum
[b] according to Skogsdata for a 100 year old forest (2003: http://www.slu.se/en/webbtjanster-miljoanalys/forest-statistics/skogsdata/)
[c] used as calibration parameter
[d] used as driving data





**Table 3 Prior values of variables used for model calibration and accepted relative uncertainty (A), and posterior model performance indicators (B): mean error (ME) between simulated and measured values, standard variation of ME (std), and summed log-likelihood of all accepted runs for simulated standing plant biomass (g C m$^{-2}$) and soil C/N ratio after the 100 year simulation period**


| A | PRIOR | | | |
|---|---|---|---|---|
| | Plant biomass (g C m$^{-2}$) | | Soil C/N ratio | |
| | Mean | relative uncertainty (%) | Mean | relative uncertainty (%) |
| Lycksele | 5371 | 0.1 | 32 | 0.1 |
| Mora | 7815 | 0.1 | 29.1 | 0.1 |
| Nässjö | 10443 | 0.1 | 27.2 | 0.1 |
| Ljungbyhed | 11501 | 0.1 | 19.8 | 0.1 |

| B | | POSTERIOR | | | | | | Runs |
|---|---|---|---|---|---|---|---|---|
| | | Plant biomass (g C m$^{-2}$) | | | Soil C/N ratio | | | |
| | | ME | std | loglike | ME | std | loglike | accepted (%) |
| *nonlim* | Lycksele | 37.6 | 531.1 | -7.7 | -5.8 | 1.3 | -3.8 | 25 |
| | Mora | 38.7 | 1098.2 | -8.4 | -3.9 | 1.4 | -3.0 | 41 |
| | Nässjö | 42.2 | 1021.3 | -8.3 | -2.7 | 1.6 | -2.6 | 48 |
| | Ljungbyhed | 1.0 | 1155.6 | -10.2 | 0.3 | 1.8 | -2.1 | 48 |
| *implicit* | Lycksele | -107.2 | 535.0 | -7.7 | -1.1 | 3.3 | -2.7 | 42 |
| | Mora | -98.3 | 787.1 | -8.1 | -1.1 | 2.7 | -2.5 | 45 |
| | Nässjö | -86.0 | 1036.2 | -8.0 | -1.0 | 2.5 | -2.4 | 46 |
| | Ljungbyhed | 100.1 | 1143.2 | -8.5 | 0.5 | 1.6 | -2.0 | 50 |
| *explicit* | Lycksele | -162.3 | 534.9 | -7.7 | -0.5 | 3.4 | -2.7 | 29 |
| | Mora | -215.4 | 809.1 | -8.2 | -0.3 | 2.7 | -2.4 | 32 |
| | Nässjö | -222.3 | 1041.2 | -8.1 | 0.0 | 2.5 | -2.3 | 30 |
| | Ljungbyhed | -139.0 | 1137.6 | -8.5 | 1.0 | 1.7 | -2.1 | 32 |






**Table 4 Comparison between modelled soil C and N of this study and literature value**

| Reference | Site | Ecosystem type | Forest age (years) | Soil C change (g C m$^{-2}$ yr$^{-1}$) | Soil N change (g N m$^{-2}$ yr$^{-1}$) |
|---|---|---|---|---|---|
| Svensson et al. 2008a | Lycksele | Coniferous on podzol | 100 | -5 | |
| | Mora | | | -2 | |
| | Nässjö | | | 9 | |
| | Ljungbyhed | | | 23 | |
| Lindroth et al. 2008 | Flakaliden | Coniferous on podzol | 39-42 (in 2002) | -79[a] | |
| | Knottåsen | | | -133[a] | |
| | Asa | | | -24 | |
| This study | Lycksele | Coniferous on podzol | 100 | | |
| | Mora | | | -6 to 13.1[b] | -0.2 to 0.6[b] |
| | Nässjö | | | -8.7 to -1.6[c] | -0.2 to -0.1[c] |
| | Ljungbyhed | | | | |

[a] mean of the highest and lowest error estimates

[b] implicit approach

[c] explicit approach