# Peer review of "Simulating ectomycorrhiza in boreal forests: implementing ectomycorrhizal fungi model MYCOFON into CoupModel (V5)"

_Geoscientific Model Development, 2017_

## Referee Comment (RC1) · L. L. Taylor (Referee) · 29 Sep 2017

**1   General remarks**

This paper addresses an important but often neglected aspect of boreal forests (and indeed many temperate and even tropical forests): ectomycorrhizal fungi. These fungi control the nutrient uptake of the infected roots and the relationship of the host tree with the soil organic matter. I was very interested to see what these authors are doing to rectify this omission in their model.

The work seems generally sound, and most of the points made below can be addressed without too much difficulty I should think. however, I think submission must have been rushed as the manuscript is riddled with grammatical and typographical errors. I started to make a list of these (included below) but lost the will to continue midway through; the authors need to have a fluent English speaker go through the text.

Introduction Consider putting in a table showing clearly what the different models described on line 48 thourgh 69 do.

Line 128 ECM growth is driven by sink strength of what?

Line 141 I had to read this sentence twice as I thought the authors were comparing the approach for ECM and root respiration to the approach of something else. however I think they have just treated ECM respiration the same way they have treated root respiration. Perhaps it would be clearer to say that there are two components (maintenance and growth) for both ECM and root respiration.

Line 159 Is NUPT$_{FRACMAX}$ the fraction of total soil N available for uptake, or is it the fraction of mineral N available for uptake? Please clarify.

section 2.1.5 My first reaction was that degree of mycorrhization had not been taken into account; then I realised mycorrhization degree was covered in section 2.1.6. Consider switching these two sections.

Line 166 Please add the scientific name for spruce. As this is Sweden it is probably *Picea abies*.

Line 211 I see the point of spinning up the vegetation from the time of establishment over the lifetime of the trees (100 years in this study), but soil C pools may take considerably longer than that to come to equilibrium. For example, 500 years is a more typical spinup to initialise soil C pools in dynamic vegetation models (DVMs). The legacy of recalcitrant C from previous forest growth in the soil must be accounted for by the initial standing C stock and C/N initial values in Table 2 which the footnote says are calibration parameters; maybe make this clear in the text. Unlike the calibration parameters of Table 1, the initial values assumed for soil C pools shown in Table 2 do not have minimum and maximum values associated with them, and standing stock does not appear in Table 3.

Line 212 I do not understand what is meant by this sentence: *A minimum of specific regional data were used at input values.* Does this refer to the number of driving variables input to the model (six in Table 2 plus two calibration parameters) or the amount of data used in the Bayesian analysis for each driving variable (30-year averages rather than time series or multiple values for each region)? I also don't understand at input values; does this mean *as input values* or does it mean something else? What is specific about the regional data?

Line 230 *The data likelihood function which determines the parameter sets **being candidate of** the posterior distribution* sounds odd; I assume that this sentence refers to the likelihood function determining **acceptance** of the parameter sets which will comprise the posterior distribution?

Line 235 Please make clear that $\omega_i$ is a vector.

Line 244 Replace $q_i + 1 = q_i + \varepsilon$ with $\theta_i + 1 = \theta_i + \epsilon$, using the same $\epsilon$ on lines 244 (the equation) and 245. Also, consider numbering the equations.

Line 280 Surely it is just parameters that are being calibrated and not processes?

Line 306 Do fungi take up the same amount of organic N when there is sufficient mineral N available?

Line 333 should *thus* be *and*? Is the sentence referring to N mineralisation? A higher organic matter turnover should mean higher N mineralisation.

Section 3.2.3 Is it necessary to list all these correlations? The figures are better for this; perhaps only discuss the most interesting ones?

Section 4 I take the authors' point that there is a dearth of comparison data, especially related to ECM, but are there really zero data? is there not one observation that can be compared with the model results? What about the Lindroth *et al* paper cited on line 419? How does the coupling of Mycofon to CoupModel affect the simulated soil respiration, for example? it is a bit difficult to claim that the model delivers "accurate" results (line 464) without any comparison to observations. Table 4 shows the Svensson *et al* model results so consistency with this other model could be worth showing in a figure.

**2   General remarks on figures**

Please include units and self-explanatory axes labels in all figures. Many readers will look at the abstract and figures before deciding to read the text; don't make readers go searching through the text for basic information. Where possible, don't even make readers read the captions carefully. In general, don't make readers do more work than absolutely necessary to understand what is being shown in the figures.

Figure 3 Is *total N litter production* the N released during decomposition or the N being added to the litter pool with fresh litter?

Figure 4 There is room to add *implicit model* and *explicit model* to the right of the figure so that readers can see immediately what the upper and lower graphs mean.

Figure 5 Is GPP in this figure simulated or measured? Any possibility of showing both simulated and measured GPP?

Figure 6 There is room to add *nonlim*, *implicit* and *explicit* to the right of the three panels, and to show the north-south gradient to the left of the Y axis of each panel.

Figure 7 Show the N-S gradient to the left of the panels (*ie* N next to Ly, and an arrow leading to S next to Lj). Thanks for adding *implicit* and *explicit*; please also add the meanings of the parameters on the X axis (*eg* $K_H$ is the humus decomp. coeff.) so that readers can see at a glance what is going on without having to search the text and tables.

Figure 8 Please give the units, especially for the rates. What is fungal litter rate, the rate of uptake from litter, or the rate at which hyphae die and contribute to the litter pool?

Figure 9 Does *C assimilates* mean NPP? Please make clear what parameters are being shown, so readers don't have to go searching (they probably won't have read the paper and won't realise the information is in one of the tables). Is the colour scheme here the same as in previous figures?

**3   Tables**

Table 2 Can it be made clearer that soil C/N and standing stock of C are calibration parameters and the other data are all driving data?

Table 3 Why are there no mean and uncertainty columns for soil C standing stock? according to Table 2 it's a calibration parameter.

Table 4 The Lindroth *et al* data shown here are means of the highest and lowest estimates, but the full ranges are shown for the Mycofon results. Would it not be better to show ranges for both?

Table 4 The Svensson *et al* data generally fall within the Mycofon model ranges; are these the ranges from the posterior distributions? why is the implicit approach shown for one site and the explicit approach for the other, and what are the results for the mean of the posterior? Could this material (Svensson *et al. vs.* model approaches) be presented as a figure? If Lindroth *et al.* measured respiration, surely that is a CoupModel output which could be compared to those measurements?

**4   Grammatical or typographical errors**

Here is a partial list of lines with errors, including suggested corrections.

In some cases I suggest rewordings of awkward clauses, in others I try to show the grammatical/typgraphical error and how to fix it. Original text is to the left of the arrow, and the replacement text to the right of the arrow. Actual changes (deletions to the left of the arrow, additions to the right of the arrow) are in **boldface**. I have tried to include enough text to make it clear why the change is necessary, such as where a grammatically plural noun is coupled with a grammatically singular verb.

Generally, models and approaches are preceded by *the*, which is omitted repeatedly throughout the text. A global change is not possible because there are a few occasions where *the* is present, or where it is OK to leave it out.

... Coup-Mycofon model provide → Coup-Mycofon model provide**s**

**known as** → **which are**

**the** ecosystem → ecosystem research show → research show**s**

Moore → **the** Moore

ANAFORE → **the** ANAFORE

ECM models ... simulate**s** → ECM models ... simulate that coupled → that **is** coupled approach which → approach**es** which

The "ECM implicit" does not → The "ECM implicit" **approach** does not incorporat**ing** → incorporat**es**

Plants ... **does** not → Plants ... **do** not

**in** Meyer → **by** Meyer (NB this is my personal preference but check the journal's policy: are citations considered to be the name of the paper, in which case *in* is fine, or do they refer to the authors who wrote the paper, in which case *by* makes more sense?)

**are distinguished** between → **distinguish** between follow → follow**s**

**to prevent fungi to die** → **preventing fungal death**

... as fungi **have** are more efficient → as fungi are more efficient plant **uptaking of** organic N → plant **taking up** organic N

203,213 **Tab.** 2 → **Table** 2 (likewise Table 3 in section 3.1.1; check the journal's policy, but in any case be consistent as *Table* is spelled out earlier in the manuscript)

management**s** → management

206, see 100 in Svensson → by Svensson effects **is** not → effects **are** not both**,** the → both the

**for a better constrain of** posterior → **to better constrain** posterior using the **Markov chain Monte Carlo method, also the Metropolis-Hastings walk** → using the **Metropolis-Hastings random walk Markov Chain Monte Carlo algorithm** (and please cite van Oijen et al 2005 here too)

The **random numbers are generated normally distributed having** a mean of zero → The **normally distributed random numbers** $\epsilon$ **have** a mean of zero

**parameter: ConstantNsupply for the spruce tree, is selected as calibration parameters** → **parameter ConstantNsupply for the spruce tree is a calibration parameter**

(C/Nmyc)**,** → (C/Nmyc)

The posterior model ... show → The posterior model ... show**s**

than that **of** using the "implicit" **and** "explicit" approach → than that using the "implicit" **or** "explicit" approach generally **N more** limited → generally **more N** limited southern site, Ljungbyhed than → southern site, Ljungbyhed**,** than

**show overestimation** by "implicit" approach but **change to underestimation** when "explicit" approach is used → **is overestimated** by **the** "implicit" approach but **colorredunderestimated** when **the** "explicit" approach is used

**the more processes and parameters included for calibration, less likely of finding an accept combination of parameter sets** → **as more parameters are included for calibration, acceptable combinations of parameter sets become less likely**

approach **show a much larger uncertainties than that of ECM "implicit" and** "explicit" approaches → approach **shows much larger uncertainties than either the "implicit" or** "explicit" approaches approach **simulate soil sequestration of N up to 2 g N** m$^{-2}$ y$^{-1}$ → approach **simulates up to 2 g soil N** m$^{-2}$ y$^{-1}$

**Besides the simulated soil C balance by** "nonlim" approach → **The simulated soil C balance by the** "nonlim" approach the soil **sequestrate C at most north site, Lycksele** but → the soil **sequesters C at the most northerly site, Lycksele,** but and decoupled → and **are** decoupled and "implicit" approach → and the "implicit" approach

**sites overall loss soil C by 6 and 5 g C m$^{-2}$ y$^{-1}$** → **soils lose 6 and 5 g C m$^{-2}$ y$^{-1}$, respectively**

**sites gain soil C by 3 and 13 g C m$^{-2}$ y$^{-1}$** → **soils gain 3 and 13 g C m$^{-2}$ y$^{-1}$, respectively**

For "explicit" approach → For **the** "explicit" approach in "implicit" approach → in **the** "implicit" approach show **an** overall minor C and N losses → show overall minor C and N losses

305,306 in "explicit" model → in **the** "explicit" model using "implicit" approaches → using the "implicit" approaches favour climate → favour**able** climate but "explicit" approach show a → but **the** "explicit" approach show**s** a in "explicit" approach → in the "explicit" approach

**show explicitly account** for ECM → **shows that explicitly accounting** for ECM

except **a** larger uncertainties in the "explicit" . → except **for** larger uncertainties in the "explicit" **approach**.

than **that of** the southern → than **for** the southern

The rest of the manuscript is riddled with errors like the ones above; please go through and fix them.

---

## Referee Comment (RC2) · Anonymous Referee #2 · 7 Oct 2017

The authors coupled an ectomycorrhizal fungi (ECM) model MYCOFON with a terrestrial biogeochemistry model to show that it is important to consider the plant-ECM interaction to properly model the ecosystem nitrogen dynamics. While I could see the legitimacy of their statement, I agree with the other reviewer that the paper seems submitted in a hurry: there are too many problems with grammars, syntaxes and formats, making it unreadable to some extent. Thus a thorough rewritten is needed before it can be better judged.

The language problem becomes more severe as the paper goes closer to the end. For instance, the description of 2.3.2 is pretty much a mess. I guess it is really awkward that

a paper would use "Bayesian calibration procedure" as section title. Personally, I think "Bayesian calibration of models" would be much more appropriate. The use of "data likelihood function" is also not accordant with the general terminology in data assimilation or Bayesian inference based model calibration. I strongly suggest the authors to read more relevant papers and revise the description to make it more readable.

As for the description of MCMC method, there are many excellent papers on this topic, however, the authors barely mention them and the description is again very poor.

As the paper reaches the results section, there are many more language/presentation problems. Many of the sentences are incomplete, such as missing verbs or wrong use of juxtapositions. The other reviewer has listed many of those issues and I won't add more to the list.

Further, I don't know why Appendix is shown in the middle of the paper. Have the authors carefully checked their submission? Is the wrong version uploaded?

Overall I suggest rejecting the paper for a resubmission.

———————————————

---

## Author Comment (AC1) · 7 Oct 2017

We welcome the comments made by the two referees for our discussion paper. Both referees mentioned the language problem. We feel sorry about that and we are now sending this paper for language editing by Copernicus Publications. We believe an updated version with improved English would be available soon.

---

## Short Comment (SC1) · 10 Oct 2017

Hongxing

As explained in https://www.geoscientific-model-development.net/about/manuscript_types.html GMD is expecting that authors upload the program code of models and the used data sets as a supplement or make the code and data available at a data repository preferable with an associated DOI (digital object identifier) for the exact model version described in the paper. If for some reason your code and/or data for the MYCOFON model cannot be made available in this form as the code availability section in your paper suggests you need to state the reasons why the code is not available or why

access is restricted. Please note that in the code accessibility section you can still point the reader to your web site for updates even if you provide the code as supplement or use a DOI for a release.

All the best Lutz Gross GMD Executive Editor

---

## Author Comment (AC2) · 11 Oct 2017

Dear referee, We are sorry about the language problem in the previous version of the paper. We have now improved the language by professional language editting service in UK. We hope this version (see supplymentary) would help you for a better judgement of the scientific mertits of the paper. We also welcome the comment on better discription of Bayesian calibration procedure and will include this and other comments later during comprehensive revision.

Please also note the supplement to this comment:

https://www.geosci-model-dev-discuss.net/gmd-2017-170/gmd-2017-170-AC2-supplement.pdf

**Supplement:**

**Simulating ectomycorrhiza in boreal forests: implementing ectomycorrhizal fungi model MYCOFON into CoupModel (V5)**

Hongxing He[1], Astrid Meyer[1,a], Per-Erik Jansson[2], Magnus Svensson[2], Tobias Rütting[1], Leif Klemedtsson[1]

[1] Department of Earth Sciences, University of Gothenburg, Po Box 460, Gothenburg 40530, Sweden

[2] Department of Land and Water Resources Engineering, Royal Institute of Technology (KTH), Brinellvägen 28, Stockholm 100 44, Sweden

[a] now at: Institute of Groundwater Ecology, Helmholtz Zentrum München, Ingolstädter Landstraße 1, Neuherberg 85764, Germany

*Correspondence to:* Hongxing He (hongxing.he@gu.se)

**Abstract**

Ectomycorrhizal fungi (ECM), the symbiosis between a host plant and mycorrhizal fungi, are shown to considerably influence the C and N flux between the soil, rhizosphere, and plants in boreal forest ecosystems. However, ECM are either neglected or presented as an implicit, non-dynamic term in most ecosystem models which can potentially reduce their predictive power.

In order to investigate the necessity of an explicit consideration of ECM in ecosystem models, we implement the previously developed MYCOFON model into a detail process-based, soil-plant-atmosphere model, CoupModel MYCOFON, which explicitly describes the C and N fluxes between ECM and roots. This new Coup-MYCOFON model approach (ECM explicit) is compared with two simpler model approaches; one containing ECM implicitly as a non-dynamic N uptake function (ECM implicit), and the other a version where plant growth has a constant N availability (nonlim). Parameter uncertainties are quantified using Bayesian calibration where the model outputs are constrained to current forest growth and soil conditions for four forest sites along a climate and N deposition gradient in Sweden over a 100-year period.

Our results show that the nonlim approach does not describe both the forest growth and soil C and N conditions properly. The ECM implicit/explicit approaches are able to describe current conditions with acceptable uncertainty. Meanwhile, the ECM explicit Coup-MYCOFON model provides a more detailed description of internal ecosystem flux and feedback of C and N fluxes between plants, soil and ECM. Our modeling highlights the need to incorporate ECM into current ecosystem models. We also provide a key set of posterior fungal parameters which can be further investigated and evaluated in future ECM studies.

**1. Introduction**

Boreal forests cover large areas on the Earth's surface and are generally considered as substantial carbon (C) sinks (Dixon et al., 1994). The sink strength is determined through the balance between major C uptake and release processes, i.e., plant photosynthesis and both autotrophic and heterotrophic respiration, and is largely controlled by nitrogen (N) availability (Magnani et al., 2007). Numerous studies have shown that soil nitrogen availability is the main driver for plant and microbial growth (Klemedtsson et al., 2005; Lindroth et al., 2008; Luo et al., 2012; Mäkiranta et al., 2007; Martikainen et al., 1995). Thus, a proper description of N dynamics in ecosystem models

is prerequisite for precisely simulating plant-soil C dynamics and greenhouse gas (GHG) balance (Maljanen et al., 2010; Schulze et al., 2009; Huang et al., 2011). Ecosystem models, however, vary considerably in their representation of N fluxes: from very simplified presentations (e.g., the LPJguess model: Sitch et al., 2003; Smith et al., 2011) to very complex approaches which aim to capture the whole N cycle (e.g., LandscapeDNDC: Haas et al., 2012; CoupModel: Jansson and Karlberg, 2011).

Ectomycorrhizal fungi (ECM) are common symbionts of trees in boreal forests. ECM are more efficient than roots in taking up different N sources from the soil (Plassard et al., 1991), as well as store vast amounts of N in their tissues (Bååth and Söderström, 1979) and can cover a large fraction of their host plants' N demand (Leake, 2007; van der Heijden et al., 2008). Further, ECM are shown to respond sensitively to ecosystem N availability and are generally considered as adaptation measures to limited N conditions (Wallenda and Kottke, 1998; Read and Perez Moreno, 2003; Kjoller et al., 2012; Bahr et al., 2013). Previous research shows ECM can receive between 1 and 25% of the plants' photosynthates and constitute as much as 70% of the total soil microbial biomass, thus having a major impact on soil C sequestration in boreal forests (Staddon et al., 2003; Clemmensen et al.; 2013). Overall, the functions and abundance of ECM fungi constitute numerous pathways for N turnover in the ecosystem and considerably influence the magnitude and dynamics of C and N fluxes.

Nevertheless, ECM have rarely been considered in ecosystem models (for an overview about modeling ectomycorrhizal traits see Deckmyn et al., 2014). To our knowledge, only five ecosystem models have implemented ECM to various degrees: The ANAFORE model (Deckmyn et al., 2008), the MoBiLE environment (Meyer et al., 2012), the MyScan model (Orwin et al., 2011) and more recently the Moore et al. (2015) and Baskaran et al. (2016) ECM models. In the ANAFORE model, ECM are described as separate C and N pools. However, this model it does not distinguish between mycorrhizal hyphae and mantle. The C allocated from the host tree to ECM is simulated as a zero order function, further regulated by nutrient and water availability. ECM can also facilitate organic matter decomposition in the ANAFORE model. The MyScan model uses a similar approach for ECM C uptake and dynamics but does not, to our knowledge, include the influence of water availability on ECM. In both models, ECM transfer of N to the host is regulated by the C/N ratios of the plant and fungi. In the MoBiLE model, C allocation to ECM is more complex than that in ANAFORE and MyScan models, and the N allocation to the host by the ECM can feed back into their C gains. Although, the N allocation to the host plant is described similarly to the other two models. In MoBiLE, mycorrhiza are further distinguished between hyphae and mantle, but cannot degrade organic matter. Hyphae and mantle differ in their capacity to take up N, and the mantle has a slower litter production rate than that of hyphae. Both Moore et al. (2015) and Baskaran et al.'s (2016) ECM models represent the ECM as a separate model pool and explicitly simulate ECM decomposition, but with much simpler process descriptions, and the interaction with environmental functions are neglected.

The overall aim of this study is to present a new version of the CoupModel, coupled with an explicit description of ECM, to investigate how the explicit consideration of ECM affects overall model performance and uncertainty. Thus, we implement the previously developed MYCOFON model (Meyer et al., 2010; Meyer et al., 2012) into the well-established soil-plant-atmosphere model, CoupModel (Jansson, 2012). The implemented MYCOFON model contains a very detailed description of fungal C and N pools, and all major C and N exchange processes (i.e., litter production, respiration, C uptake, N uptake). Fungal growth and N uptake respond dynamically to environmental functions and plant C supply in the new Coup-MYCOFON model (Fig. 1). This detailed ECM

explicit modeling approach (hereafter called "ECM explicit") is further compared with two simpler modeling approaches – the "ECM implicit" and "nonlim" approaches – which already exist in CoupModel. The "ECM
80  implicit" approach does not represent the ECM as a separate pool but incorporates ECM into the roots implicitly. Plants are thus allowed to take up additional organic N sources statically, and do not respond to environmental functions. The "ECM implicit" approach has been used in a similar way by Kirschbaum and Paul, (2002) and Svensson et al. (2008a). The "nonlim" approach assumes an open N cycle and plant growth are limited by a constant N availability (e.g., in Franklin et al., 2014). These three ECM modeling approaches constitute most of
85  the current ECM representations in ecosystem models, and are tested by four forest sites situated along a climate and N fertility gradient across Sweden (Fig. 2). Bayesian calibration is used to quantify the uncertainty of model parameters and identify key parameter sets.

**2. Methods**

**2.1 Model description**

90  The CoupModel ("Coupled heat and mass transfer model for soil-plant-atmosphere systems", Jansson and Karlberg, 2011) is a one-dimensional process-oriented model, simulating all the major abiotic and biotic processes (mainly C and N) in the soil-plant-atmosphere system. The basic structure is a depth profile of the soil for which water and heat flows are calculated based on defined soil properties. Plants can be distinguished between understory and overstory vegetation, which allows simulating competition for light, water, and N between plants.
95  The model is driven by measured climate data – precipitation, air temperature, relative humidity, wind speed, and global radiation – and can simulate ecosystem dynamics in hourly/daily/yearly resolutions. A general structural and technical overview of the CoupModel can be found in Jansson and Moon (2001) and Jansson and Karlberg (2011), and a recent overview of the model was also given by Jansson (2012). The model is freely available at www.coupmodel.com. The CoupModel is complemented with an ectomycorrhizal module (MYCOFON, Meyer
100  et al., 2010) which allows the direct simulation of the C and N uptake processes of ECM. The MYCOFON model is described in detail by Meyer et al. (2010), and here only the key processes of plant and fungal growth, as well as 
[revised manuscript text omitted]

parameters of the respective ECM modeling approaches ("nonlim", "implicit", and "explicit"), as well as analyze model uncertainties and dependencies between parameters. Uncertainties in parameter values are expressed as probability distributions. The posterior probability distributions of parameters are estimated by considering the prior distribution and likelihood function in the calibration procedure. The likelihood function is determined by the measured data on output variables and the respective error estimates of the simulated model output. The Bayesian calibration as applied in this study is briefly described below, however for a detailed description of the general methodology see e.g., van Oijen et al. (2005) or Klemedtsson et al. (2008).

**2.3.2 Bayesian calibration of models**

The data likelihood function which determines the parameter sets being candidate of the posterior distributions is based on the assumption that the model errors, i.e., the differences between simulated and observed values, are normally distributed and uncorrelated (van Oijen et al., 2005). Furthermore, model errors are assumed to be additive so that the log-likelihood function reads:

$$\log L = \sum_{i=1}^{n} \left( -0.5 \left( \frac{y_i - f(\omega_i \cdot \theta_i)}{\sigma_i} \right) - 0.5 \cdot \log(2\pi) \right) - \log(\sigma_i)$$

where $y_i$ = observed values, $f(\omega_i \cdot \theta_i)$ = simulated values for a given model input $\omega_i$ and parameter set $\theta_i$, $\sigma_i$ = standard deviation across the measured replicates, and $n$ = number of variables measured.

In this study, a measured uncertainty of 10% for both the soil C/N ratio and the standing stock biomass data is used. The uncertainty estimate is low (van Oijen et al., 2005) as our intention is to force the model to simulate tree biomass and soil C/N ratio precisely, to better constrain posterior parameter distributions for the respective model approaches and sites. Candidate parameter sets are generated by investigating the parameter space using the Metropolis-Hastings random walk Markov Chain Monte Carlo algorithm (van Oijen et al., 2005). For each simulation, the model's likelihood is evaluated for a certain parameter set. After each run, a new parameter set is generated by adding a vector of random numbers ε to the previous parameter vector:

$$q_{i+1} = q_i + e$$

where $\theta_i$ = previous parameter vector, $\theta_{i+1}$ = new parameter vector, and $\varepsilon$ = random numbers.

The normally distributed random numbers $\varepsilon$ have a mean of zero and a step length of 0.05; i.e., 5% of the prior parameter range as proposed by van Oijen et al. (2005). We performed $10^4$ runs for each ECM modeling approach and site to ensure posterior convergence.

**2.3.3 Model parameters chosen for calibration**

The different ECM modeling approaches are calibrated for a comprehensive set of key parameters which are chosen according to their function as regulating factors of the C and N fluxes in the plant-soil-mycorrhiza continuum (Table 3). In the "nonlim" approach, the constant N supply parameter *ConstantNsupply* for the spruce tree is a calibration parameter. In the "implicit" approach, the fraction of organic N available for plant uptake (*NUPT_ORGFRACMAX*) is included in the calibration based on Svensson et al. (2008a). For the ECM "explicit"

approach, all fungal parameters in MYCOFON including fungal growth (C and N assimilation and uptake, C and N losses), overall N uptake and plant N supply, respiration, and littering are calibrated. For all three approaches, the humus decomposition rate ($K_H$), the C/N ratio of microbes ($CN_{mic}$) regulating soil mineralization, and the fraction of plant C assimilates allocated to the rooting zone ($F_{ROOT}$) regulating fungal growth are also calibrated.

260 Overall, we include a rather generous number of parameters for Bayesian calibration following Klemedtsson et al. (2008) who emphasize the importance of a holistic perspective when considering model parameters. Prior distributions of parameters are assumed to be uniform; i.e., each value is equally probable with given minimum and maximum values (Table 3). Values were chosen based on either previous modeling applications (e.g., plant parameters determined by Svensson et al. 2008a, b) or literature data (Table 3).

265 **3. Results**

**3.1 Comparison of the three modeling approaches**

**3.1.1 General ability to reproduce tree growth and soil C/N**

The three modeling approaches show different abilities in reproducing current plant growth and soil C/N ratio after calibration (Table 3B). The posterior model in the "implicit" and "explicit" approaches shows better

270 performance of simulating soil C and N, as indicated by the soil C/N ratio, than the "nonlim" approach. The latter tends to simulate a lower soil C/N ratio, indicated by the negative mean errors (ME) in the posterior model (ME is the difference between the simulated and measured values) (Table 3B). The ME by the "nonlim" approach is also two to five times higher than that when using the "implicit" or "explicit" approach (Table 3B). The "nonlim" approach tends to overestimate plant growth as the posterior mean of ME for plant C is always positive, while the

275 "implicit" and "explicit" approaches tend to show an underestimation (Table 3B).

All posterior models underestimate soil C/N for the northern sites which are generally more N limited, but gradually switch to overestimation at the southern sites. The model with the "nonlim" approach simulates better plant growth for the most southern site, Ljungbyhed, than the other sites. Further, modeled plant growth at Ljungbyhed is overestimated by the "implicit" approach, but underestimated when the "explicit" approach is used

280 (Table 3B). The acceptance of model runs in posterior is higher for the "nonlim" (25 to 48%) and "implicit" approaches (42 to 50%), followed by the "explicit" approach (30 to 33%) which can be explained by model complexity; i.e. as more parameters are included for calibration, accepted combinations of parameter sets become less likely. No major differences are found for the summed log-likelihood for both calibration variables (Table 3B).

285 **3.1.2 C and N budget**

Modeled major ecosystem N fluxes, soil C, and N balance in the posterior are shown in Figure 3. In general, the "nonlim" approach shows much greater uncertainties in the modeled N fluxes than either the ECM "implicit" or "explicit" approaches. The "nonlim" approach simulates soil sequestration of N up to 2 g N m$^{-2}$ yr$^{-1}$ for all the sites, but much lower or close to zero values are found when using other two modeling approaches (Fig. 3). Soil

290 N is expected to reach a steady state over a period of 100 years (Svensson et al., 2008a). Therefore, the "nonlim" approach largely overestimates soil N sequestration which can be attributed to the assumed "virtual" constant N uptake from the unlimited source. According to our model predictions, this "virtual" N fraction accounts for 20 to

30% of the total plant N uptake. The simulated soil C balance by the "nonlim" approach also contrasts with that of soil N, where the soil sequesters C at the most northern site, Lycksele, but loses C at a rate of 6 to 17 g C m$^{-2}$ yr$^{-1}$ for the other three sites (Fig. 3). Therefore, soil C and N are not in steady states and are decoupled in the "nonlim" approach over the simulated 100-year period.

However, the "implicit" and "explicit" approaches show a strong coupling between soil C and N (Fig. 3). That is, for the "implicit" approach, Lycksele and Mora soils lose 6 and 5 g C m$^{-2}$ yr$^{-1}$ respectively, while Nässjö and Ljungbyhed soils gain 3 and 13 g C m$^{-2}$ yr$^{-1}$ respectively. Similarly, Lycksele and Mora lose N by 0.2 and 0.1 g N m$^{-2}$ yr$^{-1}$, while Nässjö and Ljungbyhed gain N by 0.3 and 0.6 g N m$^{-2}$ yr$^{-1}$. For the "explicit" approach, soil C and N losses at the two northern sites are slightly higher than that in the "implicit" approach (Fig. 3). In contrast to the "implicit" approach, the two southern sites also show overall minor C and N losses with large standard deviations (Fig. 3). Modeled N litter production increases by 1 to 30% compared to the "implicit" approach, but N losses due to uptake and leaching also increase by 10 to 50% for Lycksele and Ljungbyhed respectively (Fig. 3). The increased litter addition of easily degradable C and N stimulates microbial activity, thus leading to a higher microbial respiration which explains the minor losses of C and N in the southern sites in the "explicit" model. The higher N leaching in the "explicit" model can be attributed to a higher uptake from organic N (eqs. 11.5, 11.6, Table 1.B) and a stimulated microbial growth thus increases net mineralization, both of which leave more mineral N in the soil (Fig. 4).

Simulated plant gross primary production (GPP) using the "explicit" and "implicit" approaches shows an increasing trend from the North to South due to more favorable climates and N availability for spruce forest growth, but the "explicit" approach shows a higher predicted GPP than the "implicit"; i.e., 7% in Ljungbyhed to 12% in Lycksele (Fig. 5). C losses from autotrophic respiration are lower in the "explicit" approach (Fig. 5). Trees in the northern regions seems slightly more efficient in taking up C shown by the higher biomass efficiency (NPP/GPP, Fig. 5). Overall, our results show that explicitly accounting for ECM in boreal forest ecosystems can have a considerable impact on the predicted C and N dynamics both for the plants and soil.

**3.2. Posterior parameter distributions**

**3.2.1. Posterior distributions of common parameters**

The posterior distributions differ from the prior uniform distributions for all modeling approaches and parameters, reflecting the efficiency of Bayesian calibration (Fig. 6 and Fig. 7). The posterior *constantNsupply* parameter in the "nonlim" approach shows the lowest values at Lycksele and the highest at Ljungbyhed. This means a higher N supply is necessary at the southern sites to explain the observed tree biomass and soil C/N ratio. No significant differences in parameter values – microbial C/N ratio ($CN_{MIC}$), humus decomposition coefficient ($K_H$), and the fraction of C allocated to roots, $F_{ROOT}$ – in the "nonlim" approach are found for the different sites (data not shown). The organic N uptake parameter in the "implicit" and "explicit" approaches ($NUPT_{ORGFRACMAX}$) shows an opposite pattern with the highest values for Lycksele and lowest for Ljungbyhed (Fig. 6). Similar posterior distributions of $NUPT_{ORGFRACMAX}$ parameters are found in both approaches, except for larger uncertainties in the "explicit" approach. Besides, a much wider range of posterior parameter values are found for the northern sites than for the southern sites (Fig. 6). This also explains the smaller simulated ME of soil C/N in the southern sites (Table 3). Both approaches demonstrate that the plant and soil conditions at the northern sites could not be simulated without an enhanced uptake of organic N.

When the "implicit" approach is used, the posterior humus decomposition coefficient $K_H$ shows higher values for the northern sites and decreases along the studied transect, demonstrating a modeled higher organic matter turnover and thus soil mineralization for northern sites (Fig. 7). A less clear tendency is identified for the fraction of C allocated to roots, $F_{ROOT}$ parameter, but with a slight tendency towards higher values at the southern sites. Microbial C/N ratio and $CN_{MIC}$ parameters for both "implicit" and "explicit" approaches show similar posterior distributions for the three northern sites. However, much lower values are obtained for the southernmost Ljungbyhed site (Fig. 7), reflecting a more soil N rich environment. Overall, parameters are less constrained and only minor differences between sites are found when the "explicit" approach is used (Fig. 7).

**3.2.2 Fungal specific parameters**

The posterior distributions of all fungal specific parameters are constrained to log-normal or normal distributions (data not shown). The mean values of N uptake parameters ($NORG_{RATE}$, $NH4_{RATE}$, $NO3_{RATE}$) show a decreasing trend from the northern to southern sites (Fig. 8). This again means a higher ECM fungal N uptake rate is necessary to explain the observed soil and plant data at the more N-limited northern sites. Similarly, lower values for the northern and higher values for the southern regions are also found for the minimum fungal C/N ratio parameter ($CN_{FMIN}$). The optimum ratio between fungal and root C content, $FRAC_{OPT}$, tends to be higher at the northern sites and lower at the southern sites, also implying a modeled higher ECM biomass at the northern sites (Fig. 8). $MIN_{SUPL}$, the minimum supply of N from fungi to the host plant parameter, does not show a clear trend. Further, differences of the other ECM parameters for the four regions are minor (Fig. 8).

**3.2.3 Correlation between parameters**

An overview of correlations for all posterior model parameters can be found in the Appendix in Tables A2, A3, and A4. Parameters showing correlation with each other (defined here as a Pearson correlation coefficient $r \geq 0.3$ or $\leq -0.3$) are identified as the key parameter sets, shown in Figure 9. When the "implicit" approach is used, a significant positive correlation is obtained between the humus decomposition rate, $K_H$, and the fraction of C allocated to rooting zone, $F_{Root}$. The organic N uptake parameter, $NUPT_{ORGFRACMAX}$ and microbial C/N ratio, $CN_{MIC}$ are negatively correlated, except for a weak correlation for Ljungbyhed (Fig. 9). A weak correlation between $NUPT_{ORGFRACMAX}$ and $F_{ROOT}$ is found for the Nässjö site only (see Table A2 Appendix). For the "explicit" approach, the correlation coefficients between $K_H$ and $F_{ROOT}$ are decreased, and there is also a weaker correlation between $NUPT_{ORGFRACMAX}$ and $CN_{MIC}$ for all sites compared to the "implicit" approach (Fig. 9). No clear correlation between common and fungal parameters is obtained. Further, a negative correlation occurred between microbial C/N ratio, $CN_{MIC}$, and the fungal N uptake rates ($Norg_{RATE}$, $NH4_{RATE}$, $NO3_{RATE}$), but only for the Northern sites Lycksele and Mora (Table A4). A moderate correlation is found for $K_H$ and the fungal litter rate, $L$ for Ljungbyhed. Among fungal parameters, the N uptake rates moderately correlate to the litter production rate, $L$ at the northern sites, but correlations at Nässjö and Ljungbyhed are either non-existent or weak (Table A4). Our identified inter-connections and correlations between the parameters in general reflect the complex and interrelated nature of ECM, soil, and plant interactions. But more importantly, they also highlight the need to calibrate a number of parameters simultaneously rather than calibrating just one single parameter when applying such detailed ecosystem models (He et al., 2016; Klemedtsson et al., 2008).

**4. Discussion**

370      Our new version of the CoupModel provides a detailed model predictive framework to explicitly account for ECM in the plant-soil-ECM continuum. Model comparison to two earlier ECM modeling approaches show that ECM have to be included in ecosystem models ("implicitly or explicitly") to be able to describe the long term plant and soil C and N development. Overall, the models perform similarly in the "implicit or explicit" approaches, while the "nonlim" approach significantly overestimates soil N uptake. Our results thus confirm that ECM have a

375      substantial effect on soil C and N storage, and can also impact forest plant growth. But more importantly, including them into ecosystem models is both important and feasible.

**4.1 ECM alter plant-soil C and N dynamics**

The "nonlim" model in this study shows overestimations of plant growth and also larger biases in soil N than "implicit and explicit" approaches even after calibration (Table 3). A previous CoupModel application by Wu et

380      al. (2012) demonstrated that the "nonlim" approach could possibly describe short term carbon and water dynamics for a Finnish forest site. The same approach with open N cycle was also used in Franklin et al. (2014) to simulate Swedish forest biomass growth and its competition with ECM. It therefore seems that plant growth and thus the C cycle can be simulated reasonably with the "nonlim" approach, although a slight trend of overestimation is exhibited. However, our modeling further indicates that this simplified approach has an uncoupled soil C and N

385      in its model structure (Fig. 3) and is thus not recommended for future long term soil C and N predictions. This is also reflected in the posterior model parameter distributions where the *constantNSupply* rate parameter shows primary control on the modeled plant growth and soil conditions. Other parameters have minor or no importance for the model results, reflecting an oversimplified soil C and N model structure. Thus, the following discussion focuses on the other two modeling approaches.

390      Moore et al. (2015) demonstrate that ECM substantially affect soil C storage, and the impact is largely dependent on plant growth. The present study additionally shows that ECM representation in ecosystem models could also feedback into the predicted plant growth through N. As when ECM are implicitly included, the model simulates a 78 (average of four sites, std: 102) g C m$^{-2}$ lower plant biomass compared to the "nonlim" approach. Further, when they are explicitly included, the difference becomes even larger, 214 (50) g C m$^{-2}$ (Table 3). Including ECM

395      in the model thus shows decreased plant growth. This somehow differs from the generally assumption that growth should be higher in mycorrhized plants, i.e., boreal forests, due to optimized nutrient supply (Pritsch et al., 2004; Finlay et al., 2008, see also review by Smith and Read, 2008). This discrepancy could be possibly due to: *1)* an enhanced root litterfall due to a higher turnover of fungal mycelia, shown by a higher litter turnover rate (calibrated litter rate of ECM is 0.0075 d$^{-1}$, Fig. 8, whereas the litter rate of roots is 0.0027 d$^{-1}$, Table A1(d)). When ECM is

400      explicitly considered, litter production is modeled higher than in the "implicit" approach (difference from 50 to 110 g C m$^{-2}$ yr$^{-1}$, data not shown). These two modeling approaches thus show large differences in simulating litter production. Field data are further needed to clarify this. The discrepancy could also be due to: *2)* an enhanced N immobilization in ECM under N-limited conditions because ECM retain more N in their own biomass in response to plant allocation of newly assimilated C (Nehls et al., 2008). The constrained optimum fungi C allocation fraction

405      parameter shows an increasing trend towards the more northern sites (Fig. 8). This indicates a higher proportional C "investment" by the forest plants on ECM in northern, N limited conditions. The resulting ECM-plant competition for N could then potentially result in decreased plant N uptake, and thus plant growth (Näsholm et al.,

2013). Finally, the discrepancy could be due to *3)* biases in simulating ECM N uptake due to model/parameter uncertainties caused by high variability among fungal species and the scarcity of direct measurements in the field

410    (Smith and Read, 2008; Clemmensen et al., 2013). The current "explicit" approach implements many biotic interactions and internal feedbacks within the plant-soil-ECM continuum. However, increasing the number of processes and interactions in an already complex ecosystem model will not necessarily generate more reliable model predictions; as shown here, the parameters in the "explicit" approach have a larger uncertainty range even after calibration. Thus, future model evaluation, together with more detailed ECM data, are needed to better

415    understand the tightly coupled soil-ECM-plant continuum.

Both approaches simulate the soil C and N stock well (Table 3). The respective net change in the soil C pools of the "implicit" approach corresponds well to the results by Svensson et al. (2008a) who also suggest a small loss of soil C in the north while a gain in the south. However, when the "explicit" approach is used, the soils in the south are also predicted to lose C and N, mostly due to enhanced soil respiration (see section 3.1.2). It is difficult

420    to evaluate which approach gives a more realistic prediction as field data are not available. However, Lindroth et al. (2008), who measured C fluxes at three sites in Sweden at comparable latitudes and on comparable soils, found a similar trend in the soil net C change as simulated by the "explicit" approach here, but with a higher loss rate of between 24 and 133 g C $m^{-2}$ $year^{-1}$ (Table 4).

**4.2 Parameter and model responses to different environmental conditions**

425    Our modeling results show a consistent pattern with observations (e.g., Hyvönen et al., 2008; Näsholm et al., 2013) that at the northern N limited sites, organic N uptake by ECM is highly important for plant growth, becoming less important as N availability increases southwards. As indicated by the "explicit" approach, the mycorrhization degree of tree roots at Lycksele and Mora (> 90%) is much higher than that of Ljungbyhed (15%), thus the majority of modeled N uptake is through fungal mycelia in northern sites. The constrained fungal organic and mineral N

430    uptake parameters also show a decreasing trend (Fig. 8). Similarly, the organic N uptake parameter, $NUPT_{ORGFRACMAX}$, in the "implicit" approach decreases from north to south, but with a clearer site to site difference; thus indicating a stronger response to environmental conditions (Fig. 6). This is expected as more detailed ECM processes in the "explicit" approach should result in more internal interaction and feedback, thus damping the direct environmental regulations. Current modeling also indicates a higher mineralization shown by the humus

435    decomposition coefficient, $K_H$, in the northern sites. However, the decomposition of mineralization is also enhanced when ECM is "explicitly" considered (Fig. 7). This collaborates well with findings from field measurements and recent modeling studies that ECM are able to degrade complex N polymers in humus layers, thus enhancing soil N transformation under low N conditions (Moore et al., 2015; Lindahl and Tunlid, 2015; Baskaran et al., 2016).

440    In the "implicit" approach, the humus decomposition coefficient, $K_H$, was found to correlate with the fraction of C that allocates to the rooting zone, $F_{ROOT}$. As ECM are implicitly included in the roots, this correlation therefore indirectly indicates a strong connection of the root-ECM symbiosis and soil N availability. But when ECM are explicitly considered, this becomes less important, again due to a more detailed internal cycling of N supply and uptake from the fungi; i.e., plant N supply is further regulated by simulated higher litter input and N uptake from

445    the soil in the "explicit" model (Fig. 3, Fig. 9). Our modeling shows that fungal litter rates correlate to fungal N uptake rates in the "explicit" model, and that fungal N uptake rates have significant correlations to the microbial

C/N ratio, $CN_{MIC}$, for the northern sites (Fig. 9). This indicates the close coupling between fungal N uptake (N loss from the soil) and fungal litter production (N input to the soil). Such an incorporated tight cycle is of major importance for the overall plant N supply, and thus C and N dynamics of plant and soil at the N limited sites in the boreal forests.

Most fungal parameters in the "explicit" approach are not – or only weakly – dependent on the differing environmental conditions along the modeled transect, except for the N uptake parameters and fungal minimum C/N ratio, $CN_{FMIN}$, which show different mean values (Fig. 8). As such, these parameters need to be calibrated carefully when further applying the model to other sites with different soil nutrient levels or climate conditions.

One of the major difficulties of the explicit inclusion of ECM in ecosystem models is the unknown turnover of fungal mycelia (Ekblad et al., 2013). Previously reported turnover rates of newly formed mycelia vary from days to weeks, even up to 10 years (Staddon et al., 2003; Wallander et al., 2004), mostly due to the high variability in ECM species and structures (see review by Ekblad et al., 2013). Besides, root turnover rates can also vary considerably between species, soils, and climate zones (Brunner et al., 2012). Thus far, very few studies have reported parameterization of C and N cycling for ECM in boreal forests. Our calibration study thus provides a key set of ECM parameters that can be further tested through field observation, and more importantly, together with the identified correlations with the variables, can act as a guidelines for future ECM modeling studies.

**5. Conclusions**

The key components and features of the Coup-MYCOFON model have been described. The new version of CoupModel simulates C and N fluxes and pools, with the capacity of explicitly accounting for the links and feedback between the ECM, soil, and plant. The comparison of three common ECM modeling approaches which differ in complexity demonstrates that the simple "nonlim" approach cannot describe the measured soil C/N ratio, and also overestimates measured forest growth. When including ECM either implicitly or explicitly, both models deliver accurate long-term quantitative predictions on forest C and N cycling with simultaneous considerations of the impact of ECM fungi on ecosystem dynamics. However, they slightly underestimate forest growth. The ECM explicit Coup-MYCOFON model provides a more detailed description of internal ecosystems flux and feedback of C and N. The constrained ECM parameter distributions presented in this study can be used as guidelines for future model applications. Overall, our model implementation and comparison suggest ecosystem models need to incorporate ECM fungi into their model structure for a better prediction of ecosystem C and N dynamics, and the new version of CoupModel provides such an option.

**6. Code and data availability**

The model and extensive documentation with tutorial excises are freely available from the CoupModel home page http://www.coupmodel.com/ (CoupModel, 2015). The source code can be requested for non-commercial purposes from Per-Erik Jansson (pej@kth.se). CoupModel is written in the C programming language (code also available in Fortran) and run mainly under Windows/Linux systems. Inputs and outputs are in binary format. The version used as the basis for the present development was version 5 from 12 April 2017. The simulation files including the model and calibration set-up, the used parameterization, and corresponding input and validation files can be requested from Hongxing He (hongxing.he@gu.se). However, the majority of the input and output data used for

the current modeling is available publicly through SMHI or previous publications, i.e., Svensson et al. (2008).
Please contact the first author of this publication or Per-Erik Jansson if you plan an application of the model and further collaboration.

*Acknowledgements*: Financial support came from the Swedish Research Council for Environment, Agricultural Sciences and Spatial Planning (FORMAS), the strategic research area BECC (Biodiversity and Ecosystem services in a Changing Climate, www.cec.lu.se/research/becc), and the Linnaeus Centre LUCCI (Lund University Centre for studies of Carbon Cycle and Climate Interactions).

APPENDIX:
**Table A.1 Model functions describing plant growth, fungal growth, model parameters, and response functions of plant and ECM. Parameters are always entitled with capital letters**

495

**Table A.1 (a) Description of plant model functions. (i = fine roots, coarse roots, stem, leaves, grain, mobile)**

| No. | Equation |
|---|---|

Plant photosynthesis (g C m$^{-2}$ d$^{-1}$):

500    1    $$c_{atm \to plant} = \varepsilon_L \times f(T_1) \times f(CN_1) \times f\left(\frac{E_{ta}}{E_{tp}}\right) \times r_S$$

$\varepsilon_L$ = coefficient for radiation use efficiency, $f(T_l)$, $f(CN_l)$, $f(E_{ta}/E_{tp})$ = response functions to leaf temperature, leaf CN, and air moisture (see Table A.1 (c)), $r_s$ = global radiation absorbed by canopy.

Plant maintenance respiration (g C m$^{-2}$ d$^{-1}$):

505    2.1    $$c_{plantM \to atm} = c_i \times K_{RMi} \times f(T_l)$$

$c_i$ = C content of each respective plant compartment i (g C m$^{-2}$) and $K_{RMi}$ is a coefficient.

Plant growth respiration (g C m$^{-2}$ d$^{-1}$):

2.2    $$c_{plantG \to atm} = c_{m \to i} \times K_{RGi}$$

$c_{m \to i}$ = C gain (growth) of each plant compartment i (g C m$^{-2}$ d$^{-1}$) and $K_{RGi}$ is a coefficient.

510

Plant litter production (g C m$^{-2}$ d$^{-1}$):

3    $$c_{i \to lit} = c_i \times L_i$$

where $C_i$ is the C content of each plant compartment i (g C m$^{-2}$) and $L_i$ (= 0.0027 d$^{-1}$) is a coefficient.

515    Plant nitrate and ammonium uptake (g N m$^{-2}$ d$^{-1}$) (only shown for nitrate, equivalent for ammonium):

4.1    $$n_{NO3 \to plant} = dem_{Nplant} \times r_{NO3}$$    if $f(n_{minavail})$ x $n_{NO3soil} \geq dem_{Nplant}$ x $r_{NO3}$

4.2    $$n_{NO3 \to plant} = f(n_{minavail}) \times n_{NO3soil} \times dem_{Nplant}$$    if $f(n_{minavail})$ x $n_{NO3soil} \leq dem_{Nplant}$ x $r_{NO3}$

and where

4.3    $$dem_{Nplant} = \sum \frac{c_{a \to i} - c_{i \to atm}}{CN_{iMIN}}$$

520    $f(n_{NO3avail})$ = fraction of soil NO$_3$ available for plant uptake (see response functions Table A.1 (d) ), n$_{NO3soil}$ = soil NO$_3$-N content (g N m$^{-2}$), dem$_{Nplant}$ = plant N demand (g N m$^{-2}$ d$^{-1}$), r$_{NO3}$ = fraction of soil NO$_3$-N in total mineral soil N, c$_{a \to i}$ = plant C gain ( g C m$^{-2}$ d$^{-1}$), c$_{i \to atm}$ = respiration of respective plant compartment i (g C m$^{-2}$ d$^{-1}$), CN$_{iMIN}$ = defined minimum C:N ratio of each plant compartment i.

Plant organic N uptake (g N m$^{-2}$ d$^{-1}$) from the humus layer:

525    4.4    $$n_{hum \to plant} = dem_{Nplant} \times r_{hum}$$    if $f(n_{humavail})$ x n$_{humsoil} \geq dem_{Nplant}$ x $r_{hum}$

4.5 $\quad n_{hum \to plant} = f(n_{humavail}) \times n_{humsoil}$ $\qquad$ if $f(n_{humavail})$ x $n_{humsoil} < dem_{Nplant}$ x $r_{hum}$

$f(n_{humavail})$= response function for plant available N from the humus layer, $n_{humsoil}$ = soil N content in humus layer (g N m$^{-2}$).

530 **Table A.1 (b) Functions describing processes related to fungal growth and N exchange to plant**

Fungal maximum C supply (g C m$^{-2}$ d$^{-1}$):

5.1 $\quad c_{a \to fungi} = c_{a \to root} \times FRAC_{FMAX} \times f(c_{fungiavail})$

Fungal actual growth (g C m$^{-2}$ d$^{-1}$):

5.2 $\quad c_{a \to fungi} = ((c_{frt} \times FRAC_{OPT}) - c_{fungi}) \times f(n_{supply})$

535 $\qquad c_{a \to root}$ = C available for root and mycorrhiza growth (g C m$^{-2}$ d$^{-1}$), $FRAC_{FMAX}$ = maximum fraction of total root and mycorrhiza available C which is available for ECM, $f(c_{fungiavail})$ = response function which relates fungal growth to N availability, $c_{frt}$ = total root C content (g C m$^{-2}$), $FRAC_{OPT}$ = optimum ratio between root and fungal C content, $c_{fungi}$ = total ECM C content (g C m$^{-2}$), $f(n_{supply})$ = response function of fungal growth to the amount of N which is transferred from ECM to plant.

540 Minimum fungal C supply (g C m$^{-2}$ d$^{-1}$):

5.3 $\quad c_{a \to fungi} = c_{fungi \to atm}$ $\qquad$ if $c_{a \to root} \le 0$

Total fungal respiration (g C m$^{-2}$ d$^{-1}$):

6.1 $\quad c_{fungi \to atm} = c_{mfungi \to a} + c_{gfungi \to a}$

545 $\qquad$ where $c_{mfungi \to a}$ = fungal maintenance respiration and $c_{gfungi \to a}$ = fungal growth respiration (all in g C m$^{-2}$ d$^{-1}$).

Fungal maintenance respiration (g C m$^{-2}$ d$^{-1}$):

6.2 $\quad c_{mfungi \to a} = c_{fungi} \times K_{RM} \times f(T_l)$

$\qquad c_{fungi}$ = total ECM C content (g C m$^{-2}$), $K_{RM}$ = maintenance respiration coefficient, $f(T_l)$ = temperature
550 $\qquad$ response function.

Fungal growth respiration (g C m$^{-2}$ d$^{-1}$):

6.3 $\quad c_{gfungi \to a} = c_{a \to fungi} \times K_{RG}$

$\qquad c_{a \to fungi}$ = fungal growth (g C m$^{-2}$d$^{-1}$), $K_{RG}$ = growth respiration coefficient.

555 Fungal C and N litter production ($c_{fungi \to lit}$: g C m$^{-2}$ d$^{-1}$, $n_{fungi \to lit}$: g N m$^{-2}$ d$^{-1}$):

$\qquad$ If fungal growth = simple

7.1 $\quad c_{fungi \to lit} = c_{fungi} \times L$

7.2 $\quad n_{fungi \to lit} = n_{fungi} \times L - nret_{fungi}$

7.3 $\quad nret_{fungi} = n_{fungi} \times L \times (1 - N_{RET})$

560      $c_{fungi}$ = ECM C content (g C m$^{-2}$), $n_{fungi}$ = fungal N content (g N m$^{-2}$), $L$ = litter rate, $nret_{fungi}$: fungal N which is retained in fungal tissue, $N_{RET}$ = fraction of N retained in fungal tissue from senescence.

If fungal growth = detailed

7.4  $c_{fungi \rightarrow lit} = c_{fungi} \times (FRAC_{MYC} \times L_{MYC} + ((1 - FRAC_{MYC}) \times L_M))$

7.5  $n_{fungi \rightarrow lit} = n_{fungi} \times (FRAC_{MYC} \times L_{MYC} + ((1 - FRAC_{MYC}) \times L_M)) - nret_{fungi}$

565 7.6  FRAC$_{MYC}$ = fraction of mycorrhizal hyphae in total fungal biomass, L$_{MYC}$ = litter rate of mycorrhizal hyphae, L$_M$= litter rate of fungal mantle tissue.

Fungal biomass (g C m$^{-2}$, g N m$^{-2}$)

8.1  $c_{fungi} = c_{a \rightarrow fungi} - c_{fungi \rightarrow litter} - c_{fungi \rightarrow a}$

570 8.2  $n_{fungi} = n_{N \rightarrow fungi} - n_{fungi \rightarrow litter} - n_{fungi \rightarrow plant}$

Mycorrhization degree

9   $m = \dfrac{c_{fungi}}{c_{frt} \times FRAC_{OPT} \times M_{OPT}}$

$c_{frt}$ = fine root biomass (g C m$^{-2}$), $FRAC_{OPT}$ = coefficient defining optimum ratio between fungal and fine

575   root biomass, $M_{OPT}$ = optimum mycorrhization degree.

Uptake and transfer processes of ECM and plant

N transfer from ECM to plant (g N m$^{-2}$ d$^{-1}$)

10.1  $n_{fungi \rightarrow plant} = dem_{Nplant}$         if dem$_{Nplant}$ ≤ n$_{fungiavail}$

580  10.1  $n_{fungi \rightarrow plant} = n_{fungiavail}$         if dem$_{Nplant}$ > n$_{fungiavail}$

$dem_{Nplant}$ = plant N demand, $n_{fungiavail}$ = fungal available N for transfer to plant (all g N m$^{-2}$ d$^{-1}$)

10.2  $n_{fungiavail} = n_{fungi} - \dfrac{c_{fungi}}{CN_{FMAX}}$

$c_{fungi}$ = ECM biomass (g C m$^{-2}$), $CN_{FMAX}$ = maximum C:N ratio of fungal tissue, which allows N transfer to plant.

585

Fungal nitrate and ammonium uptake (given for nitrate, equivalent for ammonium with ammonium specific parameter)

11.1  $n_{NO3 \rightarrow fungi} = n_{NO3pot \rightarrow fungi} \times r_{NO3} \times f(n_{demfungi})$   if $N_{NO3pot \rightarrow fungi}$ < $n_{NO3soil}$ x f($n_{avfungi}$)

11.2  $n_{NO3 \rightarrow fungi} = n_{NO3soil} \times f(n_{avfungi})$      if $N_{NO3pot \rightarrow fungi}$ > $n_{NO3soil}$ x f($n_{avfungi}$)

590 11.3  $n_{NO3pot \rightarrow fungi} = NO3_{RATE} \times c_{fungi} \times FRAC_{MYC}$

$n_{NO3pot \to fungi}$ = potential ECM nitrate uptake (g N m$^{-2}$ d$^{-1}$), $r_N$ = fraction of ammonium-N and total mineral-N in the soil, $f(n_{demfungi})$ = N uptake response to N demand, $n_{NO3soil}$ = soil nitrate content (g N m$^{-2}$), $f(n_{avfungi})$ = N uptake response to soil availability, NO3$_{RATE}$ = nitrate specific uptake rate (g N m$^{-2}$ d$^{-1}$), $c_{fungi}$ = fungal biomass (g C m$^{-2}$), FRAC$_{MYC}$ = fraction of mycorrhizal mycelia in total fungal biomass.

595    Fungal organic N uptake from litter and humus (given for litter, equivalent for humus with humus specific parameter)

11.4    $n_{lit \to fungi} = n_{litpot \to fungi} \times r_{lit} \times f(n_{demfungi})$    if n$_{litpot \to fungi}$ x $r_{lit}$ < $n_{litsoil}$ x $f(n_{litavfungi})$x$r_{lit}$

11.5    $n_{lit \to fungi} = n_{litsoil} \times f(n_{litavfungi}) \times r_{lit}$    if n$_{litpot \to fungi}$ x $r_{lit}$> $n_{litsoil}$ x $f(n_{litavfungi})$ x$r_{lit}$

11.6    $n_{litpot \to fungi} = LIT_{RATE} \times c_{fungi} \times FRAC_{MYC}$

600    where n$_{litpot \to fungi}$ = potential ECM organic N uptake from litter (g N m$^{-2}$ d$^{-1}$), $r_{lit}$ = fraction of litter-N in total organic-N in the soil, $f(n_{demfungi})$ = N uptake response to N demand, $n_{litsoil}$ = soil litter content (g N m$^{-2}$), NLIT$_{RATE}$ = litter specific uptake rate (g N g C$^{-1}$ d$^{-1}$), $c_{fungi}$ = fungal biomass (g C m$^{-2}$), FRAC$_{MYC}$ = fraction of mycorrhizal mycelia in total fungal biomass.

605    **Table A1 (c) Overview of response functions of plant and fungal growth and N uptake**

| No. | Equation | | |
|---|---|---|---|
| Plant response to air temperature | | | |

$$f(T_l) = \begin{cases} 0 & T_1 < P_{mn} \\ (T_1 - p_{mn}) / (p_{O1} - p_{mn}) & p_{mn} \le T_1 \le p_{O1} \\ 1 & p_{O1} < T_1 < p_{O2} \\ 1- (T_1 - p_{O2}) / (p_{mx} - P_{O2}) & p_{O2} < T_1 < p_{mx} \\ 0 & T_1 > p_{mx} \end{cases}$$

(610, 12)

where $T_1$ = leaf temperature (°C) and P$_{MN}$ (-4°C), PO1 (10°C), PO$_2$ (25°C), P$_{MX}$ (40°C) are coefficients.

615    Photosynthetic response to leaf C/N ratio

13    $$f(CN_l) = \begin{cases} 1 & CN_1 < p_{CNOPT} \\ 1 + \left( \dfrac{cn_l - p_{CNOPT}}{p_{COPT} - p_{CNTH}} \right) & p_{CNTH} \le CN_1 \ge p_{CNOPT} \\ 0 & CN_l > p_{CNTH} \end{cases}$$

[revised manuscript text omitted]

---

## Author Response (AR1)

Answers to the reviewers

First we are very glad over your positive response, both reviewers agree on the legitimacy of our study and think this addressing a very important but so far neglected parts in ecosystem modeling.

The intention and aim of our study was to close the knowledge gap of missing ectomycorrhizal fungi (ECM) in current ecosystem models and compare three modeling approaches of different complexity at explaining plant and soil development across a climate and N deposition gradient. Both reviewers thought the language of the paper was weak. We feel sorry for the grammatical and typographical errors in the previous submission. We have now rewritten the manuscript thoroughly and it has been further edited thought professional language edition by native English speaking person who also has a PhD degree in relevant field, and hope you find this much better.

We further invite associated professor, Annemarie Reurslag Gärdenäs (annemieke.gardenas@bioenv.gu.se) who is specialized at ECM, soil microbes and soil biogeochemical model development as an external reviewer for this paper. She has both detailed comments on the content but also the language (her review reports attached). In the revised version of this paper, we incorporated all of her comments.

Besides, we have now rewritten the description of Bayesian calibration procedure and added more references to make it comprehensive and easier to understand. However, we would like to emphasize that the paper is to present a new model considering ECM and further compare this to two simpler approaches on explaining the observed data. We employed Bayesian calibration as a common procedure to estimate the parameter uncertainties associated with the 3 different models. Therefore the purpose is surely not only demonstrating how reduction of statistical uncertainty by Bayesian calibration can be made. Also other statistical methods can be used to demonstrate the same phenomena with respect to the link between parameter uncertainty and model structure uncertainty, providing the same data are used to constrain the model. Of course we agree with the reviewer 2 that a thoroughly and detailed description of what has been used is needed and is now added.

Last but not least, we have thoroughly improved all the figures and tables to make it easier to follow. We added Table 1 for better model comparison. More importantly, we changed previous Table 4 and Figure 5 into a new Figure 5 to compare our modelling results better with the measured data. Previously only soil C balance was compared, we have now added all the major C cycling variables: GPP, ecosystem respiration, soil respiration, NEE, and also soil N balance (see new Fig. 5). This gives a much more comprehensive comparison of the modeling approaches and data, which also additionally show large difference of litter addition and soil respiration between the "explicit"/"implicit" approach. These are all additionally included in the result section. We also have made a more thoroughly discussions on the modeling approaches, uncertainties and possible explanations. The abstract and conclusions are also improved. The information-rich parameter correlation tables (Tables A2, A3 and A4.) are further moved into supplementary to make the paper more concise and easier to read.

We will here answer the comments raised by the Reviewers, and how these were met by changes in the text and figures (answers are marked with blue).

Comments by Reviewer 1

**Specific comments**

Introduction Consider putting in a table showing clearly what the different models described on line 48 thourgh 69 do.

Table 1 is now added to make it clear and also added Coup-MYCOFON according to external reviewer's comment

Line 128 ECM growth is driven by sink strength of what?

Now rewrite the whole section to make it clear.

Line 141 I had to read this sentence twice as I thought the authors were comparing the approach for ECM and root respiration to the approach of something else. however I think they have just treated ECM respiration the same way they have treated root respiration. Perhaps it would be clearer to say that there are two components (maintenance and growth) for both ECM and root respiration.

Changed accordingly.

Line 159 Is NUPT$_{FRACMAX}$ the fraction of total soil N available for uptake, or is it the fraction of mineral N available for uptake? Please clarify.

Clarified. NUPT$_{FRACMAX}$ determines the fraction of mineral N available for uptake.

section 2.1.5 My first reaction was that degree of mycorrhization had not been taken into account; then I realised mycorrhization degree was covered in section 2.1.6. Consider switching these two sections.

We agree and switched accordingly

Line 166 Please add the scientific name for spruce. As this is Sweden it is probably Picea abies.

Added

Line 211 I see the point of spinning up the vegetation from the time of establishment over the lifetime of the trees (100 years in this study), but soil C pools may take considerably longer than that to come to equilibrium. For example, 500 years is a more typical spinup to initialise soil C pools in dynamic vegetation models (DVMs). The legacy of recalcitrant C from previous forest growth in the soil must be accounted for by the initial standing C stock and C/N initial values in Table 2 which the footnote says are calibration parameters; maybe make this clear in the text. Unlike the calibration parameters of Table 1, the initial values assumed for soil C pools shown in Table 2 do not have minimum and maximum values associated with them, and standing stock does not appear in Table 3.

First, we have now redesigned the Table 3 to show the calibration data and forcing data clearly. The model is constrained by the forest standing biomass and the soil C/N ratio, not the soil C or N pools as such. Previous 100-year simulation by Svensson et al. (2008a) considering the regions and similarly by Berggren Klejal et al. (2007) considering specific representative sites for the same regions, both showed that the model can describe a consistent pattern of C pools and C/N ratios. None of the regions are in a perfect steady state but the difference from a steady state is small and not possible to constrain from measured changes of the soil C pools. Thus adding soil C would not further constrain the parameters. Moreover, Svensson et al. (2008a) demonstrated that the soil C/N ratio showed consistent patterns of different N supply assumptions and expected turnover rate from differences in climate forcing and current C-pools. These previous applications thus provide a base for our current model designs and evaluations. We have described this in more detail and also added why these observational constraints (i.e. soil C/N) are selected in section 2.3.1.

However, we agree that initialization problem of the soil pools exist but this is mostly a general problem independent of the three model approaches. Ideally, the initialization of each soil organic C pool required a spin-up simulation over a longer-term (e.g. 500 years) to find a soil C equilibrium for undisturbed vegetation. After the spin-up using undisturbed vegetation, the reconstructed disturbance history was then used to get a close estimate of the SOC pools. But this two-step method requires informative historical data which are not available in our case. Besides, another significant uncertainty in this spin-up type runs is the initial estimate (500 yrs ago) of inert or very slowly decomposing organic C, which again we do not know.

Thus in our study, we use another approach by following Svensson et al., (2008) and Berggren Klejal et al., (2007) who use the measured current soil C and N content data from similar soils that only different with respect to climate. And they assume that the current soil was in close to equilibrium with respect to C/N ratios for the different regions. Eliasson et al. (2013) investigated the soil balance in Swedish forests over 300 years by different modeling approaches and also found the soil C balance generally reach equilibrium after 100 years. It should be noted that the intention was to evaluate how the ECM affect the C and N cycling in plant-soil over the lifetime of the trees in different regions providing basic assumptions on the carbon pools. Our investigation cannot be used to justify some new suggestion on the current rate of change of soil carbon pools in the different regions.

Line 212 I do not understand what is meant by this sentence: A minimum of specific regional data were used at input values. Does this refer to the number of driving variables input to the model (six in Table 2 plus two calibration parameters) or the amount of data used in the Bayesian analysis for each driving variable (30-year averages rather than time series or multiple values for each region)? I also don't understand at input values; does this mean as input values or does it mean something else? What is specific about the regional data?

We rewrite this to make it clear

Line 230 The data likelihood function which determines the parameter sets **being candidate of** the posterior distribution sounds odd; I assume that this sentence refers to the likelihood function determining **acceptance** of the parameter sets which will comprise the posterior distribution?

Changed accordingly

Line 235 Please make clear that $\omega_i$ is a vector.

Changed accordingly

Line 244 Replace $q_i + 1 = q_i + \varepsilon$ " with $\theta_i + 1 = \theta_i + \varepsilon$, using the same _ on lines 244 (the equation) and 245. Also, consider numbering the equations.

Changed accordingly

Line 280 Surely it is just parameters that are being calibrated and not processes?

Removed "processes"

Line 306 Do fungi take up the same amount of organic N when there is sufficient
mineral N available?

No, the uptake is both driven by demand and by N availability, and in our case, mineral N will first
regulate the mineral uptake and reduce the demand of organic N. This is describe in detail in the 2.1.5

Line 333 should thus be and? Is the sentence referring to N mineralisation? A higher
organic matter turnover should mean higher N mineralisation.

Changed accordingly

Section 3.2.3 Is it necessary to list all these correlations? The figures are better for
this; perhaps only discuss the most interesting ones?

We have now moved this detailed correlation tables into supplementary files, Fig. 9 show the
important and interesting ones.

Section 4 I take the authors' point that there is a dearth of comparison data, especially
related to ECM, but are there really zero data? is there not one observation that
can be compared with the model results? What about the Lindroth et al paper
cited on line 419? How does the coupling of Mycofon to CoupModel affect the
simulated soil respiration, for example? it is a bit difficult to claim that the model
delivers "accurate" results (line 464) without any comparison to observations.
Table 4 shows the Svensson et al model results so consistency with this other
model could be worth showing in a figure.

We have now added a new figure 5 to make this clear. The simulated regions do not have detailed
measured data. In previous papers the carbon balance have been compared with the data of eddy flux
measurements from some few years of each site (Svensson et al., 2008a, b). We compare the nearby
sites that have been intensively measured by eddy covariant technology. We both compare the soil
respiration, total respiration, GPP, NEE and also change of soil C and soil N.

**2 General remarks on figures**

Please include units and self-explanatory axes labels in all figures. Many readers will
look at the abstract and figures before deciding to read the text; don't make readers
go searching through the text for basic information. Where possible, don't even make
readers read the captions carefully. In general, don't make readers do more work than
absolutely necessary to understand what is being shown in the figures.

We have now redesign the figures and take all of these into consideration

Figure 3 Is total N litter production the N released during decomposition or the N being
added to the litter pool with fresh litter?

N total litter production is the total N in litter being added to the soil litter pool by fresh litter, we have
added this in the caption to make it clear

Figure 4 There is room to add implicit model and explicit model to the right of the figure so that readers can see immediately what the upper and lower graphs mean.

Added accordingly

Figure 5 Is GPP in this figure simulated or measured? Any possibility of showing both simulated and measured GPP?

This is modelled GPP and now the new Figure 5 also include measured GPP data from nearby sites with similar conditions which are comparable to our study. We further also compare our results to Svensson et al., (2008a)

Figure 6 There is room to add nonlim, implicit and explicit to the right of the three panels, and to show the north-south gradient to the left of the Y axis of each panel.

Added accordingly

Figure 7 Show the N-S gradient to the left of the panels (ie N next to Ly, and an arrow leading to S next to Lj). Thanks for adding implicit and explicit; please also add the meanings of the parameters on the X axis (eg $K_H$ is the humus decomp. coeff.) so that readers can see at a glance what is going on without having to search the text and tables.

Added accordingly

Figure 8 Please give the units, especially for the rates. What is fungal litter rate, the rate of uptake from litter, or the rate at which hyphae die and contribute to the litter pool?

Added now and the Fungal litter rate refers to the rate at which hyphae die and add to the soil litter pool, we have added this in caption to make it clear

Figure 9 Does C assimilates mean NPP? Please make clear what parameters are being shown, so readers don't have to go searching (they probably won't have read the paper and won't realise the information is in one of the tables). Is the colour scheme here the same as in previous figures?

C assimilates refers to the C taken up by the plant, so GPP, although of course respiratory losses and litter are subtracted later so that only net growth remains. We have now added this in figure caption and also show the meaning of the parameters briefly in the figure to make the reader easier to follow. The color scheme are the same in all the figures. We rephrase the figure caption to make this clear.

**3 Tables**

Table 2 Can it be made clearer that soil C/N and standing stock of C are calibration parameters and the other data are all driving data?

Now redesigned to make it clearer

Table 3 Why are there no mean and uncertainty columns for soil C standing stock?

according to Table 2 it's a calibration parameter.

You misunderstood here and only soil C/N is a calibration parameter, not soil C stock. See above and also details are added in section 2.3.1

Table 4 The Lindroth et al data shown here are means of the highest and lowest estimates, but the full ranges are shown for the Mycofon results. Would it not be better to show ranges for both?

We have added these and integrate previous figure 5 and Table 4 into the new Figure 5 which show more detailed ecosystem C processes, model data comparison.

Table 4 The Svensson et al data generally fall within the Mycofon model ranges; are these the ranges from the posterior distributions? why is the implicit approach shown for one site and the explicit approach for the other, and what are the results for the mean of the posterior? Could this material (Svensson et al. vs. model approaches) be presented as a figure? If Lindroth et al. measured respiration, surely that is a CoupModel output which could be compared to those measurements?

We have included a new Figure 5 which includes the measured data from Lindroth et al. (2008) also Svensson et al. (2008a). However it should be noted that Svensson et al. (2008a) data are results from single model simulation established by subjective calibration and not from an ensemble approach, like in our study.

**4 Grammatical or typographical errors**

Here is a partial list of lines with errors, including suggested corrections.
In some cases I suggest rewordings of awkward clauses, in others I try to show the grammatical/typgraphical error and how to fix it. Original text is to the left of the arrow, and the replacement text to the right of the arrow. Actual changes (deletions to the left of the arrow, additions to the right of the arrow) are in **boldface**. I have tried to include enough text to make it clear why the change is necessary, such as where a grammatically plural noun is coupled with a grammatically singular verb.
Generally, models and approaches are preceded by the, which is omitted repeatedly throughout the text. A global change is not possible because there are a few occasions where the is present, or where it is OK to leave it out.

We have now rewritten the paper thoroughly and include all the following into revision. The language was also improved by British language editing companies. We believe the language of paper is now significantly improved.

27 ... Coup-Mycofon model provide ! Coup-Mycofon model provide**s**
43 **known as** ! **which are**
46 **the** ecosystem ! ecosystem
48 research show ! research show**s**
56 Moore ! **the** Moore
60 ANAFORE ! **the** ANAFORE
68 ECM models ... simulate**s** ! ECM models ... simulate
70 that coupled ! that **is** coupled
78 approach which ! approach**es** which
79 The "ECM implicit" does not ! The "ECM implicit" **approach** does not
79 incorporat**ing** ! incorporat**es**
80 Plants ... **does** not ! Plants ... **do** not
100 **in** Meyer ! **by** Meyer (NB this is my personal preference but check the journal's

policy: are citations considered to be the name of the paper, in which case in is fine, or do they refer to the authors who wrote the paper, in which case by makes more sense?)

315 118 **are distinguished** between ! **distinguish** between

131 follow ! follow**s**

132 **to prevent fungi to die** ! **preventing fungal death**

159 ... as fungi **have** are more efficient ! as fungi are more efficient

193 plant **uptaking of** organic N ! plant **taking up** organic N

320 203,213 **Tab.** 2!**Table** 2 (likewise Table 3 in section 3.1.1; check the journal's policy, but in any case be consistent as Table is spelled out earlier in the manuscript)

205 management**s** ! management

206, see 100 in Svensson ! by Svensson

212 effects **is** not ! effects **are** not

325 237 both**,** the ! both the

239 **for a better constrain of** posterior ! **to better constrain** posterior

241 using the **Markov chain Monte Carlo method, also the Metropolis-Hastings walk** ! using the **Metropolis-Hastings random walk Markov Chain Monte Carlo algorithm** (and please cite van Oijen et al 2005 here too)

330 245 The **random numbers are generated normally distributed having** a mean of zero ! The **normally distributed random numbers _ have** a mean of zero

252 **parameter: ConstantNsupply for the spruce tree, is selected as calibration parameters** ! **parameter ConstantNsupply for the spruce tree is a calibration parameter**

335 257 (C/Nmyc)**,** ! (C/Nmyc)

268 The posterior model ... show ! The posterior model ... show**s**

272 than that **of** using the "implicit" **and** "explicit" approach ! than that using the "implicit" **or** "explicit" approach

275 generally **N more** limited ! generally **more N** limited

340 277 southern site, Ljungbyhed than ! southern site, Ljungbyhed**,** than

278 **show overestimation** by "implicit" approach but **change to underestimation** when "explicit" approach is used ! **is overestimated** by **the** "implicit" approach but **colorredunderestimated** when **the** "explicit" approach is used

281 **the more processes and parameters included for calibration, less likely of**
345 **finding an accept combination of parameter sets** ! **as more parameters are included for calibration, acceptable combinations of parameter sets become less likely**

286 approach **show a much larger uncertainties than that of ECM "implicit" and** "explicit" approaches ! approach **shows much larger uncertainties than either**
350 **the "implicit" or** "explicit" approaches

287 approach **simulate soil sequestration of N up to 2 g N** m$_2$ y$_1$ ! approach **simulates up to 2 g soil N** m$_2$ y$_1$

292 **Besides the simulated soil C balance by** "nonlim" approach ! **The simulated soil C balance by the** "nonlim" approach

355 293 the soil **sequestrate C at most north site, Lycksele** but ! the soil **sequesters C at the most northerly site, Lycksele,** but

294 and decoupled ! and **are** decoupled

297 and "implicit" approach ! and the "implicit" approach

297 **sites overall loss soil C by 6 and 5 g C m**$_2$ **y**$_1$ ! **soils lose 6 and 5 g C m**$_2$
360 **y**$_1$**, respectively**

298 **sites gain soil C by 3 and 13 g C m**$_2$ **y**$_1$ ! **soils gain 3 and 13 g C m**$_2$ **y**$_1$**, respectively**

299 For "explicit" approach ! For **the** "explicit" approach

300 in "implicit" approach ! in **the** "implicit" approach

365 301 show **an** overall minor C and N losses ! show overall minor C and N losses

305,306 in "explicit" model ! in **the** "explicit" model

309 using "implicit" approaches ! using the "implicit" approaches
310 favour climate ! favour**able** climate
311 but "explicit" approach show a ! but **the** "explicit" approach show**s** a
370 312 in "explicit" approach ! in the "explicit" approach
314 **show explicitly account** for ECM!**shows that explicitly accounting** for ECM
326 except **a** larger uncertainties in the "explicit" . ! except **for** larger uncertainties
in the "explicit" **approach**.
327 than **that of** the southern ! than **for** the southern
375 The rest of the manuscript is riddled with errors like the ones above; please go through
and fix them.

380 Anonymous Referee #2

The authors coupled an ectomycorrhizal fungi (ECM) model MYCOFON with a terrestrial
biogeochemistry model to show that it is important to consider the plant-ECM
interaction to properly model the ecosystem nitrogen dynamics. While I could see the
385 legitimacy of their statement, I agree with the other reviewer that the paper seems submitted
in a hurry: there are too many problems with grammars, syntaxes and formats,
making it unreadable to some extent. Thus a thorough rewritten is needed before it
can be better judged.

390 The paper has been rewritten thoroughly. The language was also edited by British language edition
services. Please also see answers above.

The language problem becomes more severe as the paper goes closer to the end. For
instance, the description of 2.3.2 is pretty much a mess. I guess it is really awkward that
395 a paper would use "Bayesian calibration procedure" as section title. Personally, I think
"Bayesian calibration of models" would be much more appropriate. The use of "data
likelihood function" is also not accordant with the general terminology in data assimilation
or Bayesian inference based model calibration. I strongly suggest the authors to
read more relevant papers and revise the description to make it more readable.
400 As for the description of MCMC method, there are many excellent papers on this topic,
however, the authors barely mention them and the description is again very poor.
As the paper reaches the results section, there are many more language/presentation
problems. Many of the sentences are incomplete, such as missing verbs or wrong use
of juxtapositions. The other reviewer has listed many of those issues and I won't add
405 more to the list.

We have now rewritten the entire description of Bayesian calibration (section 2.3) and added more
references to make it comprehensive and easier to understand. Specifically, we describe the
observational constraints more clearly and justify the reason for select plant biomass and soil C/N as
410 the accepted criteria. We add more text explaining the defined measured error. The sub-sections:
parameters chosen for calibration and Bayesian calibration of models are switched in position in order
to be followed easily by the reader. Besides, we have described the MCMC algorithm and Bayesian
method in much more detail, also more literature for comparison.

415 Again, we would like to emphasize that the paper is to present a new model considering ECM and
further compare this to two previous simpler approaches of explaining the observed data. We employed
Bayesian calibration to estimate the parameter uncertainties but the purpose is surely not only
demonstrating the reduction of statistical uncertainty by Bayesian Calibration as such.

420

Further, I don't know why Appendix is shown in the middle of the paper. Have the authors carefully checked their submission? Is the wrong version uploaded? Overall I suggest rejecting the paper for a resubmission.

We do agree that the long appendix might reduce the readability and also consider comments from reviewer 1 that parameter correlation tables (Table A2, A3 and A4) might not needed in that detail. So we now move those into supplementary and substantially reduce the appendix. However, the equations and explanation of the parameters can be helpful for the reader to get into details of the model buildup.

Our paper overall presents a new ecosystem model (version) that can explicitly include ECM, where so far the other models cannot. Modeling comparison also clearly demonstrate the importance and legitimacy of incorporating ECM into ecosystem models. Of this, both reviewers agree on. Again we are sorry for the language issue but we believe the language has been largely improved in the revised version.

As explained in https://www.geoscientific-model-development.net/about/manuscript_ types.html GMD is expecting that authors upload the program code of models and the used data sets as a supplement or make the code and data available at a data repository preferable with an associated DOI (digital object identifier) for the exact model version described in the paper. If for some reason your code and/or data for the MYCOFON model cannot be made available in this form as the code availability section in your paper suggests you need to state the reasons why the code is not available or why access is restricted. Please note that in the code accessibility section you can still point the reader to your web site for updates even if you provide the code as supplement or use a DOI for a release.
All the best Lutz Gross GMD Executive Editor

Dear editor,
Thanks for the comment, we will now add the model software and version. Mycofon-CoupModel is derived from CoupModel with the implementation presented in section 2 here. The general code of CoupModel will be made available from www.coupmodel.com, which means freely available for everyone after registration. The general information about how to install CoupModel and its different branches and tutorials are also made available from www.coupmodel.com. The manuscript has been updated with these information and the new version used would also be specified in the section code availability.

475

**References mentioned**

Svensson, M., Jansson, P. E. and Kleja, D. B.: Modelling soil C sequestration in spruce forest ecosystems along a Swedish transect based on current conditions, Biogeochemistry, 89, 95–119,
480    2008a.

Berggren Kleja, D., Svensson, M., Majdi, H., Jansson, P.E., Langvall, O., Bergkvist, B., Johansson, M.-B., Weslien, P., Truusb, L., Lindroth, A.,  Agren, G.: Pools and fluxes of carbon in three Norway spruce ecosystems along a climatic gradient in Sweden, Biogeochemistry, 89:7–25, 2008.
485

Eliasson et al. (2013) forest carbon balances at the landscape scale investigated with the Q model and the CoupModel responses to intensified harvests, Forest ecology and management, 290, 67-78.

Svensson, M., Jansson, P. E., Gustafsson, D., Kleja, D. B., Langvall, O., Lindroth, A., Bayesian
490    calibration of a model describing carbon, water and heat fluxes for a Swedish boreal forest stand, Ecological Modelling, 213, 331-344, 2008b.

495

**Simulating ectomycorrhiza in boreal forests: implementing ectomycorrhizal fungi model MYCOFON into CoupModel (V5)**

Hongxing He[1], Astrid Meyer[1,a], Per-Erik Jansson[2], Magnus Svensson[2], Tobias Rütting[1], Leif Klemedtsson[1]

[1] Department of Earth Sciences, University of Gothenburg, Po Box 460, Gothenburg 40530, Sweden

[2] Department of Land and Water Resources Engineering, Royal Institute of Technology (KTH), Brinellvägen 28, Stockholm 100 44, Sweden

[a] now at: Institute of Groundwater Ecology, Helmholtz Zentrum München, Ingolstädter Landstraße 1, Neuherberg 85764, Germany

*Correspondence to:* Hongxing He (hongxing.he@gu.se)

**Abstract**

The symbiosis between plants and Ectomycorrhizal fungi (ECM) are shown to considerably influence the carbon (C) and nitrogen (N) fluxes between the soil, rhizosphere and plants in boreal forest ecosystems. However, ECM are either neglected or presented as an implicit, non-dynamic term in most ecosystem models which can potentially reduce the predictive power of models.

In order to investigate the necessity of an explicit consideration of ECM in ecosystem models, we implement the previous developed MYCOFON model into a detail process-based, soil-plant-atmosphere model, Coup. MYCOFON, which explicitly describes the C and N fluxes between ECM and roots. This new Coup-MYCOFON model approach (ECM explicit) is compared with two simpler model approaches; one containing ECM implicitly as a dynamic uptake of organic N considering the plant roots to represent the ECM (ECM implicit), and the other a static N approach where plant growth is limited to a fixed N level (nonlim). Parameter uncertainties are quantified using Bayesian calibration where the model outputs are constrained to current forest growth and soil C/N ratio for four forest sites along a climate and N deposition gradient in Sweden and simulated over a 100-year period.

The "nonlim" approach could not describe the soil C/N ratio, due to largely overestimation of soil N sequestration but simulate the forest growth reasonably well. The ECM "implicit"/ "explicit" approaches are both able to describe the soil C/N ratio well but slightly underestimate the forest growth. The "implicit" approach simulated lower litter production and soil respiration than the "explicit" approach. The ECM "explicit" Coup-Mycofon model provides a more detailed description of internal ecosystems fluxes and feedbacks of C and N between plants, soil and ECM. Our modelling highlights the need to incorporate ECM and organic N uptake into ecosystem models, and the "nonlim" approach is not recommended for future long-term soil C and N predictions. We also provide a key set of posterior fungal parameters which can be further investigated and evaluated in future ECM studies.

**1. Introduction**

Boreal forests cover large areas on the Earth's surface and are generally considered as substantial carbon (C) sinks (Dixon et al., 1994; Pan et al., 2011). The sink strength is determined through the balance between major C uptake and release processes, i.e., plant photosynthesis and both autotrophic and heterotrophic respiration, and is largely controlled by nitrogen (N) availability (Magnani et al., 2007; Högberg et al., 2017). Numerous studies have shown that soil N availability is the main driver for plant and microbial dynamics (Vitousek and Howarth, 1991; Klemedtsson et al., 2005; Lindroth et al., 2008; Luo et al., 2012; Mäkiranta et al., 2007; Martikainen et al., 1995). Thus, a proper description of N dynamics in ecosystem models is prerequisite for precisely simulating plant-soil C dynamics and greenhouse gas (GHG) balance (Maljanen et al., 2010; Schulze et al., 2009; Huang et al., 2011). Ecosystem models, however, vary considerably in their representation of N fluxes: from very simplified presentations (e.g., the LPJguess model: Sitch et al., 2003; Smith et al., 2011) to very complex approaches which aim to capture the whole N cycle (e.g., LandscapeDNDC: Haas et al., 2012; CoupModel: Jansson and Karlberg, 2011).

Ectomycorrhizal fungi (ECM) are common symbionts of trees in boreal forests. ECM are more efficient than roots in taking up different N sources from the soil (Plassard et al., 1991), as well as store vast amounts of N in their tissues (Bååth and Söderström, 1979) and can cover a large fraction of their host plants' N demand (Leake, 2007; van der Heijden et al., 2008). Further, ECM are shown to respond sensitively to ecosystem N availability and are generally considered as adaptation measures to limited N conditions (Wallenda and Kottke, 1998; Read and Perez Moreno, 2003; Kjoller et al., 2012; Bahr et al., 2013; Choma et al., 2017). Previous research shows that ECM can receive between 1 and 25% of the plants' photosynthates and constitute as much as 70% of the total soil microbial biomass, thus having a major impact on soil C sequestration in boreal forests (Staddon et al., 2003; Clemmensen et al., 2013). Overall, the functions and abundance of ECM fungi constitute numerous pathways for N turnover in the ecosystem and considerably influence the magnitude and dynamics of C and N fluxes.

Nevertheless, ECM have rarely been considered in ecosystem models (for an overview about modelling ectomycorrhizal traits see Deckmyn et al., 2014). To our knowledge, only five ecosystem models have implemented ECM to various degrees: The ANAFORE model (Deckmyn et al., 2008), the MoBiLE environment (Meyer et al., 2012), the MyScan model (Orwin et al., 2011) and more recently the Moore et al. (2015) and Baskaran et al. (2016) ECM models (Table 1). In the ANAFORE model, ECM are described as separate C and N pools. However, this model does not distinguish between mycorrhizal mycelia and mantle. The C allocated from the host tree to ECM is simulated as a zero order function, further regulated by nutrient and water availability. ECM can also facilitate organic matter decomposition in the ANAFORE model. The MyScan model uses a similar approach for ECM C uptake and dynamics but does not, to our knowledge, include the influence of water availability on ECM. In both models, ECM transfer of N to the host is regulated by the C/N ratios of the plant and fungi. In the MoBiLE model, C allocation to ECM is more complex than that in ANAFORE and MyScan models, and the N allocation to the host by the ECM can feed back into their C gains. Although, the N allocation to the host plant is described similarly to the other two models. In MoBiLE, mycorrhiza are further distinguished between mycelia and mantle, but cannot neither degrade organic matter nor take up organic N forms. Mycelia and mantle differ in their capacity to take up N, and the mantle has a slower litter production rate than that of mycelia. Both Moore et al. (2015) and Baskaran et al.'s (2016) ECM models

represent the ECM as a separate model pool and also explicitly simulate ECM decomposition, but with much more simpler process descriptions, and the interaction with environmental functions are neglected (Table 1).

The overall aim of this study is to improve understanding of ecosystem internal C and N flows related to symbiosis between ECM and host tree, in order to improve the model predictive power in assessment of C sequestration and climate change. This is done by presenting a new version of the CoupModel, that is coupled with an explicit description of ECM, and also to investigate how the explicit consideration of ECM affects the overall model performance and model uncertainty. We Thus, wSpecifically, wethus implemented the previously developed MYCOFON model (Meyer et al., 2010; Meyer et al., 2012) into the well-established soil-plant-atmosphere model, CoupModel (Jansson, 2012). The implementedWe choose the MYCOFON model because; first, it contains a very detailed description of ECM fungal C and N pools, and all major C and N ECM exchange processes (i.e., litter production, respiration, C uptake, N uptake), and second, ECM can also additionally responses to the soil N availability (Table 1). Fungal Therefore, ECM ggrowth and N uptake, both mineral and organic N forms, respond dynamically to environmental functions and plant C supply in the new Coup-MYCOFON model (Fig. 1). This detailed ECM explicit modelling approach (thereafter hereafter called "ECM explicit") is further compared with two simpler modelling approaches: , the "ECM implicit" and "nonlim" approaches , which already exist in CoupModel. The "ECM implicit" approach does not represent the ECM as a separate pool but incorporates ECM into the roots implicitly. Plants are thus allowed to take up additional organic N sources staticallyfrom soil organic pools, and do not respond to environmental functions. SimilarThe "ECM implicit" approach was usedhas been previouslyused in a similar way inby Kirschbaum and Paul, (2002) and Svensson et al. (2008a). The "nonlim" approach assumes an "open" N cycle and plant growth areis limited by a constant N availability thus to a static fixed level (e.g., in Franklin et al., (2014)). These three ECM modelling approaches represent constitute most of the current ECM representations in ecosystem models, and are tested by four forest sites situated along a climate and N fertility gradient across Sweden (Fig. 2). Bayesian calibration is used to quantify the uncertainty of model parameters and identify key parameter sets.

**2. Data and Methodologys**

**2.1 Model description**

The CoupModel ("Coupled heat and mass transfer model for soil-plant-atmosphere systems", Jansson and Karlberg, 2011) is a one-dimensional process-orientated model, simulating all the major abiotic and biotic processes (mainly C and N) in the soil-plant-atmosphere systemterrestrial ecosystem. The basic structure is a depth profile of the soil for which water and heat flows are calculated based on defined soil properties. Plants can be distinguished between understoreyunderstory and overstoreey vegetation, which allows simulating competition for light, water, and N between plants. The model is driven by measured climate data: precipitation, air temperature, relative humidity, wind speed, and global radiation and can simulate ecosystem dynamics in hourly/daily/yearly resolutions. A general structural and technical overview of the CoupModel can be found in Jansson and Moon (2001) and Jansson and Karlberg (2011), and. Aa recent overview of the model was also given by Jansson (2012). The model is freely available at www.coupmodel.com. The CoupModel (V5) was is complemented with an ectomycorrhizal module (MYCOFON, Meyer et al., 2010) which allows to directlythe direct simulate simulation ofthe the C and N uptake processes of ECM. The MYCOFON model is described in detail by Meyer et al. (2010),

and here only the key processes of plant and ECM fungal growth, N uptake as well as litterfalling and respiration are described.

**2.1.1 Plant growth in CoupModel**

An overview of model functions is given in Table A.1 in the Appendix Table A.1. Plant growth is simulated according to a "radiation use efficiency approach" where the rate of photosynthesis is assumed to be proportional to the global radiation absorbed by the canopy, but limited by temperature, water conditions, and N availability (eq. 1, Table A.1(a)). Assimilated C is allocated into five main different plant C compartments: $C_{root}$, $C_{leaf}$, $C_{stem}$, $C_{grain}$ and $C_{mobile}$. The Same compartments also represent the corresponding N amounts. The "mobile" pool ($C_{mobile}$, $N_{mobile}$) contains embedded reserves which are reallocated during certain time periods of the year, e.g., during leafing. Respiration is distinguished between maintenance and growth respiration, where a $Q_{10}$ function response was is used, respectively for maintenance respiration (eqs. 2.1, 2.2, Table A.1(a)). Plant litter is calculated as fractions of standing biomass (eq. 3, Table A.1(a)).

**2.1.2 ECM Fungal C and N pools**

The ECM are closely linked to the trees' fine roots and consist of a C and N pools. The C pool is distinguished between the mycelia, which are responsible for N uptake, and the fungal mantle, which covers the fine roots tips. The C pool is the difference between C gains by supply from the plant supply and C losses due to respiration and litter production (eq. 8.1, Table A.1(b)). Accordingly, the fungal N pool is the result of the difference between N gains by uptake, and N losses by litter production, and N transfer to the plant (eq. 8.2, Table A.1(b)). ECM fFungal C and N pools distinguish between mycelia and mantle which is of importance for when simulating N uptake (only the mycelia is able to take up N), and also when simulating litter production if thea more complex approach for simulating fungal litter production is chosen (see section 2.1.4). The ratio between mycelia and mantle is determined by the parameter $FRAC_{MYC}$ which defines the fraction of mycelia C in total ECM fungal C. For all other N and C exchange processes (growth, respiration, and N transfer to plant), the separation between mycelia and mantle is disregarded.

**2.1.3 Growth of ectomycorrhizal fungi**

ECM growth is limited by a defined maximum, ; i.e., only a certain amount of tree host assimilates will beare directed to the ECM. This maximum ECM growth is determined by a potential C supply from the plant, and limited by N availability (eq. 5.1, Table A.1 (b)). The C supply is defined by a constant fraction of the root C gain and is leveled off by the function $f(c_{fungiavail})$ as soon as a defined value of soil available total N is exceeded; i.e., in the model the potential ECM growth declines with rising soil N. This scaling function is based on observations from field and laboratory experiments, which showed that the ECM biomass of mycelia and mantle can be as much as 30-50% of fine root biomass, and the majority of ECM decreases in abundance and functioning when the soil N levels are high (e.g., Wallander, 2005; Wallenda and Kottke, 1989; Högberg et al., 2010). This is defined according to the results from field and laboratory studies that, that the ECM biomass of mycelia and mantle can be as much as 30-50% of fine root biomass. Besides, ECM growth is driven by sink strength (see overview by Smith and Read, 2008). The actual ECM growth is limited by the maximum growth and calculated by a pre-defined fraction of assimilated root C, assuming that the production of an optimum mycorrhization degree requires

[revised manuscript text omitted]
, the potential ECM uptake is first defined. This is determined by the size of ECM C pool, the fraction of ECM C which is capable of N uptake (the mycelia, $FRAC_{MYC}$), and an uptake rate ($NO3_{RATE}$, $NH4_{RATE}$, $NORG_{RATE}$ (eqs. 11.1, 11.3, 11.4, 11.6, Table A.1 (b)). This function is based on the assumption that only the ECM fungal mycelia can take up N. Values for $NO3_{RATE}$, $NH4_{RATE}$, and $NORG_{RATE}$ are derived from published values but with wide ranges (Table 2). The actual N uptake is dependent on the available soil N as well as the ECM N demand (eq. 11.2, Table A.1). The N availability function $f(n_{avfungi})$ determines the fraction of soil N which is available for ECM fungal uptake, and is controlled by the parameters $NUPT_{ORGFRACMAX}$ (the fraction of organic N available for uptake) and $NUPT_{FRACMAX}$ (the fraction of mineral N available for uptake). N availability for ECM corresponds to the plant available N (eq. 16, Table A.1), but as ECM are more efficient in the uptake of nutrients, the availability is enhanced for both mineral and organic N (eqs. 17.1, 17.2, 17.3, Table A.1). To prevent the ECM N demand being covered by only one N form, the parameters $r_{NO3}$, $r_{NH4}$, $r_{LIT}$ and $r_{HUM}$ are included, corresponding to the ratio of nitrate and ammonium in total available soil N (litter and humus). If the potential N uptake exceeds the available soil N, the actual uptake corresponds to the available N (eq. 11.2 and eq. 11.5, Table A.1 (b)).

**2.2 Transect modeling approach**

**2.2.1 Three ECM modeling approaches**

Three modeling approaches of different complexity were applied in this study. The basic "nonlim" approach was conducted to test if plant N uptake can be described as proportional to the C demand of the plants of the respective sites. In this case, the plant N uptake is not regulated by the actual soil N availability, and N is used from a virtual source potentially exceeding the soil N availability, thus as an "open" N cycle. The "ECM implicit" approach simulates plant uptake of organic N which is assumed to be via ECM, i.e. ECM are considered implicitly as being responsible for the N uptake, but are not physically represented in the model. The rate of the organic N uptake is determined by the plant N demand and restricted by the availability of organic N in the soil humus pools (eqs. 4.4, 4.5, Table A.1). Plants can also additionally take up ammonium and nitrate (eqs. 4.1, 4.2, Table A.1). In the "ECM explicit" approach, ECM fungi are fully physically considered as described above. ECM growth interacts dynamically with plant growth and responds to changes in soil N availability and soil temperature. ECM fungi can take up both mineral and organic N forms.

**2.2.2 Simulated regions and database**

Simulations were performed for four forests sites – Lycksele, Mora, Nässjö, and Lungbyhed – situated along a climate and N deposition gradient in Sweden (Fig. 2). Climate and site information is given in Table 2 and the climate data were taken from the Swedish Meteorological and Hydrological Institute (SMHI). Data on forest standing stock volumes and forest management were derived from the database and practical guidelines of the Swedish Forest Agency (2005), and applied as previously described by Svensson et al. (2008a). Soil C content as well as soil C/N ratio previously determined by Berggren Kleja et al. (2008) and Olsson et al., (2007), and used to describe soil properties in the initial model setup. For all simulated sites and modeling approaches, the development of managed Norway spruce forests was simulated in daily step over a 100-year period from a newly established to a closed mature forest. Climate input data were quadrupled in order to cover the entire period, and thus climatic warming effects are not considered here. A minimum of specific regional data including the meteorological data, N deposition and soil data were used as input values (Table 2). Otherwise, model parameters were kept identical between modeling approaches in order to evaluate the general model applicability. An overview of the parameter values is shown in Table A.1 (d) in the Appendix. For a more detailed site description and CoupModel setup, see Svensson et al. (2008a).

**2.3 Brief description of Bayesian calibration**

**2.3.1 Observational constraints**

We performed a Bayesian calibration for all modeling approaches and sites. In this study, we emphasize the models' predictability in precisely describing the long term plant and soil developments, also aiming at maximized model flexibility. This allows us to compare the different model approaches in terms of explaining the measured data, and also to investigate distributions and uncertainty of key parameters. The previous modeling study by Svensson et al. (2008a) demonstrated that the changes of soil C in these sites were rather small over a 100-year period while the soil C/N ratio showed large variabilities with different N supply assumptions. Therefore, in this study the measured C/N ratio of soil organic matter and standing stock biomass were used as observational constraints. The measured error (also called relative uncertainty in Table 4) for both the soil C/N ratio and the standing stock biomass were difficult to assume due to lack of information. An uncertainty estimate of 30% was generally recommended under such conditions (van Oijen et al., 2005). In order to reduce the weight of values close to zero on behalf of large peaks, a minimum measured error that is 10% of the measured value was defined in this study (Klemedtsson et al., 2008). This is also because our intention was to force the model to simulate tree biomass and soil C/N ratio precisely, to better constrain posterior parameter distributions for the respective model approach and site. This allows us to investigate the distributions and uncertainty of key parameters of the respective ECM modeling approaches ("nonlim", "implicit", and "explicit"), as well as analyze model uncertainties and dependencies between parameters. Uncertainties in parameter values are expressed as probability distributions. The posterior probability distributions of parameters are estimated by considering the prior distribution and the

likelihood function in the calibration procedure. The likelihood function is determined by the measured data on output variables and the respective error estimates of the simulated model output. The Bayesian calibration as applied in this study is briefly described below, however, for a detailed description of the general methodology see e.g., van Oijen et al. (2005) or Klemedtsson et al. (2008) and van Oijen et al. (2005).

**2.3.2 Model parameters chosen for calibration**

The different ECM modeling approaches were calibrated for a comprehensive set of key parameters which are chosen according to their function as regulating factors of the C and N fluxes in the plant-soil-mycorrhiza continuum (Table 2). In the "nonlim" approach, the constant N supply parameter *ConstantNsupply* for the spruce tree was a calibration parameter. In the "implicit" approach, the fraction of organic N available for plant uptake ($NUPT_{ORGFRACMAX}$) was included in the calibration based on Svensson et al. (2008a). For the ECM "explicit" approach, all ECM fungal parameters in MYCOFON including ECM growth (C and N assimilation and uptake, C and N losses), overall N uptake and plant N supply, respiration, and littering were calibrated. For all three approaches, the humus decomposition rate ($K_H$), the C/N ratio of microbes ($CN_{mic}$) regulating soil mineralization thus soil N availability, and the fraction of plant C assimilates allocated to the rooting zone ($F_{ROOT}$) regulating ECM fungal growth were additionally calibrated.

**2.3.3 Bayesian calibration of models**

The prior distributions of the parameters were chosen as uniform and non-correlated, with wide ranges of possible values (Table 2). Bayesian calibration combines the prior information about the parameters, and the observational constraints on model outputs to obtain a revised probability distribution or called posterior distribution (Yeluripati et al., 2009). The posterior probability of any parameter vector is proportional to the product of its prior probability and its corresponding data likelihood (eq. (1)). The data likelihood function which determines acceptance of the parameter sets as the posterior distributions, is based on the assumption that the model errors (the differences between simulated and observed values) are normally distributed and uncorrelated (van Oijen et al., 2005). Furthermore, model errors are assumed to be additive so that the log-likelihood function reads:

The data likelihood function which determines the parameter sets being candidate of the posterior distributions is based on the assumption that the model errors, i.e. the differences between simulated and observed values, are normally distributed and uncorrelated (van Oijen et al., 2005). Furthermore, model errors are assumed to be additive so that the log-likelihood function reads:

$$\log L = \sum_{i=1}^{n} \left( -0.5 \left( \frac{y_i - f(\omega_i \cdot \theta_i)}{\sigma_i} \right) - 0.5 \cdot \log(2\pi) \right) - \log(\sigma_i) \tag{1}$$

where $y_i$ = observed values, $f(\omega_i \cdot \theta_i)$ = simulated values for a given model input vector $\omega_i$ and parameter set $\theta_i$,

$\sigma_i$ = standard deviation across the measured replicates, and $n$ = number of variables measured.

In this study, a measured uncertainty of 10% for both the soil C/N ratio and the standing stock biomass data is is used. The uncertainty estimate is low (van Oijen et al., 2005), as our intention was is to force the model to simulate tree biomass and soil C/N ratio precisely, to better constrain posterior parameter distributions for the respective model approaches and sites.

To construct the posterior parameter distribution, many sets of parameter $\theta$ were sampled. In this study, candidate parameter sets  were generated  using the Metropolis-Hastings random walk Markov Chain Monte Carlo (MCMC) algorithm (van Oijen et al., 2005; Vrugt, 2016). Briefly, a parameter ensemble of "walkers" move around randomly and the integrand value at each step was calculated. A few number of tentative steps may further be made to find a parameter space with high contribution to the integral. MCMC thus increases the sampling efficiency by using information about the shape of the likelihood function to preferentially sample in regions where the posterior probability is high (Rubinstein and Kroese, 2016). For each simulation, the model's likelihood  was evaluated for a certain parameter set. After each run, a new parameter set was generated by adding a vector of random numbers ε to the previous parameter vector:

$$q_{i+1} = q_i + e \tag{2}$$

where $\theta_i$ = previous parameter vector, $\theta_{i+1}$ = new parameter vector, and ε = random numbers.

The normally distributed random numbers ε have a mean of zero and a step length of 0.05; i.e., 5% of the prior parameter range as proposed by van Oijen et al. (2005). After a sufficiently long iteration (referred to as the "burn-in" period), the Markov chain reaches a stationary distribution that converges to the joint parameter posterior (Ricciuto et al., 2008). Van Oijen et al. (2005) recommended chain lengths in the order of $10^4$–$10^5$ for modelling forest ecosystems with many observational constraints. In this trial study,  we performed $10^4$ runs for each ECM modeling approach and site . This is because a length of $10^4$ model runs with a burn-in length of around $10^3$ runs results in numerically stable results for our current considered problem. The step sizes used in this study result in acceptance rates between 25 to 50% (Table 4), which is also generally the most efficient range for the MCMC algorithm (Harmon and Challenor, 1997).

**2.3.3 Model parameters chosen for calibration**

The different ECM modeling approaches are calibrated for a comprehensive set of key parameters which are chosen according to their function as regulating factors of the C and N fluxes in the plant-soil-mycorrhiza continuum (Table 3). In the "nonlim" approach, the constant N supply parameter *ConstantNsupply* for the spruce tree is a calibration parameter. In the "implicit" approach, the fraction of organic N available for plant uptake ($NUPT_{ORGFRACMAX}$) is included in the calibration based on Svensson et al. (2008a). For the ECM "explicit" approach, all fungal parameters in MYCOFON including: fungal growth (C and N assimilation and uptake, C and N losses), overall N uptake and plant N supply, respiration, and littering are calibrated. For all three approaches, the humus decomposition rate ($K_H$), the C/N ratio of microbes ($CN_{mic}$) regulating soil mineralization, and the fraction of plant C assimilates allocated to the rooting zone ($F_{ROOT}$), regulating fungal growth are also calibrated. Overall, we include a rather generous number of parameters for Bayesian calibration following Klemedtsson et al. (2008) which who emphasized the importance of a holistic perspective when considering model parameters. Prior distributions of parameters are assumed to be uniform, ; i.e., each value is equally probable, with a given minimum and maximum values (Table 3). Values were chosen based on either previous modeling applications (e.g., plant parameters determined by Svensson et al. 2008a, b), or literature data (Table 3).

**3. Results**

**3.1 Comparison of the three modeling approaches**

**3.1.1 General ability to reproduce tree growth and soil C/N**

Three modeling approaches show different accuracies in reproducing current plant growth and soil C/N ratio after calibration (Table 4B). The posterior model in the "implicit" and "explicit" approaches shows better performance of simulating soil C and N, as indicated by the soil C/N ratio, than the "nonlim" approach. The latter tends to simulate a lower soil C/N ratio, indicated by the negative mean errors (ME, difference between the simulated and measured values) in the posterior model (Table 4B). The ME by the "nonlim" approach is also two to five times higher than that when using the "implicit" or "explicit" approach (Table 4B). The "nonlim" approach tends to overestimate the plant growth as the posterior mean of ME for plant C is always positive, while the "implicit" and "explicit" approaches tend to show an underestimation (Table 4B).

All posterior models underestimate soil C/N for the northern sites which are generally more N limited, but gradually switch to overestimation at the southern sites. The model with the "nonlim" approach simulates better plant growth for the southernmost site, Ljungbyhed, than the other sites. Further, modeled plant growth at Ljungbyhed is overestimated by the "implicit" approach but underestimated when the "explicit" approach is used (Table 4B). The acceptance of model runs in posterior is higher for the "nonlim" (25 to 48%), and "implicit" approaches (42 to 50%), followed by the "explicit" approaches (30 to 33%). No major differences are found for the summed log-likelihood for both calibration variables (Table 4B).

**3.1.2 Ecosystem C-N and N-C fluxes and comparison to measured data**

Modeled major ecosystem N fluxes in the posterior are shown in Figure 3. The modeled N litterfall, uptake and leaching fluxes differ significantly from one modeling approach to another where the "nonlim" approach always gives the highest fluxes. The "explicit" and "implicit" approaches show similar modeled N fluxes for the northernmost site, Lycksele. However, the differences between these two approaches become larger when moving towards south where higher fluxes are simulated by the "explicit" approach (Fig. 3). For instance, modeled N litter production in "explicit" approach increases by 1 to 30% compared to the "implicit" approach, but N losses due to uptake and leaching also increase by 10 to 50% for Lycksele and Ljungbyhed, respectively (Fig. 3). The modeled N pool sizes for these two sites also differ where the "explicit" approach shows a larger mineral N in the soil and a smaller organic N pool compare to the "implicit" approach (Fig. 4).

In general, the "nonlim" approach shows much greater uncertainties in the modeled N fluxes than either the "implicit" or "explicit" approaches. The "nonlim" approach simulates soil sequestration of N up to 2 g N m$^{-2}$ yr$^{-1}$ for all the sites, but much lower or close to zero values are found when using other two modeling approaches (Fig. 3). Therefore, the "nonlim" approach largely overestimates the soil N sequestration. This can be attributed to the assumed "virtual" constant N uptake from the unlimited source. According to our model predictions, this "virtual" N fraction accounts for 20 to 30% of the total plant N uptake. The simulated soil C balance by the "nonlim" approach also contrasts with that of soil N, where the soil sequesters C at the most northern site, Lycksele, but losses C at a rate of 6 to 17 g C m$^{-2}$ yr$^{-1}$ for the other three sites (Fig. 3). Therefore,

soil C and N are not in a steady states, and are decoupled in the "nonlim" approach over the simulated 100, year period.

However, the "implicit" and "explicit" approaches show a strong coupling between soil C and N (Fig. 3). ). That is,i.e., in for the "implicit" approach, Lycksele and Mora soils lose 6 and 5 g C m$^{-2}$ yr$^{-1}$, respectively, while Nässjö and Ljungbyhed soils gain 3 and 13 g C m$^{-2}$ yr$^{-1}$, respectively. Similarly, Lycksele and Mora loses N by 0.2 and 0.1 g N m$^{-2}$ yr$^{-1}$, while Nässjö and Ljungbyhed gain N by 0.3 and 0.6 g N m$^{-2}$ yr$^{-1}$. For the "explicit" approach, soil C and N losses at the two northern sites are slightly higher than that in the "implicit" approach (Fig. 3). In contrast with to the "implicit" approach, the two southern sites also show overall minor C and N losses with large standard deviations (Fig. 3). ModellModeled N litter production increases by 1 to 30% compared to the "implicit" approach, but N losses due to uptake and leaching also increase by 10 to 50 % (for Lycksele and Ljungbyhed, respectively, (Fig. 3). The increased litter addition of easily degradable C and N stimulates microbial activity, thus leading to a higher microbial respiration, which explains the minor losses of C and N in the southern sites in the "explicit" model. The higher N leaching in the "explicit" model can be attributed to a higher uptake from organic N (eqs. 11.5, 11.6, Table 1.B) and, and also a stimulated microbial growth thus increases, net mineralization, both of which leaves more mineral N in the soil (Fig. 4).

Figure 5 shows the modeled major ecosystem C fluxes and comparison with previous results by Svensson et al. (2008a) and measured data from three other Swedish sites (Flakaliden, Knottåsen and Asa, Fig. 2) at comparable latitudes and on comparable soils by Lindroth et al. (2008). The simulated plant gross primary production (GPP) using three approaches all show an increasing trend from the northern sites to the southern sites, due to a more favorable climates and N availability for spruce forest growth. For the studied four sites, the "nonlim" approach simulates the highest GPP followed by the "explicit" and lastly the "implicit" approach. The variation of modeled GPP between the "explicit" and "implicit" approach ranges from 12% in northernmost Lycksele site to 7% in the southernmost Ljungbyhed site (Fig. 5). Simulated GPP in this study are generally higher than that by Svensson et al. (2008a) but comparable with the measured data from Lindroth et al. (2008). It should be noted that the GPP at the southern site, Asa was only measured for one year thus can associated with large uncertainties due to annual variations. Modeled ecosystem respiration generally follows the pattern of GPP. The net ecosystem exchange (NEE) predicted by the three approaches all show an overall atmospheric C uptake for all the sites where the "explicit" approach seems to have a higher uptake strength than the others (Fig. 5). Current estimates of NEE are again within the measured range by Lindroth et al., (2008), although a small net release of C was measured at Knottåsen, likely caused by the abnormal high temperature during those measured years. In addition, explicitly including ECM also increase the soil respiration for the four sites except the northernmost Lyckesele site. The simulated ranges however are somehow smaller than that by Svensson et al. (2008a).

The "nonlim" approach generally shows much higher uncertainties in the modeled N fluxes than either the "implicit" or "explicit" approaches. The "nonlim" approach simulated soil N sequestration up to 2 g N m$^{-2}$ yr$^{-1}$ for all the sites, but much lower or close to zero values were found when using the other two modeling approaches (Fig. 5). The simulated soil C balance by the "nonlim" approach also contrasts with that of soil N, where the soil sequesters C at the northernmost site, Lycksele, but loses C at a rate of 6 to 17 g C m$^{-2}$ yr$^{-1}$ for the other three sites (Fig. 5). Therefore, soil C and N are not in steady state and are decoupled in the "nonlim" approach over the simulated 100-year period. The "implicit" and "explicit" approaches, however, show a strong coupling between soil C and N (Fig. 5). That is, for the "implicit" approach, Lycksele and Mora soils lose 6 and 5 g C m$^{-2}$ yr$^{-1}$

respectively, while Nässjö and Ljungbyhed soils gain 3 and 13 g C m$^{-2}$ yr$^{-1}$ respectively. Similarly, Lycksele and Mora lose N by 0.2 and 0.1 g N m$^{-2}$ yr$^{-1}$, while Nässjö and Ljungbyhed gain N by 0.3 and 0.6 g N m$^{-2}$ yr$^{-1}$. For the "explicit" approach, soil C and N losses at the two northern sites are slightly higher than that in the "implicit" approach. The respective net change in the soil C and N pools of the "implicit" approach corresponds well to the results by Svensson et al. (2008a) who also suggest a small loss of soil C in the north whereas soils in the south gain C. However, when the "explicit" approach is used, the soils in the south are also predicted to lose C and N. Lindroth et al. (2008) found a similar trend in the soil net C change as simulated by the "explicit" approach here, but with a higher loss rate between 24 and 133 g C m$^{-2}$ yr$^{-1}$ (Fig. 5). Overall, our results show that accounting ECM in boreal forest ecosystems can have a considerable impact on the predicted C and N dynamics both for the plants and soil.

 spruce forest to growth, but the "explicit" approach shows a  higher GPP than the "implicit"": i.e., 7% in Ljungbyed to 12% in Lycksele (Fig. 5). C losses from autotrophic respiration are lower in the "explicit" approach (Fig. 5). Trees in the northern regions seems slightly more efficient in taking up C shown by the higher biomass efficiency (NPP/GPP, Fig. 5). Overall, our results show that explicitly accounting for ECM in boreal forest ecosystems can have a considerable impact on the predicted C and N dynamics both for the plants and soil.

**3.2. Posterior parameter distributions**

**3.2.1. Posterior distributions of common parameters**

The posterior distributions differ from the prior uniform distributions for all modeling approaches and parameters, reflecting the efficiency of Bayesian calibration (Fig. 6 and Fig. 7). The posterior *constantNsupply* parameter in the "nonlim" approach shows the lowest values at Lycksele and the highest at Ljungbyhed. This means a higher N supply is necessary at the southern sites to explain the observed tree biomass and soil C/N ratio. No significant differences in parameter values: microbial C/N ratio ($CN_{MIC}$), humus decomposition coefficient ($K_H$), and the fraction of C allocated to roots, $F_{ROOT}$ in the "nonlim" approach are found for the different sites (data not shown). The organic N uptake parameter in the "implicit" and "explicit" approaches ($NUPT_{ORGFRACMAX}$) shows an opposite pattern with the highest values for Lycksele and lowest for Ljungbyhed (Fig. 6).  and larger parameter uncertainties are found  for the "explicit" approach (Fig. 6).  Parameter values for the northern sites also have a much wider range compared with the southern sites (Fig. 6)  also also explains the  larger simulated ME of soil C/N in the  northern sites (Table 4). Both approaches demonstrate that the plant and soil conditions at the northern sites could not be simulated without an enhanced uptake of organic N.

When the "implicit" approach is used, the posterior humus decomposition coefficient $K_H$ shows a higher values for the northern sites and  decreases along the studied transect, demonstrating a modeled modeled higher enhancement of organic matter decomposition  and thus soil mineralization for northern sites (Fig. 7). A less clear tendency towards higher values at the southern sites is identified for the fraction of C allocated to roots, $F_{ROOT}$ parameter . Microbial C/N ratio  $CN_{MIC}$ parameters for both "implicit" and "explicit" approaches show similar posterior

distributions for the three northern sites. However, much lower values are obtained for the southernmost Ljungbyhed site (Fig. 7), reflecting a more soil N rich environment. Overall, parameters are less constrained and only minor differences between sites are found when the "explicit" approach is used (Fig. 7).

**3.2.2 ECM ungal specific parameters**

The posterior distributions of all ECM ungal specific parameters are  constrained to log-normal or normal distributions (data not shown). The mean values of  N uptake parameters ($NORG_{RATE}$, $NH4_{RATE}$, $NO3_{RATE}$) show a decreasing trend from the north to south sites (Fig. 8). This again means an  enhanced ECM fungal N uptake  is necessary to explain the observed soil and plant data at the more N-limited northern sites. Similarly, lower values for the northern and higher values for the southern regions are also found for the minimum ECM fungal C/N ratio parameter ($CN_{FMIN}$). The optimum ratio between ECM and root C content, $FRAC_{OPT}$, tends to be higher at the northern sites and lower at the southern sites, also implying  a model higher ECM biomass at the northern sites (Fig. 8). $MIN_{SUPL}$, the minimum supply of N from ECM to the host plant parameter, does not show a clear trend. Further, differences of the other ECM parameters for the four  sites are minor (Fig. 8).

**3.2.3 Correlation between parameters**

An overview of correlations for all posterior model parameters can be found in the supplementary in Table A2, A3, and A4. Key parameter sets  show correlation with each other (defined here as a Pearson correlation coefficient r $\geq$ 0.3 or $\leq$ -0.3) are  shown in Fig. 9. When the "implicit" approach is used, a significant positive correlation is obtained between the humus decomposition rate, $K_H$, and the fraction of C allocated to rooting zone, $F_{Root}$. The organic N uptake parameter, $NUPT_{ORGFRACMAX}$ and microbial C/N ratio, $CN_{MIC}$ are significantly negative correlated, except for a weak correlation for Ljungbyhed (Fig. 9). A weak correlation between $NUPT_{ORGFRACMAX}$ and $F_{ROOT}$ is also found for the Nässjö site  (see Table A2 ). For the "explicit" approach, the correlation coefficients between $K_H$ and $F_{ROOT}$  are decreased, and there is also  a weaker correlation between NUPT_{ORGFRACMAX} and $CN_{MIC}$ for all sites  compared to the "implicit" approach (Fig. 9). No clear correlation between common and ECM  parameters is obtained. Further, a negative correlation occurred between microbial C/N ratio, $CN_{MIC}$, and the fungal N uptake rates ($Norg_{RATE}$, $NH4_{RATE}$, $NO3_{RATE}$), but only for the Northern sites Lycksele and Mora (Table A4). A moderate correlation is found for $K_H$ and the fungal litter rate, $L$ for Ljungbyhed. Among fungal parameters, the N uptake rates  moderately correlate to the litter production rate, $L$ at the northern sites, but correlations at Nässjö and Ljungbyhed are either weak  or non-existent  (Table A4). Our identified inter-connections and correlations between the parameters in general reflect the complex and interrelated nature of ECM, soil, and plant interactions (He et al., 2016; Klemedtsson et al., 2008). But more importantly, they also highlight the different process interactions and explanations provided by the applied modeling approaches, for the observational constraints .

**4. Discussion**

Our new version of  CoupModel provides a detailed  predictive model framework to explicitly account for ECM in the plant-soil-ECM continuum. Model comparison to two  simpler ECM modeling approaches show  large variations in N dynamic simulations, and that ECM and organic N uptake have to be included in ecosystem models to be able to describe the long-term plant and soil C and N development. Our results confirm that ECM have a substantial effect on soil C and N storage, and can also impact forest plant growth. But more importantly, including them into ecosystem models is both important and feasible.

~~ECM have to be included in ecosystem models ("implicitly or explicitly") to be able to describe the long long term plant, and soil C and N development. Overall, the models perform similarly in the "implicitly or explicitly" approaches, while the "nonlim" approach significantly overestimates soil N uptake. Our results thus confirm that ECM have a substantial effect on soil C and N storage, and can also impact on forest plant growth. But more importantly, including them into ecosystem models is both important and feasible.~~

**4.1 Comparison of the three ECM modeling approach**

The "nonlim" approach in this study shows an overestimation of plant growth and also larger biases in soil N than the "implicit" and "explicit" approaches even after calibration (Table 4). Soil N is expected to reach a steady state over a period of 100 years (Svensson et al., 2008a). Therefore, the "nonlim" approach largely overestimates soil N sequestration which can be attributed to the assumed "virtual" constant N uptake from the unlimited source. According to our model predictions, this "virtual" N fraction accounts for 20 to 30% of the total plant N uptake. A previous CoupModel application by Wu et al. (2012) demonstrated that the "nonlim" approach could possibly describe short-term C and water dynamics for a Finnish forest site. The same "nonlim" approach was also used in Franklin et al. (2014) to simulate Swedish forest biomass growth and its competition with ECM. These seem to suggest that plant growth and the C cycle can be simulated reasonably with the "nonlim" approach, although a slight trend of overestimation is exhibited. However, our modeling exercise further indicates that in this simplified approach soil C and N are uncoupled (Fig. 5) and therefore this approach is not recommended for future long-term soil C and N predictions. This is also reflected in the posterior model parameter distributions where the *constantNSupply* rate parameter shows primary control on the modeled plant growth and soil conditions. Other parameters have minor or no importance for the model results, reflecting an oversimplified model structure of N. Thus, the following discussion focuses on the other two modeling approaches.

Moore et al. (2015) demonstrated that accounting ECM in ecosystem models would substantially affect soil C storage, and that the impact is largely dependent on plant growth. Our study additionally shows that ECM representation in ecosystem models could further feedback into the predicted plant growth through N. When ECM are implicitly included, the model simulates a 48 g C m$^{-2}$ (average of four sites, ±std: 86) lower plant biomass compared to the measured data. When they are explicitly included, the difference becomes even larger, 185 (±35) g C m$^{-2}$ (Table 4). Including ECM explicitly in the model therefore results in decreased plant growth. This somehow differs from the general assumption that growth should be higher in mycorrhized plants, i.e., boreal forest trees, due to optimized nutrient supply (Pritsch et al., 2004; Finlay et al., 2008, see also review by Smith and Read, 2008). This discrepancy can be possibly due to: *1)* an enhanced root litterfall due to a higher turnover of ECM mycelia. Simulated litter production is 50 to 110 g C m$^{-2}$ yr$^{-1}$ higher by the "explicit" approach compared to the "implicit" approach. This could be explained by the conceptually considering the ECM implicitly into the roots where the litterfall rate of roots is c.a. three times lower than that of ECM (calibrated litter rate of ECM is

0.0075 d$^{-1}$, Fig. 8, whereas the litter rate of roots is 0.0027 d$^{-1}$, Table A1(d)). These two approaches thus show large differences in simulating litter production. The discrepancy could also be due to: *2)* an enhanced N immobilization in ECM under N-limited conditions based on the assumption that ECM retain more N in their own biomass in response to plant allocation of newly assimilated C (Nehls et al., 2008). The increasing trend towards the northern sites shown by the constrained optimum ECM fungi C allocation fraction parameter (Fig. 8) also indicates a higher proportional C "investment" by the forest plants in ECM in northern, N limited conditions. The resulting ECM-plant competition for N could then potentially result in decreased plant N uptake, and thus plant growth (Näsholm et al., 2013). Finally, the discrepancy could be due to *3)* biases in simulating ECM N uptake due to model/parameter uncertainties caused by high variability among ECM species and the scarcity of direct measurements in the field (Smith and Read, 2008; Clemmensen et al., 2013). The current "explicit" approach implements many biotic interactions and internal feedbacks within the plant-soil-ECM continuum. However, increasing the number of processes and interactions in an already complex ecosystem model will not necessarily generate more reliable model predictions; as shown here, the parameters in the "explicit" approach have a larger uncertainty range even after calibration. This is also shown by the smaller accepted ratio in the calibration (Table 4) which can be explained by model complexity; i.e. as more parameters are included for calibration, accepted combinations of parameter sets become less likely.

It should also be noted that the "explicit" and "implicit" approaches show considerable difference in estimating soil respiration. Compared to the "implicit" approach, the "explicit" approach simulates a 15% higher soil respiration for the northernmost site and 40% for the southernmost site. The measured soil respiration at Flakaliden is 400 to 590 g C m$^{-2}$ yr$^{-1}$ (Coucheney et al., 2013) and 460 to 520 g C m$^{-2}$ yr$^{-1}$ at Asa (Von Arnold et al., 2005) and these data generally align better with the modeled results by the "explicit" approach (Fig. 5). The estimated higher soil respiration is partly due to the higher litter production and consequently soil respiration in the "explicit" approach, but also due to a higher decomposition of the old organic matter (humus) as shown by the constrained higher humus decomposition coefficient, $K_H$ in the "explicit" approach (Fig. 7). This collaborates well with findings from field measurements and recent modeling studies that ECM are able to degrade complex N polymers in humus layers, thus enhancing soil N transformation under low N conditions (Hartley et al., 2012; Moore et al., 2015; Lindahl and Tunlid, 2015; Parker et al., 2015; Baskaran et al., 2016). The modeled higher soil respiration further explains the minor losses of soil C and N in the southern sites, and also a higher mineral N pool thus higher N leaching in the "explicit" approach (Fig. 3 and Fig. 4).

**4.2 Constrained parameters**

Our constrained parameters generally indicate a shift in the role of ECM from northern to southern sites with a corresponding shift in both climate and soil conditions (Fig. 6, Fig. 7 and Fig. 8). The ECM N uptake parameters show a decreasing trend with increasing soil N availability in the "explicit" approach. This is consistent with observations that at the northern N limited sites, organic N uptake by ECM is highly important for plant growth, becoming less important as N availability increases southwards (e.g., Hyvönen et al., 2008; Näsholm et al., 2013). Shown by the "explicit" approach, the mycorrhization degree of tree roots at Lycksele and Mora (> 90%) is much higher than that of Ljungbyhed (15%), thus the majority of modeled N uptake is through fungal mycelia in northern sites. Similar trend is also found for the organic N uptake parameter in the "implicit" approach, but with a larger site to site difference, thus indicating a stronger response to soil N conditions (Fig. 6). This is expected as more

detailed ECM processes in the "explicit" approach should result in more internal interactions and feedbacks, thus more resilience to the change of environmental conditions.

Most ECM fungal parameters in the "explicit" approach are not – or only weakly – dependent on the differing environmental conditions along the modeled transect, except for the N uptake parameters, $NORG_{RATE}$ and ECM fungal minimum C/N ratio, $CN_{FMIN}$, which show different mean values (Fig. 8). As such, these parameters need to be calibrated carefully when further applying the model to other sites with different soil nutrient levels or climate conditions.

The correlation between the humus decomposition coefficient, $K_H$ and the fraction of C that is allocated to the rooting zone, $F_{ROOT}$, reflects the strong connection of the root-ECM symbiosis and also soil N availability. When ECM are explicitly modeled, this becomes less important, which can be explained by a more detailed internal cycling of N supply and uptake from the ECM; i.e., plant N supply is further regulated by simulated higher root litter input and N uptake from the soil (Fig. 3, Fig. 9). The correlations between the ECM fungal litter rate and ECM fungal N uptake rates in the "explicit" model, and that between fungal N uptake rates, $NORG_{RATE}$ and the microbial C/N ratio, $CN_{MIC}$, for the northern sites (Fig. 9) further indicate the close coupling between ECM fungal N uptake (N loss from the soil) and litter production (N input to the soil). Such an incorporated tight cycle is of major importance for the overall plant N supply, and thus for the C and N dynamics of plant and soil at the N limited sites in the boreal forests. One of the major difficulties of explicitly including ECM in ecosystem models is the unknown turnover of ECM mycelia (Ekblad et al., 2013). Previously reported turnover rates of newly formed mycelia vary from days to weeks, even up to 10 years (Staddon et al., 2003; Wallander et al., 2004), mostly due to the high variability in ECM species and structures (see review by Ekblad et al., 2013). Additionally, root turnover rates can also vary considerably between species, soils, and climate zones (Brunner et al., 2012). Thus far, very few studies have reported parameterization of C and N cycling for ECM in boreal forests. The present model calibration thus provides a key set of ECM parameters that can be further tested by field observations, and more importantly, can in combination with the identified model parameter correlation, act as a guideline for future ECM modeling studies. ECM alter plant-soil C and N dynamics

The "Nnonlim" model in this study shows overestimations of plant growth and also a clear larger biases in soil N than "implicit and explicit" approaches even after calibration (Table 3). A previous CoupModel application by Wu et al. (2012) demonstrated that the "nonlim" approach could can possibly describe short term carbon and water dynamics in for a Finnish forest site. The Ssame approach with open N cycle was also used in Franklin et al. (2014) to simulate the Swedish forest biomass growth and its competition with ECM. It therefore seems that plant growth and thus the C cycle can be simulated reasonably with the "nonlim" approach, plant growth thus C cycle can be simulated reasonably, although with a slightly trend of overestimation ias shownexhibited here. However, our modellmodeling further indicates that this simplified approach has an uncoupled soil C and N in its model structure (Fig. 3) and is thus not recommended for future long term soil C and N predictions. This is also reflected in the posterior model parameter distributions where the constantNSupply rate parameter shows, primary control on the modellmodeled plant growth and soil conditions. Other parameters have minor or no importance for the model results, reflecting an oversimplified soil C and N model structure. Thus, in the following discussion we focusfocuses on the other two modellmodeling approaches.Soil N is expected to reach a steady state over a period of 100 years (Svensson et al., 2008a).

Moore et al. (2015) demonstrated that ECM have a substantially affect effect on soil C storage, and the its impact is largely dependentd on plant growth. The Ppresent study additionally shows that ECM representation in ecosystem models could also feedback on into the predicted plant growth through the feedback of N. As when ECM are implicitly included, the model simulates a 78 (average of four sites, std: 102) g C m$^{-2}$ lower plant biomass comparing compared to the "nonlim" approach. Further, when they are and when explicitly included, the differencet becomes even larger, 214 (50) g C m$^{-2}$ (Table 3). Including ECM in the model thus shows a decreased plant growth. This somehow differs from with what is the generally assumed assumption that growth should be higher in mycorrhized plants, i.e., boreal forests, due to optimized nutrient supply (Pritsch et al., 2004; Finlay et al., 2008, see also review by Smith and Read, 2008). This discrepancy could be possibly due to: ; 1) an enhanced root litterfall, due to a higher turnover of fungal mycelia, shown by a higher litter turnover rate (calibrated litter rate of ECM is 0.0075 d$^{-1}$, Fig. 8, whereas the litter rate of roots is 0.0027 d$^{-1}$, Table A1(d)). When ECM is explicitly considered, litter production is modelled modeled to be higher than in the "implicit" approach (difference from 50 to 110 g C m$^{-2}$ yr$^{-1}$, data not shown). These two modell modeling approaches thus show large differences in simulating litter production. Field data are further needed to clarify this. The discrepancy could also be due to: out; 2) an enhanced N immobilization in ECM under N-limited conditions, due to thebecause ECM retains more N in its their own biomass in response to plant allocation of newly assimilated C (Nehls et al., 2008). The constrained optimum fungi C allocation fraction parameter shows an increasing trend towards the more northern sites (Fig. 8). This, indicating indicates a higher proportional C "investment" by the forest plants on ECM in more northern, N limited conditions. The resulting ECM-plant competition for N could then potentially result in a decreased plant N uptake, and thus plant growth (Näsholm et al., 2013). Finally, ; and the discrepancy could be due to 3) biases in simulating ECM N uptake due to model/parameter uncertainties caused by high variability among fungal species and the scarcity of direct measurements in the field (Smith and Read, 2008, ; Clemmensen et al., 2013). The Ccurrent "explicit" approach implementeds many biotic interactions and internal feedbacks within the plant-soil-ECM continuum. However, increasing the number of processes and interactions in an already complex ecosystem model will not necessarily generate more reliable model predictions; as shown here, the parameters in the "explicit" approach have a larger uncertainty range even after calibration. Thus, future model evaluation, together with more detailed ECM data, are needed is of need to better understand the tightly coupled soil-ECM-plant continuum.

Both approaches simulate the soil C and N stock well (Table 3). The respective net change in the soil C pools of the "implicit" approach corresponds well to the results by Svensson et al. (2008a) who, also suggesting a small loss of soil C in the north while a gain in the south. However, when the "explicit" approach is used, the soils in the south are also predicted to loss lose C and N, mostly due to an enhanced soil respiration (see section 3.1.2). It is difficult to evaluate which approach gives a more realistic prediction as, since field data are not available. However, Lindroth et al. (2008), who measured C fluxes at three sites in Sweden, which are situated at comparable latitudes and on comparable soils, found. They also found a similar trend in the soil net C change as simulated by the "explicit" approach here, but with a higher loss rate of between 24 to and 133 g C m$^{-2}$ year$^{-1}$ (Table 4).

4.2 Parameter and model responses to different environmental conditions

Our modeling results show a consistent pattern with observations (e.g., Hyvönen et al., 2008; Näsholm et al., 2013) that at the northern N limited sites, organic N uptake by ECM is highly important for plant growth, and it becomes less important as N availability increases southwards. As indicated by the "explicit" approach, the mycorrhization degree of tree roots at Lycksele and Mora (> 90%) is much higher than that of Ljungbyhed (15%), thus the majority of modeled N uptake is through fungal mycelia in northern sites. The constrained fungal organic and mineral N uptake parameters also show a decreasing trend (Fig. 8). Similarly, the organic N uptake parameter, $NUPT_{ORGFRACMAX}$, in the "implicit" approach decreases from north to south, but with a more clear site to site difference, indicating a stronger response to environmental conditions (Fig. 6). This is expected as more detailed ECM processes in the "explicit" approach should result in more internal interactions and feedbacks, thus damping the direct environmental regulations. Current modeling also indicates a higher mineralization, shown by the humus decomposition coefficient, $K_H$, in the northern sites. However, the mineralization is also enhanced when ECM is "explicitly" considered (Fig. 7). This collaborates well with findings from the field measurements and recently modeling studies, that ECM are able to degrade complex N polymers in humus layers, thus enhancing soil N transformation under low N conditions (Moore et al., 2015; Lindahl and Tunlid, 2015; Baskaran et al., 2016).

In the "implicit" approach, the humus decomposition coefficient, $K_H$, was found to correlate with the fraction of C that allocates to the rooting zone, $F_{ROOT}$. As ECM are implicitly included in the roots, this correlation therefore indirectly indicates a strong connection of the root-ECM symbiosis and soil N availability. But when ECM are explicitly considered, this becomes less important, again due to a more detailed internal cycling of N supply and uptake from the fungi, i.e., plant N supply is further regulated by simulated higher litter input and N uptake from the soil in the "explicit" model (Fig. 3, Fig. 9). Our modeling shows that fungal litter rates correlate to fungal N uptake rates in the "explicit" model, and that fungal N uptake rates have significant correlations to the microbial C/N ratio, $CN_{MIC}$, for the northern sites (Fig. 9). This indicates the close coupling between fungal N uptake (N loss from the soil) and fungal litter production (N input to the soil). Such an incorporated tight cycle is of major importance for the overall plant N supply, and thus C and N dynamics of plant and soil at the N limited sites in the boreal forests.

Most fungal parameters in the "explicit" approach are not – or only weakly – dependent on the differing environmental conditions along the modeled transect, except for the N uptake parameters and fungal minimum C/N ratio, $CN_{FMIN}$, which show different mean values (Fig. 8). As such, these parameters need to be calibrated carefully when further applying the model to other sites with different soil nutrient levels or climate conditions.

One of the major difficulties of the explicitly inclusion of ECM in ecosystem models is the unknown turnover of fungal mycelia (Ekblad et al., 2013). Previously reported turnover rates of newly formed mycelia vary from days to weeks, even up to 10 years (Staddon et al., 2003; Wallander et al., 2004), mostly due to the high variability in ECM species and structures (see review by Ekblad et al., 2013). Besides, root turnover rates can also vary considerably between species, soils, and climate zones (Brunner et al., 2012). Thus far, very few studies have reported parameterization of C and N cycling for ECM in boreal forests. Our calibration study thus provides a key

set of ECM parameters that can be further tested through field observation, and more importantly, together with the identified correlations with the variables, can act as a guidelines for future ECM modellmodeling studies.

**5. Conclusions**

The key components and features of the Coup-MYCOFONycofon model have been described. The new version of CoupModel simulates C and N fluxes and pools, with the capacity of explicitly accountsing for the links and feedbacks between the ECM, soil, and the plant. The comparison of three commonly ECM modeling approaches which differing in complexity demonstrates that the simple "nonlim" approach cannot describe the measured soil C/N ratio, and also overestimates measured the forest growth. When including ECM either implicitly or explicitly, both models deliver accurate long-term quantitative predictions on forest C and N cycling with simultaneous considerations of the impact of ECM fungi on ecosystem dynamics. However, the "implicit" approach shows a much lower litter production and soil respiration than the "explicit" approach, and both approaches they but slightly underestimate forest growth. The ECM explicit Coup-MYCOFONycofon model provides a more detailed description of internal ecosystems fluxes and feedbacks of C and N fluxes. The constrained ECM parameter distributions presented in this study can be used as as guidelines for future model applications. Overall, Oour model implementation and comparison overall suggest that ecosystem models need to incorporate ECM fungi into their model structure for a better prediction of ecosystem C and N dynamics. and the new version of CoupModel now provides such an alternativeoption.

**6. Code and data availability**

The model and extensive documentation with tutorial excises are freely available from the CoupModel home page http://www.coupmodel.com/ (CoupModel, 2015). The source code can be requested for non-commercial purposes from Per-Erik Jansson (pej@kth.se). CoupModel is written in the C programming language (code also available in Fortran) and run mainly under Windows/Linux systems. Inputs and outputs are in binary format. The version used as the basis for the present development was version 5 from 12 April 2017. The simulation files including the model and calibration set-up, the used parameterization, and corresponding input and validation files can be requested from Hongxing He (hongxing.he@gu.se). However, the majority of the input and output data used for the current modellmodeling is public available publicly, i.e. through SMHI or previous publications, i.e. Svensson et al. (2008). Please contact the first author of this publication or Per-Erik Jansson if you plan an application of the model and further collaboration.The model and extensive documentation with tutorial excises are freely available from the CoupModel home page http://www.coupmodel.com/ (CoupModel, 2015). The source code will be available to download from the home page and a link to a repository for MS Visual studio can also be provided. CoupModel is written in the C programming language and runs mainly under Windows systems. The version used as the basis for the present development was version 5 from 12 April 2017. The simulation files including the model and calibration set-up, the used parameterization, and corresponding input and validation files can be requested from Hongxing He (hongxing.he@gu.se).

*Acknowledgements*: Financial support came from the Swedish Research Council for Environment, Agricultural Sciences and Spatial Planning (FORMAS), the strategic research area BECC (Biodiversity and Ecosystem

services in a Changing Climate, www.cec.lu.se/research/becc), and the Linnaeus Centre LUCCI (Lund University Centre for studies of Carbon Cycle and Climate Interactions).

APPENDIX:

**Table A.1 Model functions describing plant growth, ECM fungal growth, model parameters, and response functions of plant and ECM. Parameters are always entitled with capital letters**

**Table A.1 (a) Description of plant model functions. (i = fine roots, coarse roots, stem, leaves, grain, mobile)**

| No. | Equation |
| --- | --- |

Plant photosynthesis (g C m$^{-2}$ d$^{-1}$):

$$c_{atm \rightarrow plant} = \varepsilon_L \times f(T_1) \times f(CN_1) \times f\left(\frac{E_{ta}}{E_{tp}}\right) \times r_S$$

$\varepsilon_L$= coefficient for radiation use efficiency, $f(T_l)$, $f(CN_l)$, $f(E_{ta}/E_{tp})$ = response functions to leaf temperature, leaf CN, and air moisture (see Table A.1 (c)), $r_s$ = global radiation absorbed by canopy.

Plant maintenance respiration (g C m$^{-2}$ d$^{-1}$):

2.1
$$c_{plantM \rightarrow atm} = c_i \times K_{RMi} \times f(T_l)$$

$c_i$ = C content of each respective plant compartment i (g C m$^{-2}$) and $K_{RMi}$ is a coefficient.

Plant growth respiration (g C m$^{-2}$ d$^{-1}$):

2.2
$$c_{plantG \rightarrow atm} = c_{m \rightarrow i} \times K_{RGi}$$

$c_{m \rightarrow i}$ = C gain (growth) of each plant compartment i (g C m$^{-2}$ d$^{-1}$) and $K_{RGi}$ is a coefficient.

Plant litter production (g C m$^{-2}$ d$^{-1}$):

$$c_{i \rightarrow lit} = c_i \times L_i$$

where $C_i$ is the C content of each plant compartment i (g C m$^{-2}$) and $L_i$ (= 0.0027 d$^{-1}$) is a coefficient.

Plant nitrate and ammonium uptake (g N m$^{-2}$ d$^{-1}$) (only shown for nitrate, equivalent for ammonium):

4.1
$$n_{NO3 \rightarrow plant} = dem_{Nplant} \times r_{NO3}$$
if $f(n_{minavail})$ x $n_{NO3soil} \geq dem_{Nplant}$ x $r_{NO3}$

4.2
$$n_{NO3 \rightarrow plant} = f(n_{minavail}) \times n_{NO3soil} \times dem_{Nplant}$$
if $f(n_{minavail})$ x $n_{NO3soil} \leq dem_{Nplant}$ x $r_{NO3}$

and where

4.3
$$dem_{Nplant} = \sum \frac{c_{a \rightarrow i} - c_{i \rightarrow atm}}{CN_{iMIN}}$$

$f(n_{NO3avail})$ = fraction of soil NO$_3$ available for plant uptake (see response functions Table A.1 (d) ), $n_{NO3soil}$ = soil NO$_3$-N content (g N m$^{-2}$), $dem_{Nplant}$ = plant N demand (g N m$^{-2}$ d$^{-1}$), $r_{NO3}$ = fraction of soil NO$_3$-N in total mineral soil N, $c_{a \rightarrow i}$ = plant C gain ( g C m$^{-2}$ d$^{-1}$), $c_{i \rightarrow atm}$ = respiration of respective plant compartment i (g C m$^{-2}$ d$^{-1}$), $CN_{iMIN}$ = defined minimum C:N ratio of each plant compartment i.

Plant organic N uptake (g N m$^{-2}$ d$^{-1}$) from the humus layer:

4.4
$$n_{hum \rightarrow plant} = dem_{Nplant} \times r_{hum}$$
if $f(n_{humavail})$ x $n_{humsoil} \geq dem_{Nplant}$ x $r_{hum}$

4.5
$$n_{hum \rightarrow plant} = f(n_{humavail}) \times n_{humsoil}$$
if $f(n_{humavail})$ x $n_{humsoil} < dem_{Nplant}$ x $r_{hum}$

$f(n_{humavail})$= response function for plant available N from the humus layer, $n_{humsoil}$ = soil N content in humus layer (g N m$^{-2}$).

**Table A.1 (b) Functions describing processes related to ECM fungal growth and N exchange to plant**

 ECM fungal maximum C supply (g C m$^{-2}$ d$^{-1}$):

5.1 $\quad c_{a \to fungi} = c_{a \to root} \times FRAC_{FMAX} \times f(c_{fungiavail})$

 ECM fungal actual growth (g C m$^{-2}$ d$^{-1}$):

5.2 $\quad c_{a \to fungi} = ((c_{frt} \times FRAC_{OPT}) - c_{fungi}) \times f(n_{supply})$

$c_{a \to root}$ = C available for root and mycorrhiza growth (g C m$^{-2}$ d$^{-1}$), $FRAC_{FMAX}$ = maximum fraction of total root and mycorrhiza available C which is available for ECM, $f(c_{fungiavail})$ = response function which relates ECM growth to N availability, $c_{frt}$ = total root C content (g C m$^{-2}$), $FRAC_{OPT}$ = optimum ratio between root and  ECM C content, $c_{fungi}$ = total ECM C content (g C m$^{-2}$), $f(n_{supply})$ = response function of fungal growth to the amount of N (both mineral and organic N) which is transferred from ECM to plant.

Minimum ECM fungal C supply (g C m$^{-2}$ d$^{-1}$):

5.3 $\quad c_{a \to fungi} = c_{fungi \to atm} \qquad$ if $c_{a \to root} \leq 0$

Total ECM fungal respiration (g C m$^{-2}$ d$^{-1}$):

6.1 $\quad c_{fungi \to atm} = c_{mfungi \to a} + c_{gfungi \to a}$

where $c_{mfungi \to a}$ = ECM fungal maintenance respiration and $c_{gfungi \to a}$ = ECM fungal growth respiration (all in g C m$^{-2}$ d$^{-1}$).

 ECM fungal maintenance respiration (g C m$^{-2}$ d$^{-1}$):

6.2 $\quad c_{mfungi \to a} = c_{fungi} \times K_{RM} \times f(T_l)$

$c_{fungi}$ = total ECM C content (g C m$^{-2}$), $K_{RM}$ = maintenance respiration coefficient, $f(T_l)$ = temperature response function.

 ECM fungal growth respiration (g C m$^{-2}$ d$^{-1}$):

6.3 $\quad c_{gfungi \to a} = c_{a \to fungi} \times K_{RG}$

$c_{a \to fungi}$ = ECM fungal growth (g C m$^{-2}$ d$^{-1}$), $K_{RG}$ = growth respiration coefficient.

 ECM fungal C and N litter production ($c_{fungi \to lit}$: g C m$^{-2}$ d$^{-1}$, $n_{fungi \to lit}$: g N m$^{-2}$ d$^{-1}$):

If ECM fungal growth = simple

7.1 $\quad c_{fungi \to lit} = c_{fungi} \times L$

7.2 $\quad n_{fungi \to lit} = n_{fungi} \times L - nret_{fungi}$

7.3 $\quad nret_{fungi} = n_{fungi} \times L \times (1 - N_{RET})$

$c_{fungi}$ = ECM C content (g C m$^{-2}$), $n_{fungi}$ = ECM fungal N content (g N m$^{-2}$), $L$ = litter rate, $nret_{fungi}$: ECM fungal N which is retained in fungal tissue, $N_{RET}$ = fraction of N retained in fungal tissue from senescence.

If ECM fungal growth = detailed

7.4 $\quad c_{fungi \rightarrow lit} = c_{fungi} \times (FRAC_{MYC} \times L_{MYC} + ((1 - FRAC_{MYC}) \times L_M))$

7.5 $\quad n_{fungi \rightarrow lit} = n_{fungi} \times (FRAC_{MYC} \times L_{MYC} + ((1 - FRAC_{MYC}) \times L_M)) - nret_{fungi}$

7.6 $\quad FRAC_{MYC}$ = fraction of mycorrhizal mycelia in total fungal biomass, $L_{MYC}$ = litter rate of mycorrhizal mycelia, $L_M$ = litter rate of ECM fungal mantle tissue.

Fungal ECM fungal biomass (g C m$^{-2}$, g N m$^{-2}$)

8.1 $\quad c_{fungi} = c_{a \rightarrow fungi} - c_{fungi \rightarrow litter} - c_{fungi \rightarrow a}$

8.2 $\quad n_{fungi} = n_{N \rightarrow fungi} - n_{fungi \rightarrow litter} - n_{fungi \rightarrow plant}$

Mycorrhization degree

9 $\quad m = \dfrac{c_{fungi}}{c_{frt} \times FRAC_{OPT} \times M_{OPT}}$

$c_{frt}$ = fine root biomass (g C m$^{-2}$), $FRAC_{OPT}$ = coefficient defining optimum ratio between ECM fungal and fine root biomass, $M_{OPT}$ = optimum mycorrhization degree, and m=1, when $\dfrac{c_{fungi}}{c_{frt} \times FRAC_{OPT}} \geq M_{opt}$

Uptake and transfer processes of ECM and plant

N transfer from ECM to plant (g N m$^{-2}$ d$^{-1}$)

10.1 $\quad n_{fungi \rightarrow plant} = dem_{Nplant}$ $\qquad$ if dem$_{Nplant}$ ≤ n$_{fungiavail}$

$\quad n_{fungi \rightarrow plant} = n_{fungiavail}$ $\qquad$ if dem$_{Nplant}$ > n$_{fungiavail}$

$dem_{Nplant}$ = plant N demand, $n_{fungiavail}$ = fungal available N for transfer to plant (all g N m$^{-2}$ d$^{-1}$)

10.2 $\quad n_{fungiavail} = n_{fungi} - \dfrac{c_{fungi}}{CN_{FMAX}}$

$c_{fungi}$ = ECM biomass (g C m$^{-2}$), $CN_{FMAX}$ = maximum C:N ratio of fungal tissue, which allows N transfer to plant.

Fungal ECM fungal nitrate and ammonium uptake (given for nitrate, equivalent for ammonium with ammonium specific parameter)

11.1 $\quad n_{NO3 \rightarrow fungi} = n_{NO3 pot \rightarrow fungi} \times r_{NO3} \times f(n_{demfungi})$ $\qquad$ if $N_{NO3pot \rightarrow fungi} < n_{NO3soil} x f(n_{avfungi})$

11.2 $\quad n_{NO3 \rightarrow fungi} = n_{NO3soil} \times f(n_{avfungi})$ $\qquad$ if $N_{NO3pot \rightarrow fungi} > n_{NO3soil} x f(n_{avfungi})$

11.3 $\quad n_{NO3 pot \rightarrow fungi} = NO3_{RATE} \times c_{fungi} \times FRAC_{MYC}$

$n_{NO3pot \to fungi}$=potential ECM nitrate uptake (g N m$^{-2}$ d$^{-1}$), $r_N$ = fraction of ammonium-N and total mineral-N in the soil, $f(n_{demfungi})$= N uptake response to N demand, $n_{NO3soil}$ = soil nitrate content (g N m$^{-2}$), $f(n_{avfungi})$ = N uptake response to soil availability, $NO3_{RATE}$ = nitrate specific uptake rate (g N m$^{-2}$ d$^{-1}$), $c_{fungi}$ = ECM fungal biomass (g C m$^{-2}$), $FRAC_{MYC}$ = fraction of mycorrhizal mycelia in total  ECM biomass.

 ECM fungal organic N uptake from litter and humus (given for litter, equivalent for humus with humus specific parameter)

11.4 $\quad n_{lit \to fungi} = n_{litpot \to fungi} \times r_{lit} \times f(n_{demfungi})$ $\qquad$ if $n_{litpot \to fungi}$ x $r_{lit}$ < $n_{litsoil}$ x$f(n_{litavfungi})x r_{lit}$

11.5 $\quad n_{lit \to fungi} = n_{litsoil} \times f(n_{litavfungi}) \times r_{lit}$ $\qquad$ if $n_{litpot \to fungi}$ x $r_{lit}$> $n_{litsoil}$ x$f(n_{litavfungi})$ x$r_{lit}$

11.6 $\quad n_{litpot \to fungi} = LIT_{RATE} \times c_{fungi} \times FRAC_{MYC}$

where $n_{litpot \to fungi}$ = potential ECM organic N uptake from litter (g N m$^{-2}$ d$^{-1}$), $r_{lit}$ = fraction of litter-N in total organic-N in the soil, $f(n_{demfungi})$= N uptake response to N demand, $n_{litsoil}$ = soil litter content (g N m$^{-2}$), $NLIT_{RATE}$ = litter specific uptake rate (g N g C$^{-1}$ d$^{-1}$), $c_{fungi}$ = ECM fungal biomass (g C m$^{-2}$), $FRAC_{MYC}$ = fraction of mycorrhizal mycelia in total  ECM biomass.

**Table A1 (c) Overview of response functions of plant and ECM fungal growth and N uptake**

| No. | Equation |
| --- | --- |

**Plant response to air temperature**

12 $\quad f(T_l) =$
$$\begin{cases} 0 & T_l < P_{min} \\ (T_l - p_{min}) / (p_{O1} - p_{min}) & p_{min} \leq T_l \leq p_{O1} \\ 1 & p_{O1} < T_l < p_{O2} \\ 1- (T_l - p_{O2}) / (p_{max} - P_{O2}) & p_{O2} < T_l < p_{max} \\ 0 & T_l > p_{max} \end{cases}$$

where $T_l$ = leaf temperature (°C) and  $P_{min}$ (-4°C), $P_{O1}$ (10°C), $P_{O2}$ (25°C),  $P_{max}$ (40°C) are coefficients.

**Photosynthetic response to leaf C/N ratio**

13 $\quad f(CN_l) = 1 + (\dfrac{cn_l - p_{CNOPT}}{p_{COPT} - p_{CNTH}})$
$$\begin{cases} 1 & CN_l < p_{CNOPT} \\ & p_{CNTH} \leq CN_l \geq p_{CNOPT} \\ 0 & CN_l > p_{CNTH} \end{cases}$$

[revised manuscript text omitted]

Read, D. and Perez-Moreno, J.: Mycorrhizas and nutrient cycling in ecosystems - a journey towards relevance?, New Phytologist, 157, 475–492, 2003.

Ricciuto, D. M., Davis, K. J. and Keller, K., A Bayesian calibration of a simple carbon cycle model: The role of 1635 observations in estimating and reducing uncertainty: Global Biogeochemical Cycles, 22, GB2030, doi:10.1029/2006GB002908, 2008.

Rubinstein, R. Y. and Kroese, D. P.: Simulation and the Monte Carlo Method, Wiley, Hoboken, Third edition, 2016.

Schulze, E. D., Luyssaert, S., Ciais, P., Freibauer, A., Janssens, I. A., Soussana, J. F., Smith, P., Grace, J., Levin, 1640 I., Thiruchittampalam, B., Heimann, M., Dolman, A. J., Valentini, R., Bousquet, P., Peylin, P., Peters, W.,

Rodenbeck, C., Etiope, G., Vuichard, N., Wattenbach, M., Nabuurs, G. J., Poussi, Z., Nieschulze, J. and Gash, J. H.: Importance of methane and nitrous oxide for Europe's terrestrial greenhouse-gas balance, Nature Geoscience, 2, 842–850, 2009.

Sitch, S., Smith, B., Prentice, I. C., Arneth, A., Bondeau, A., Cramer, W., Kaplan, J. O., Levis, S., Lucht, W., Sykes, M. T., Thonicke, K. and Venevsky, S.: Evaluation of ecosystem dynamics, plant geography and terrestrial carbon cycling in the LPJ dynamic global vegetation model, Global Change Biology, 9, 161–185, 2003.

Smith, B., Samuelsson, P., Wramneby, A. and Rummukainen, M.: A model of the coupled dynamics of climate, vegetation and terrestrial ecosystem biogeochemistry for regional applications, Tellus Series A-dynamic Meteorology and Oceanography, 63, 87–106, 2011.

Smith, S. and Read, D.: Mycorrhizal Symbiosis, Academic Press, London, Third Edition, 2008.

Staddon, P., Ramsey, C., Ostle, N., Ineson, P. and Fitter, A.: Rapid turnover of hyphae of mycorrhizal fungi determined by AMS microanalysis of C-14, Science, 300, 1138–1140, 2003.

Svensson, M., Jansson, P. E. and Kleja, D. B.: Modelling soil C sequestration in spruce forest ecosystems along a Swedish transect based on current conditions, Biogeochemistry, 89, 95–119, 2008a.

Svensson, M., Jansson, P. E., Gustafsson, D., Kleja, D. B., Langvall, O. and Lindroth, A.: Bayesian calibration of a model describing carbon, water and heat fluxes for a Swedish boreal forest stand, Ecological Modelling, 213, 331–344, 2008b.

Taylor, A. and Alexander, I.: The ectomycorrhizal symbiosis: life in the real world, Mycoologist, 19, 102–112, 2005.

Thornley, J. and Cannell, M.: Modelling the components of plant respiration: representation and realism, Annalys of Botany, 85, 55–67, 2000.

Vitousek, P. M. and Howarth, R. W.: Nitrogen limitation on land and in the sea: How can it occur? Biogeochemistry, 13: 87-115, 1991.

von Arnold, K., Weslien, P., Nilsson, M., Svensson, B., Klemedtsson, L.: Fluxes of $CO_2$, $CH_4$ and $N_2O$ from drained coniferous forests on organic soils, For. Ecol. Manage., 210, 239-254, 2005.

Vrugt, J. A.: Markov chain Monte Carlo simulation using the DREAM software package: Theory, concepts, and MATLAB implementation, Environmental Modelling & Software, 75, 273-316, 2016.

Wallander, H. and Nilsson, L.: Direct estimates of C:N ratios of ectomycorrhizal mycelia collectes from Norway spruce forest soils, Soil Biology & Biochemistry, 35, 997–999, 2003.

Wallander, H., Göransson, H. and Rosengreen, U.: Production, standing biomass and natural abundance of 15 N and 13 C in ectomycorrhizal mycelia collected at different soil depth in two forest types, Oecologia, 139, 89–97, 2004.

Wallander, H., Fossum, A., Rosengren, U., Jones, H.: Ectomycorrhizal fungal biomass in roots and uptake of P from apatite by Pinus sylvestris seedlings growing in forest soil with and without wood ash amendment, Mycorrhiza, 15(2), 143-148, 2005.

Wallander, H., Ekblad, A. and Bergh, J.: Growth and carbon sequestration by ectomycorrhizal fungi in intensively fertilized Norway spruce forests, Forest Ecology and Management, 262, 999–1007, 2011.

Wallander, H., Ekblad, A., Godbold, D. L., Johnson, D., Bahr, A., Baldrian, P., Bjork, R. G., Kieliszewska-Rokicka, B., Kjoller, R., Kraigher, H., Plassard, C. and Rudawska, M.: Evaluation of methods to estimate

1680  production, biomass and turnover of ectomycorrhizal mycelium in forests soils - A review, Soil Biology & Biochemistry, 57, 1034–1047, 2013.

Wallenda, T. and Kottke, I.: Nitrogen deposition and ectomycorrhizas, New Phytologist, 139, 169–187, 1998.

Wu, S. H., Jansson, P. -E., and Kolari, P.: The role of air and soil temperature in the seasonality of photosynthesis and transpiration in a boreal Scots pine ecosystem, Agricultural and Forest Meteorology, 156, 85-103,

1685  10.1016/j.agrformet.2012.01.006, 2012.

Yeluripati, J. B., van Oijen, M., Wattenbach, M., Neftel, A., Ammann, A., Parton, W. J. and Smith, P.: Bayesian calibration as a tool for initialising the carbon pools of dynamic soil models, Soil Biology and Biochemistry, 41, 2579-2583, 2009.

1690

[Figure]

[Figure]

**Figure 11 A simplified overview of C and N fluxes between plants, mycorrhiza fungi, and the soil in the Coup-MYCOFON model. Light blue indicates the newly implemented MYCOFON model**

1695

[Figure]

**Figure 2 Location of the four study sites in Sweden modified from Svensson et al. (2008a). Filled cycles represent the studied four sites. Open circles are the measured sites reported in Lindroth et al. (2008) used for comparison**

Figure 2 Position of the four study sites in Sweden

1735

[Figure]

[Figure]

**Figure 33** Soil N fluxes for the nonlim (grey columns, left), implicit (white, 2nd left column), and explicit (black, 3rd left column) model approaches, same color scheme used for the other figures. Presented are the major N inputs (N deposition, total N litter production, added to the soil litter pool by fresh litter), and and outputs (N uptake from the plant/ECM fungi, N leaching), and the net change in the total soil N pool (mineral and organic). For C, the net change is presented (right column). Error bars indicate the 90th percentile of accepted model runs (posterior). Units for N are g N m⁻² yr⁻¹ and g C m⁻² yr⁻¹ for C

1745

implicit

explicit

**Figure 44** Average soil organic and mineral  content  in the implicit ECM model (upper graph) and explicit ECM model (lower graph) for the two sites Lycksele and Ljungbyhed. Box plots indicate the median (bold line), the 25th and 75th percentile (bars), and the 10th and 90th percentile (whiskers)

1750

[Figure]

[Figure]

**Figure 55 GPP (bars), Rh/GPP ratio (triangle), and NPP/GPP ratio (cross circles) for all four sites simulated with the implicit (left) and explicit (right) ECM model approachSimulated GPP, ecosystem respiration, NEE, soil respiration, change in soil C and change in soil N for all four sites with the three ECM modeling approaches and also compared with modelled data by Svensson et al. (2008a) and measurements by Lindroth et al. (2008)and the net change in the total soil N pool (mineral and organic). For C, the net change is presented (right column).**

[Figure]

[Figure]

1765 **Figure 6 Posterior parameter distributions for N uptake parameters: constant N supply rate in the "nonlim" approach ( grey) and organic N uptake capacity in the implicit (white) and explicit (black) ECM model approaches. Distributions are presented as box plots over the prior range of variation (corresponding to the range in the x-axis). Box plots depict the median (bold line), the 25th and 75th percentile (bars), and the 10th and 90th percentile (whiskers)**

1770

[Figure]

1775 Figure 7 Posterior parameter distributions for common parameters using the implicit (top: white ) and explicit (bottom:  ) ECM approaches for four different sites from North to South. Distributions are presented as box plots over the prior range of variation (corresponding to the range in the x-axis). Box plots depict the median (bold line), the 25th and 75th percentile (bars), and the 10$^{th}$ and 90$^{th}$ percentile (whiskers). The parameters

shown are: $K_H$: the humus decomposition coefficient, $F_{Root}$: the fraction of C assimilates distributed to the roots, and ECM, $CN_{MIC}$: the microbial C/N ratio

[Figure]

**Figure 8 Posterior parameter distributions of  fungal specific parameters (from top left to bottom right): organic N uptake rate ($NORG_{RATE}$), ammonium uptake rate ($NH4_{RATE}$), respiration coefficient ($K_{RM}$),  fungal litter rate coefficient (the rate at which mycelia and mantle die and add to the soil litter pool, L), minimum  fungal C/N ratio ($CN_{FMIN}$),   minimum N supply to plant ($MIN_{SUPL}$), optimum ratio between   and root C content ($FRAC_{OPT}$), and N sensitivity coefficient ($NAVAIL_{COEF}$). Distributions are presented as box plots over the prior range of**

1790     variation (corresponding to the range in the x-axis). Box plots depict the median (bold line), the mean (black point), the 25th and 75th percentile (bars), and the $10^{th}$ and $90^{th}$ percentile (whiskers)

[Figure]

[Figure]

1795

**Figure 9 Correlation between model parameters, given as the Pearson correlation coefficient, for the implicit (white) and explicit ECM (black) approaches. Top left: correlation between humus decomposition coefficient ($K_H$) and the fraction of C assimilates (GPP) directed to ECM and roots ($F_{ROOT}$). Top right: C/N of microbes ($CN_{MIC}$) and fraction of organic N available for uptake ($NUPT_{ORGFRACMAX}$). Correlation between ECM fungal parameters: bottom left: humus decomposition coefficient ($K_H$) and ECM fungal litter rate (L). Bottom right:  ECM organic N uptake ($NORG_{RATE}$) and C/N of microbes ($CN_{MIC}$)**

1800

**Table 1 Main characteristics of previous ecosystem models include ECM**

| Models | Time step | Elements included | Differentiation in mycelia and mantle | Organic matter decomposition | C allocation | Plant N uptake | Is sensitive to soil N |
|---|---|---|---|---|---|---|---|
| ANAFORE, Deckmyn et al. (2011) | hourly | C, N | No | Yes | Fraction of C allocated to roots, regulated by water and N | Function of the available mineral and organic N pools | No |
| MoBiLE and Mycofon, Meyer et al. (2010, 2012) | Daily | C, N | Yes | No | A certain ratio between root and ECM biomass exists to reach the optimum degree of mycorrhization, regulated by soil N and temperature | Separated root and mycelia mineral N uptake and regulated by plant and ECM N demand | Yes |
| MySCaN, Orwin et al. (2011) | Daily | C, N, P | No | Yes | Constant fraction of plant C assimilates, modified by nutrients | Driven by C to nutrient ratios in pools | No |
| Moore et al. (2015) model | Monthly | C | No | Yes | Constant fraction of plant C assimilates | | No |
| Baskaran et al. (2016) model | Annual | C, N | No | No | Constant fraction of plant C assimilates | Root inorganic N uptake by Michaelis-Menten function and ECM N uptake by ECM C to N ratio | No |
| Coup-MYCOFON (This study) | Daily | C, N | Yes | No | Similar to MoBiLE | Similar to MoBiLE, but allows organic N uptake for ECM | Yes |

1805

**Table 1 2** **Maximum and minimum parameters values prior to Bayesian calibration for the nonlim, implicit, and explicit model approaches**

**A.** Common parameters (all three approaches, including the "implicit" approach)

| Parameter | Unit | Min | Max |
|---|---|---|---|
| *Humus decomposition* | | | |
| $K_H$ | $d^{-1}$ | 0.0001 | 0.001 |
| *Fraction of organic N available for uptake* | | | |
| $NUPT_{ORGFRACMAX}$ | $d^{-1}$ | 0.000001 | 0.0001 |
| *Fraction of root C allocation in mobile C* | | | |
| $F_{ROOT}$ | $d^{-1}$ | 0.4 | 0.6 |
| *C/N ratio of decomposing microbes* | | | |
| $CN_{MIC}$ | $d^{-1}$ | 15 | 25 |

1810 **B.** Parameters of the "nonlim" approach

| Parameter | Unit | Min | Max |
|---|---|---|---|
| *Plant N Supply* | | | |
| ConstantNSupply | - | 0.1 | 0.7 |

**C.**  ECM fungal parameters of the "explicit" approach

| Parameter | Unit | Min | Max |
|---|---|---|---|
| * ECM N uptake* | | | |
| $NORG_{RATE}$ | g N gdw$^{-1}$ d$^{-1}$ | 0.000001[a] | 0.0001 |
| $NH4_{RATE}$ | g N gdw$^{-1}$ d$^{-1}$ | 0.000001[a] | 0.0001 |
| $NO3_{RATE}$ | g N gdw$^{-1}$ d$^{-1}$ | 0.000001[a] | 0.0001 |
| * ECM respiration coefficient* | | | |
| $K_{RM}$ | $d^{-1}$ | 0.0002[b] | 0.05 |
| * ECM litter rate* | | | |
| $L$ | $d^{-1}$ | 0.0008[c] | 0.01 |
| *Minimum ECM fungal C/N ratio* | | | |
| $CN_{FMIN}$ | $d^{-1}$ | 5[d] | 10 |
| * ECM minimum N supply to plant* | | | |
| $MIN_{SUPL}$ | $d^{-1}$ | 0.1[e] | 0.9 |
| *Optimum ECM fungi C allocation fraction* | | | |
| $FRAC_{OPT}$ | $d^{-1}$ | 0.1[f] | 0.3[f] |

| | | | |
|---|---|---|---|
| *N sensitivity coefficient* | | | |
| NAVAIL$_{COEF}$ | d$^{-1}$ | 0.0001 | 0.001 |

[a] Plassard et al. (1991), Chalot et al. (1995), and Smith and Read (2008)

[b] Set equally to trees according to Thornley and Cannell (2000)

[c] Staddon et al. (2003) and Ekblad et al. (2013)

[d] Högberg and Högberg (2002) and Wallander and Nilsson (2003)

[e] Estimated

[f] Leake (2007), Staddon et al. (2003), and Johnson et al. (2005)

Table 2 3 **Climatic and soil data, and initial settings of the four study soils applied in all model approaches**

| Sites | Location | Altitude (m asl) | Driving data | | | | | Calibration data | | |
|---|---|---|---|---|---|---|---|---|---|---|
| | | | Air temperature[a] (°C) | Precipitation[a] (mm) | N deposition (kg N ha$^{-1}$ yr$^{-1}$) | Soil C (g C m$^{-2}$) | Soil N (g N m$^{-2}$) | Soil C/N [c] | Standing stock (g C m$^{-2}$)[b,c] | N deposition (kg N ... year$^{-1}$)[d] |
| Lycksele | 64°59'N 18°66'E | 223 | 0.7 | 613 | 1.5 | 7006 | 223 | 31.5 | 5371 | 1.5 |
| Mora | 61°00'N 14°59'E | 161 | 3.3 | 630 | 3.5 | 8567 | 295 | 29.1 | 7815 | 3.5 |
| Nässjö | 57°64'N 14°69'E | 305 | 5.2 | 712 | 7.5 | 9995 | 367 | 27.2 | 10443 | 7.5 |
| Ljungby-hed | 56°08'N 13°23'E | 76 | 7.1 | 838 | 12.5 | 10666 | 539 | 19.8 | 11501 | 12.5 |

[a] 30-year (1961 to 1991) annual average /sum

[b] according According to Skogsdata for a 100- year year-old forest (2003: http://www.slu.se/en/webbtjanster-miljoanalys/forest-statistics/skogsdata/)

[c] used Used as calibration parameter

[d] used Used as driving data

1825

Table 4 Prior values of variables used for model calibration and accepted relative uncertainty (A), and posterior model performance indicators (B): mean error (ME) between simulated and measured values, standard variation of ME (std), and summed log-likelihood of all accepted runs for simulated standing plant biomass (g C m$^{-2}$) and soil C/N ratio after the 100 year simulation period

**A — PRIOR**

| | Plant biomass (g C m$^{-2}$) | | Soil C/N ratio | |
|---|---|---|---|---|
| | Mean | Relative uncertainty (%) | Mean | Relative uncertainty (%) |
| Lycksele | 5371 | 0.10 | 32 | 0.10 |
| Mora | 7815 | 0.10 | 29.1 | 0.10 |
| Nässjö | 10443 | 0.10 | 27.2 | 0.10 |
| Ljungbyhed | 11501 | 0.10 | 19.8 | 0.10 |

**B — POSTERIOR**

| | | Plant biomass (g C m$^{-2}$) | | | Soil C/N ratio | | | Runs accepted (%) |
|---|---|---|---|---|---|---|---|---|
| | | ME | std | loglike | ME | std | loglike | |
| *nonlim* | Lycksele | 37.6 | 531.1 | -7.7 | -5.8 | 1.3 | -3.8 | 25 |
| | Mora | 38.7 | 1098.2 | -8.4 | -3.9 | 1.4 | -3.0 | 41 |
| | Nässjö | 42.2 | 1021.3 | -8.3 | -2.7 | 1.6 | -2.6 | 48 |
| | Ljungbyhed | 1.0 | 1155.6 | -10.2 | 0.3 | 1.8 | -2.1 | 48 |
| *implicit* | Lycksele | -107.2 | 535.0 | -7.7 | -1.1 | 3.3 | -2.7 | 42 |
| | Mora | -98.3 | 787.1 | -8.1 | -1.1 | 2.7 | -2.5 | 45 |
| | Nässjö | -86.0 | 1036.2 | -8.0 | -1.0 | 2.5 | -2.4 | 46 |
| | Ljungbyhed | 100.1 | 1143.2 | -8.5 | 0.5 | 1.6 | -2.0 | 50 |
| *explicit* | Lycksele | -162.3 | 534.9 | -7.7 | -0.5 | 3.4 | -2.7 | 29 |
| | Mora | -215.4 | 809.1 | -8.2 | -0.3 | 2.7 | -2.4 | 32 |
| | Nässjö | -222.3 | 1041.2 | -8.1 | 0.0 | 2.5 | -2.3 | 30 |
| | Ljungbyhed | -139.0 | 1137.6 | -8.5 | 1.0 | 1.7 | -2.1 | 32 |

Table 4 Comparison between modellmodeled soil C and N of this study and literature value

| Reference | Site | Ecosystem type | Forest age (years) | Soil C change (g C m$^{-2}$yr$^{-1}$) | Soil N change (g N m$^{-2}$yr$^{-1}$) |
|---|---|---|---|---|---|
| Svensson et al. 2008a | Lycksele | Coniferous on podzol | 100 | -5 | |
| | Mora | | | -2 | |
| | Nässjö | | | 9 | |
| | Ljungbyhed | | | 23 | |
| Lindroth et al. 2008 | Flakaliden | Coniferous on podzol | 39-42 (in 2002) | -79[a] | |
| | Knottåsen | | | -133[a] | |
| | Asa | | | -24 | |
| This study | Lycksele | Coniferous on podzol | 100 | -6 to 13.1[b] | -0.2 to 0.6[b] |
| | Mora | | | | |
| | Nässjö | | | -8.7 to -1.6[c] | -0.2 to -0.1[c] |
| | Ljungbyhed | | | | |

[a] mean Mean of the highest and lowest error estimates

[b] implicit Implicit approach

[c] explicit Explicit approach

1840

---

## Author Response (AR2)

Dear authors,

thanks for preparing a revised version of your manuscript addressing reviewers' comments. The new version addresses these comments well, but there still remains a few places where grammar still needs to be improved. I marked in the attached document places where grammar should be revised.

More importantly, I encountered a problem in the interpretation of the correlations of the posterior parameter values. These correlations are an indication of problems of poor-identifiability of the model with respect to the available information for parameterization. I discuss this problem in a previous paper (Sierra et al. 2015, https://doi.org/10.1016/j.soilbio.2015.08.012). Please see the references I mention in this publication where the problem is discussed in more detail, and make changes to the interpretation of your correlations. You interpret them as complex connections among process, but this interpretation is wrong. Fortunately, your correlations are not so high, so the identifiability problem is not severe, but the issue needs to be discussed nevertheless.

Reply:

Sorry for the grammar issues and they are now revised. Also thanks for pointing out the parameter correlation issue. We believe this is not a unique characteristic of present model or present data used in our study, but instead rather a general phenomenon for detailed process-based modeling studies. We therefore have removed the previous wrong interpretation and added a revised discussion in section 4.2 to address this better. As following:

"Most of our constrained parameter distributions are not sharply peaked, but instead rather flat and few parameters show high covariance (Fig. 6, Fig. 7, Fig. 8 and Fig. 9). This is, however not a unique characteristic of the CoupModel or current used data constraints and indeed, has been previously demonstrated in numerous studies with ecosystem models of similar complexity (e.g., He et al., 2016; Klemedtsson et al., 2008; Wang et al., 2001). This on one hand, generally reflects the equifinality of models (Beven, 2006), where multiple parameter sets can lead to equally well representations of the system. One the other hand, it also indicates poor identifiability of the calibrated parameters with respect to the available information for parameterization. Here, we again show that given the same data constraints, the parameter identifiability decrease with increasing model complexity (Sierra et al., 2015). In our study, the correlation between the humus decomposition coefficient, $K_H$ and the fraction of C that is allocated to the rooting zone, $F_{ROOT}$, is smaller when ECM are modeled explicitly than implicitly (Fig. 9). However, the correlations between the ECM fungal litter rate and ECM fungal N uptake rates, and that between fungal N uptake rates, $NORG_{RATE}$ and the microbial C/N ratio, $CN_{MIC}$ (Fig. 9) further indicate these ECM fungal parameters in the more complex "explicit" model cannot be well identified without adding new dataset as additional constraints. One of the major challenges of explicitly including ECM in ecosystem models is the still sparse information about ECM, e.g., unknown turnover of ECM mycelia (Ekblad et al., 2013). Previously reported turnover rates of newly formed mycelia vary from days to weeks, even up to 10 years (Staddon et al., 2003; Wallander et al., 2004), mostly due to the high variability in ECM species and structures (see review by Ekblad et al., 2013). Additionally, root turnover rates can also vary considerably between species, soils, and climate zones (Brunner et al., 2012). Thus far, very few studies have reported parameterization of C and N cycling for ECM in boreal forests. The present model calibration thus provides a key set of ECM parameters that can be further tested by field observations, and more importantly, can act as a prior for future ECM modeling studies. "